# Causal Abstraction Learning based on the Semantic Embedding Principle

Gabriele D'Acunto [1 2]   Fabio Massimo Zennaro [3]   Yorgos Felekis [4]   Paolo Di Lorenzo [1 2]

## Abstract

Structural causal models (SCMs) allow us to investigate complex systems at multiple levels of resolution. The causal abstraction (CA) framework formalizes the mapping between high- and low-level SCMs. We address CA learning in a challenging and realistic setting, where SCMs are inaccessible, interventional data is unavailable, and sample data is misaligned. A key principle of our framework is *semantic embedding*, formalized as the high-level distribution lying on a subspace of the low-level one. This principle naturally links linear CA to the geometry of the *Stiefel manifold*. We present a category-theoretic approach to SCMs that enables the learning of a CA by finding a morphism between the low- and high-level probability measures, adhering to the semantic embedding principle. Consequently, we formulate a general CA learning problem. As an application, we solve the latter problem for linear CA; considering Gaussian measures and the Kullback-Leibler divergence as an objective. Given the nonconvexity of the learning task, we develop three algorithms building upon existing paradigms for Riemannian optimization. We demonstrate that the proposed methods succeed on both synthetic and real-world brain data with different degrees of prior information about the structure of CA.

**Code:** https://github.com/SPAICOM/calsep

## 1. Introduction

*Causal modeling and reasoning* are key to trustworthy and responsible AI (Ganguly et al., 2023; Rawal et al., 2024; Qi et al., 2024). *Structural causal models* (SCMs) provide

[1]Department of Information Engineering, Electronics and Telecommunications, Sapienza University, Rome, Italy [2]National Inter-University Consortium for Telecommunications (CNIT), Parma, Italy [3]Department of Informatics, University of Bergen, Bergen, Norway [4]Department of Computer Science, University of Warwick, Coventry, UK. Correspondence to: Gabriele D'Acunto <gabriele.dacunto@uniroma1.it>.

*Proceedings of the 42$^{nd}$ International Conference on Machine Learning*, Vancouver, Canada. PMLR 267, 2025. Copyright 2025 by the author(s).

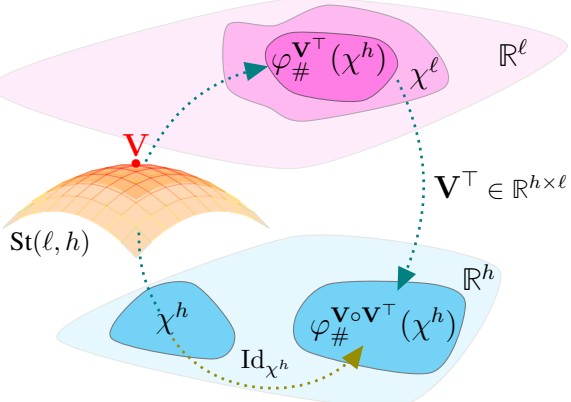

**The Semantic Embedding Principle (SEP)**
*Causal Abstractions must preserve high-level causal knowledge when embedded in the low-level.*

Figure 1: **Pictorial representation of SEP for linear CA. A linear map V belonging to the Stiefel manifold embeds a high-level causal knowledge $\chi^h$ into a low-level one, viz. $\chi^\ell$, identifying an embedded causal knowledge $\varphi_{\#}^{\mathbf{V}^\top}(\chi^h)$. Then, a linear CA $\mathbf{V}^\top$ abstracts $\varphi_{\#}^{\mathbf{V}^\top}(\chi^h)$, yielding a causal knowledge identical to $\chi^h$. Notice that the arrow $\mathrm{Id}_{\chi^h}$ underlines that commutativity holds only in one direction, that is, SEP does not imply $\varphi^{\mathbf{V}^\top \circ \mathbf{V}}(\chi^\ell) = \chi^\ell$.**

a widely adopted framework for causal reasoning (Pearl, 2009). While canonical causal theory focuses on a single SCM, scientific research often requires multiple representations of the same system at different levels of resolution. For example, biological processes can be studied at the molecular level (e.g., gene expression), cellular level (e.g., metabolic pathways), or organism level (e.g., physiological responses), each offering a different view of the same underlying system. *Causal abstraction* (CA) theory (Rubenstein et al., 2017; Beckers & Halpern, 2019) formalizes mappings between SCMs at different abstraction levels, enforcing rigorous *consistency* requirements. This makes CA a powerful tool for transitioning between resolutions, synthesizing causal evidence, and selecting the most parsimonious representation for a given task. However, CAs are unknown in practice, underscoring the need for advancing CA learning from data (Zennaro et al., 2023).

**Related works.** Seminal works on CA have focused on defining and assessing given CA maps (Rubenstein et al., 2017; Beckers & Halpern, 2019). Our approach builds on the $\alpha$-abstraction category-theoretic framework introduced by (Rischel, 2020), which neatly separates the structural and functional components of the CA. From a learning perspective, several methods have been proposed which rely on restrictive assumptions. In our work, we transform them into *non-assumptions* (NA). (Zennaro et al., 2023) addresses the learning problem under (NA1) complete specification of SCMs, which, in reality, is rarely available. (Felekis et al., 2024) assumes (NA2) knowledge of causal DAGs, which are often unknown in many applications. (Dyer et al., 2024) relies on the (NA3) availability of interventional data, which may be infeasible or unethical to obtain. (Kekić et al., 2024; Massidda et al., 2024) make (NA4) functional assumptions on the SCMs, such as linearity. (Massidda et al., 2024) implicitly assumes (NA5) alignment between data generated by two models, which requires tight coordination in sample collection. Conversely, we work under the realistic and pragmatic assumption that (A1) *at least partial prior knowledge of the structure of a CA is available.* (A1) is met in different application domains, such as neuroscience. For instance, consider the learning of a CA between two brain SCMs, the first referring to some brain region of interest (ROIs), the second to the brain lobes. A map between ROIs and brain lobes is implicitly defined by the location of ROIs, and so it would be natural to try to exploit such prior knowledge when learning the CA. Finally, we build on top of different continuous optimization frameworks, working in both the Euclidean and Riemannian spaces. Specifically, when dealing with a nonsmooth Riemannian problem, we leverage the *manifold alternating direction method of multipliers* (MADMM, Kovnatsky et al., 2016) and the *manifold proximal gradient* (ManPG, Chen et al., 2020). They are the Riemannian counterparts of the ADMM (Boyd et al., 2011) and PG (Parikh et al., 2014). Additionally, when dealing with a smooth, constrained, Riemannian problem, our solution combines the *splitting of ortogonality constraints* (SOC, Lai & Osher, 2014), the ADMM, and the *successive convex approximation* (SCA, Nedić et al., 2018) methods.

Related to CA learning are *mechanistic interpretability via causal abstraction* (Bereska & Gavves, 2024) and *causal representation learning* (Schölkopf et al., 2021). Regarding the former, Geiger et al. (2021; 2024) use CA to explain DNNs by treating the DNN as a low-level — to be read as *black-box* — SCM and aligning it via interchange intervention training (IIT) with a high-level — to be read as *human-understandable* — SCM, built from theoretical and empirical modeling work. This setting where two SCMs share structure but differ only in semantics would not constitute valid macro abstraction under our framework, nor can IIT objectives be directly adapted due to (NA1)-(NA3).

Regarding the latter, CRL treats low-level variables (e.g. pixels) as noncausal observations generated by latent high-level causal concepts, and aims to recover those concepts and their causal graph to improve interpretability and performance robustness (Schölkopf et al., 2021; Yang et al., 2021; Komanduri et al., 2022). This goal also aligns with recent developments that integrate causal reasoning into communications, leveraging a functorial formalism as well (Thomas et al., 2023; Thomas & Saad, 2023; 2024). By contrast, CA learning focuses on mappings between SCMs, where the low- and high-level variables are causal and known, to enable causal knowledge transfer across abstraction levels.

**Contributions.** *First*, we introduce the *semantic embedding principle* (SEP) for CA, informally stating that in a well-behaved CA, embedding the high-level (coarser) causal knowledge into the low-level (finer) one and then abstracting it back enables perfect reconstruction of the high-level causal knowledge. *Second*, to formalize SEP categorically, we present an alternative category-theoretic framework for CA, which allows us to focus on the semantic layer of an SCM. *Third*, we formulate a general CA learning problem based on SEP and (A1). *Fourth*, we tackle the linear CA case, showing that SEP naturally links the linear CA to the geometry of the Stiefel manifold, shaping the learning process as a Riemannian optimization problem. As an application, we consider the Gaussian setting with the Kullback-Liebler (KL) divergence as a measure of alignment between the low- and high-level SCMs. *Fifth*, we formalize and solve nonsmooth and smooth learning problems for linear CAs in this setting. For the former, we present the LinSEPAL-ADMM and LinSEPAL-PG methods; for the latter, the CLinSEPAL one. Our experiments on synthetic and brain data, across different levels of prior knowledge, confirm good performance of the proposed methods.

*Our work is a first step to bridging the gap between CA learning methods and real-world applications.*

## 2. Background on Causality and Abstraction

This section provides the notation and key concepts related to causal modeling and abstraction theory.

**Notation.** The set of integers from $1$ to $n$ is $[n]$. The vectors of zeros and ones of size $n$ are $\mathbf{0}_n$ and $\mathbf{1}_n$. The identity matrix of size $n \times n$ is $\mathbf{I}_n$. The Frobenius norm is $\|\mathbf{A}\|_{\mathrm{F}}$. The set of positive definite matrices over $\mathbb{R}^{n \times n}$ is $\mathcal{S}_{++}^n$. The Hadamard product is $\odot$. Function composition is $\circ$. The domain of a function is $\mathbb{D}[\cdot]$ and its kernel $\ker$. Let $\mathcal{M}(\mathcal{X}^n)$ be the set of Borel measures over $\mathcal{X}^n \subseteq \mathbb{R}^n$. Given a measure $\mu^n \in \mathcal{M}(\mathcal{X}^n)$ and a measurable map $\varphi^{\mathbf{V}}, \mathcal{X}^n \ni \mathbf{x} \xmapsto{\varphi^{\mathbf{V}}} \mathbf{V}^\top \mathbf{x} \in \mathcal{X}^m$, we denote by $\varphi_{\#}^{\mathbf{V}}(\mu^n) := \mu^n(\varphi^{\mathbf{V}^{-1}}(\mathbf{x}))$ the pushforward measure $\mu^m \in \mathcal{M}(\mathcal{X}^m)$.

We now present the standard definition of SCM.

**Definition 2.1** (SCM, Pearl, 2009). A (Markovian) structural causal model (SCM) $\mathsf{M}^n$ is a tuple $\langle \mathcal{X}, \mathcal{Z}, \mathcal{F}, \zeta^{\mathcal{Z}} \rangle$, where *(i)* $\mathcal{X} = \{X_1, \ldots, X_n\}$ is a set of $n$ endogenous random variables; *(ii)* $\mathcal{Z} = \{Z_1, \ldots, Z_n\}$ is a set of $n$ exogenous variables; *(iii)* $\mathcal{F}$ is a set of $n$ functional assignments such that $X_i = f_i(\mathcal{P}_i, Z_i), \forall\, i \in [n]$, with $\mathcal{P}_i \subseteq \mathcal{X} \setminus \{X_i\}$; *(iv)* $\zeta^{\mathcal{Z}}$ is a product probability measure over independent exogenous variables $\zeta^{\mathcal{Z}} = \prod_{i \in [n]} \zeta^i$, where $\zeta^i = P(Z_i)$.

A Markovian SCM induces a directed acyclic graph (DAG) $\mathcal{G}_{\mathsf{M}^n}$ where the nodes represent the variables $\mathcal{X}$ and the edges are determined by the structural functions $\mathcal{F}$; $\mathcal{P}_i$ constitutes then the parent set for $X_i$. Furthermore, we can recursively rewrite the set of structural function $\mathcal{F}$ as a set of mixing functions $\mathcal{M}$ dependent only on the exogenous variables (cf. App. C). A key feature for studying causality is the possibility of defining interventions on the model:

**Definition 2.2** (Hard intervention, Pearl, 2009). Given SCM $\mathsf{M}^n = \langle \mathcal{X}, \mathcal{Z}, \mathcal{F}, \zeta^{\mathcal{Z}} \rangle$, a (hard) intervention $\iota = \mathrm{do}(\mathcal{X}^\iota = \mathbf{x}^\iota)$, $\mathcal{X}^\iota \subseteq \mathcal{X}$, is an operator that generates a new post-intervention SCM $\mathsf{M}_\iota^n = \langle \mathcal{X}, \mathcal{Z}, \mathcal{F}_\iota, \zeta^{\mathcal{Z}} \rangle$ by replacing each function $f_i$ for $X_i \in \mathcal{X}^\iota$ with the constant $x_i^\iota \in \mathbf{x}^\iota$. Graphically, an intervention mutilates $\mathcal{G}_{\mathsf{M}^n}$ by removing all the incoming edges of the variables in $\mathcal{X}^\iota$.

Given multiple SCMs describing the same system at different levels of granularity, CA provides the definition of an $\alpha$-abstraction map to relate these SCMs:

**Definition 2.3** ($\boldsymbol{\alpha}$-abstraction, Rischel, 2020). Given low-level $\mathsf{M}^\ell$ and high-level $\mathsf{M}^h$ SCMs, an $\boldsymbol{\alpha}$-abstraction is a triple $\boldsymbol{\alpha} = \langle \mathcal{R}, m, \alpha \rangle$, where *(i)* $\mathcal{R} \subseteq \mathcal{X}^\ell$ is a subset of relevant variables in $\mathsf{M}^\ell$; *(ii)* $m : \mathcal{R} \to \mathcal{X}^h$ is a surjective function between the relevant variables of $\mathsf{M}^\ell$ and the endogenous variables of $\mathsf{M}^h$; *(iii)* $\alpha : \mathbb{D}[\mathcal{R}] \to \mathbb{D}[\mathcal{X}^h]$ is a modular function $\alpha = \bigotimes_{i \in [n]} \alpha_{X_i^h}$ made up by surjective functions $\alpha_{X_i^h} : \mathbb{D}[m^{-1}(X_i^h)] \to \mathbb{D}[X_i^h]$ from the outcome of low-level variables $m^{-1}(X_i^h) \in \mathcal{X}^\ell$ onto outcomes of the high-level variables $X_i^h \in \mathcal{X}^h$.

Notice that an $\boldsymbol{\alpha}$-abstraction simultaneously maps variables via the function $m$ and values through the function $\alpha$. The definition itself does not place any constraint on these functions, although a common requirement in the literature is for the abstraction to satisfy *interventional consistency* (Rubenstein et al., 2017; Rischel, 2020; Beckers & Halpern, 2019). An important class of such well-behaved abstractions is *constructive linear abstraction*, for which the following properties hold. By constructivity, *(i)* $\boldsymbol{\alpha}$ is interventionally consistent; *(ii)* all low-level variables are relevant $\mathcal{R} = \mathcal{X}^\ell$; *(iii)* in addition to the map $\alpha$ between endogenous variables, there exists a map $\alpha_U$ between exogenous variables satisfying interventional consistency (Beckers & Halpern, 2019; Schooltink & Zennaro, 2024). By linearity,

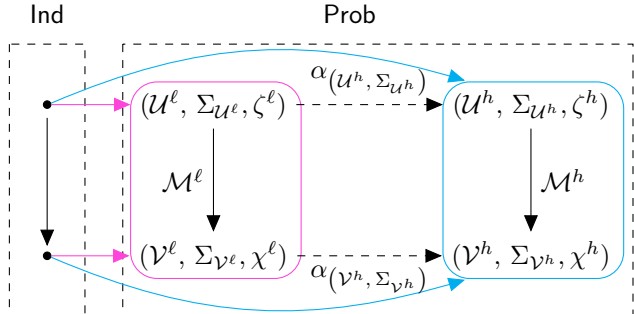

**Figure 2: An abstraction as natural transformation, that is, a set of commuting arrows in** Prob **(dashed black) from** $\mathsf{M}^\ell$ **(purple) to** $\mathsf{M}^h$ **(cyan).**

$\alpha = \mathbf{V}^\top \in \mathbb{R}^{h \times \ell}$ (Massidda et al., 2024). App. C provides formal definitions for interventional consistency, linear and constructive abstraction.

## 3. Category-theory Formalization

Standard category-theoretic formalization of CA (Rischel, 2020; Otsuka & Saigo, 2022) are based on a functorial semantics (Jacobs et al., 2019) approach mapping the graphical structure of causal models (*syntax*) onto the discrete distributions of individual variables (*semantics*). Because of our non-assumption (NA2), no knowledge of the structure of an SCM is available in our setting; thus, we propose a formalization mapping a dyadic structure (*syntax*) onto the exogenous and the endogenous probability measures implied by an SCM (*semantics*).

A crucial role in our modelling is that of the mixing functions $\mathcal{M}$, which express the data generation process as a recursive process from the exogenous functions. This allows us to define an SCM $\mathsf{M}^n$ in measure-theoretic terms as a tuple made up of the probability space of exogenous variables $(\mathcal{U}, \Sigma_{\mathcal{U}}, \zeta)$, the probability space of the endogenous variables $(\mathcal{V}, \Sigma_{\mathcal{V}}, \chi)$, and a set of measurable functions $\mathcal{M}$ given by the mixing functions (cf. App. C).

We can now rely on this representation to interpret an SCM as a category-theoretic functor from a simple index category Ind, made up only of a source and a sink object and an edge between them, to the category of probability spaces Prob, where objects $(X, \Sigma_X, p)$ are probability spaces and morphisms $\varphi$ are measurable maps:

**Definition 3.1** (Category-theoretic SCM). An SCM is a functor $\mathsf{M}^n : \mathsf{Ind} \to \mathsf{Prob}$, mapping the source node of Ind to $(\mathcal{U}, \Sigma_{\mathcal{U}}, \zeta)$, the sink node of Ind to $(\mathcal{V}, \Sigma_{\mathcal{V}}, \chi)$, and the edge of Ind to the collection $\mathcal{M}$ of measurable maps.

App. B presents basic category-theoretic concepts, whereas App. C.5 deepens Def. 3.1. CA can now be expressed as

a natural transformation between two SCMs, as shown in Fig. 2. This formulation has two important features. First, it highlights the role of exogenous variables in a constructive abstraction showing the commutativity of the paths $\mathcal{M}^h \circ \alpha_{(\mathcal{U}^h, \Sigma_{\mathcal{U}^h})}$ and $\alpha_{(\mathcal{V}^h, \Sigma_{\mathcal{V}^h})} \circ \mathcal{M}^\ell$. Second, morphisms in Prob relates measure spaces, viz. sets equipped with sigma algebras. Consequently, the natural transformation components are measurable maps with dimensionality determined by the cardinality of $\mathcal{X}^h$ and $\mathcal{X}^\ell$. To ease the notation, we will denote $\alpha_{(\mathcal{U}^h, \Sigma_{\mathcal{U}^h})}$ by $\alpha_{\mathcal{Z}}$ and $\alpha_{(\mathcal{V}^h, \Sigma_{\mathcal{V}^h})}$ by $\alpha_{\mathcal{X}}$. Then, we can formally recast the $\alpha$-abstraction in Prob.

**Definition 3.2** ($\alpha$-abstraction in Prob). Given low-level $\mathsf{M}^\ell$ and high-level $\mathsf{M}^h$ SCMs, an abstraction $\alpha = \langle \mathcal{R}, \mathcal{Q}, m, \alpha \rangle$ is a tuple, where: *(i)* $\mathcal{R}$ is the same as in Def. 2.3; *(ii)* $\mathcal{Q} \subseteq \mathcal{Z}^\ell$ is a set of relevant exogenous variables given by the union of the set of exogenous corresponding to the endogenous in $\mathcal{R}$ and those corresponding to their ancestors; *(iii)* $m = \langle m_{\mathcal{Z}}, m_{\mathcal{X}} \rangle$ is a pair of surjective functions mapping sets, $m_{\mathcal{Z}} : \mathcal{Q} \to \mathcal{Z}^h$ and $m_{\mathcal{X}} : \mathcal{R} \to \mathcal{X}^h$, respectively; *(iv)* $\alpha = \langle \alpha_{\mathcal{Z}}, \alpha_{\mathcal{X}} \rangle$ is a natural transformation made by measurable functions mapping probability spaces, $\alpha_{\mathcal{Z}}$ for the exogenous and $\alpha_{\mathcal{X}}$ for the endogenous, respectively.

As Def. 2.3, Def. 3.2 makes no reference to interventional consistency. App. D explains how intervened SCMs and interventional consistency can be represented categorically.

## 4. Problem Formulation

Within our category-theoretic framework, CA learning amounts to finding the endogenous components $m_{\mathcal{X}}$ and $\alpha_{\mathcal{X}}$ from data. We start by formulating a *general* learning problem working under the non-assumption (NA1)-(NA5), and then decline it to the case of linear CA.

Our problem formulation relies upon three key ingredients. *First*, we assume that the data generated by a constructive abstraction adheres to the *semantic embedding principle*. This principle requires that the CA component $\alpha_{\mathcal{X}}$ admits a right-inverse measurable map.

**Definition 4.1** (Semantic embedding principle, SEP). Given an $\alpha$-abstraction as in Def. 3.2, the semantic embedding principle states that $\alpha_{\mathcal{X}}$ has a right-inverse measurable map $\beta_{\mathcal{X}}$, such that $\alpha_{\mathcal{X}} \circ \beta_{\mathcal{X}} = \mathrm{Id}_{(\mathcal{V}^h, \Sigma_{\mathcal{V}^h}, \chi^h)}$. Hence, it holds

$$\chi^h = \varphi_{\#}^{\alpha_{\mathcal{X}} \circ \beta_{\mathcal{X}}}(\chi^h). \tag{1}$$

The SEP implies that going from the high-level model $\mathsf{M}^h$ to the low-level model $\mathsf{M}^\ell$ and then abstracting back to $\mathsf{M}^h$ allows for perfect reconstruction. Notice that SEP only holds in one direction, as suggested by the word embedding; thus, identity on the left inverse is not guaranteed, meaning that the abstraction from the low level to the high level can still shed information, as we would expect in CA.

*Second*, because of the non-assumption (NA3) only observational data is available. Thus, we can not explicitly use interventional consistency information to drive our learning. Only if we identify the true constructive abstraction, we are guaranteed interventional consistency. In trying to learn the abstraction, we leverage (A1), which is met in application domains as discussed in Sec. 1.

*Third*, to learn a CA, we look for a distance function quantifying the misalignment between the probability measures $\chi^\ell$ and $\chi^h$, given $\alpha_{\mathcal{X}}$. Since the probability measures belong to spaces of different dimensionality, specifically $\mathbb{R}^\ell$ and $\mathbb{R}^h$, we leverage the approach proposed in (Cai & Lim, 2022) to compute the misalignment through an embedding as $D\left(\chi^h, \varphi_{\#}^{\alpha_{\mathcal{X}}}(\chi^\ell)\right)$, where $D$ is an information-theoretic metric (e.g., p-Wasserstein) or $\phi$-divergence (e.g., Kullback-Leibler). Please refer to App. F for more details. We can now pose the following general learning problem:

> **Problem 1.** *(SEP-based CA Learning)*
> **Input**: *(i) probability measures $\chi^\ell$ and $\chi^h$; (ii) prior information about $m_{\mathcal{X}}$, and (iii) a distance function $D\left(\chi^h, \varphi_{\#}^{\alpha_{\mathcal{X}}}(\chi^\ell)\right)$.*
> **Goal**: *learn a measurable map $\alpha_{\mathcal{X}}^\star$ such that (i) it belongs to $\ker D\left(\chi^h, \varphi_{\#}^{\alpha_{\mathcal{X}}}(\chi^\ell)\right)$, (ii) it complies with SEP in Def. 4.1, and (iii) it agrees with the prior information about $m_{\mathcal{X}}$.*

The zeroing of the distance function implies $\chi^h = \varphi_{\#}^{\alpha_{\mathcal{X}}^\star}(\chi^\ell)$, which, together with Eq. (1), yields $\varphi_{\#}^{\alpha_{\mathcal{X}}^\star \circ \beta_{\mathcal{X}}}(\chi^h) = \varphi_{\#}^{\alpha_{\mathcal{X}}^\star}(\chi^\ell)$. However, despite solving Prob. 1, there is no guarantee that $\alpha_{\mathcal{X}}^\star$ coincides with the ground truth CA. In other words, the optimal solution is not unique. For a linear constructive CA, we express $m_{\mathcal{X}}$ and $\alpha_{\mathcal{X}}$ as $\mathbf{B}^\top \in \{0, 1\}^{h \times \ell}$ and $\mathbf{V}^\top \in \mathbb{R}^{h \times \ell}$, respectively. In accordance with constructivity, each row of $\mathbf{B}$ has a single nonzero entry, and each column has at least one nonzero entry. Importantly, for linear CA, a simple yet principled way to satisfy SEP is via the geometry of the Stiefel manifold:

$$\mathrm{St}(\ell, h) := \{\mathbf{V} \in \mathbb{R}^{\ell \times h} \mid \mathbf{V}^\top \mathbf{V} = \mathbf{I}_h\}. \tag{2}$$

The Stiefel manifold (see App. E for details), is a convenient choice for the following reasons: *(i)* differently from a generic pseudo-inverse matrix, the orthogonality of $\mathbf{V}$ guarantees that the geometry of the high-level space is preserved; *(ii)* the transpose eases the formulation and ensures numerical stability in optimization. Consequently, we restate SEP for the linear case as follows.

**Definition 4.2** (Semantic embedding principle, linear case). Given the linear constructive CA, viz. $\mathbf{V}^\top$, SEP implies that $\mathbf{V} \in \mathrm{St}(\ell, h)$. From Eq. (1) we get $\chi^h = \varphi_{\#}^{\mathbf{V} \circ \mathbf{V}^\top}(\chi^h)$.

A pictorial representation of Def. 4.2 is provided in Fig. 1. Def. 4.2 shapes our methodology for CA learning, posing it as a Riemannian optimization problem (Boumal, 2023). As an application, in the sequel, we will tackle an implementation of Prob. 1 for the linear constructive case $\alpha_{\mathcal{X}} = \mathbf{V}^\top$, where *(i)* $\chi^h \sim N(\mathbf{0}_h, \mathbf{\Sigma}^h)$ and $\chi^\ell \sim N(\mathbf{0}_\ell, \mathbf{\Sigma}^\ell)$; and *(ii)* $D\left(\chi^h, \varphi_\#^{\mathbf{V}}(\chi^\ell)\right) = D^{\mathrm{KL}}\left(\chi^h || \varphi_\#^{\mathbf{V}}(\chi^\ell)\right)$ where $D^{\mathrm{KL}}$ stands for KL divergence. Specifically,

$$D_{\mathbf{V}}^{\mathrm{KL}} = \mathrm{Tr}\left\{\left(\mathbf{V}^\top \mathbf{\Sigma}^\ell \mathbf{V}\right)^{-1} \mathbf{\Sigma}^h\right\} + \log \det\left\{\mathbf{V}^\top \mathbf{\Sigma}^\ell \mathbf{V}\right\} + C \,, \quad (3)$$

where $C = -\log \det\{\mathbf{\Sigma}^h\} - h$ is a constant term. Additionally, from Eq. (3) it is immediate to see that both $\mathbf{V}$ and $-\mathbf{V}$ belong to $\ker D_{\mathbf{V}}^{\mathrm{KL}}$. Such an application is highly relevant as it is common to deal in practice with Gaussian measures (or quasi) (D'Acunto et al., 2024); also, in causality, such a measure easily arises from the prominent family of linear models (Bollen, 1989; Shimizu et al., 2006) and is investigated in the CA literature (Kekić et al., 2024; Massidda et al., 2024). KL divergence is a common choice in ML and statistics, but notice that any distance vanishes when evaluated at the ground truth.

**Spectral properties entailed by SEP.** In the setting of zero-mean Gaussian measures, all causal information is encoded in the covariance matrices of $\chi^\ell$ and $\chi^h$, which may be geometrically represented as ellipsoids in $\mathbb{R}^\ell$ and $\mathbb{R}^h$. Each covariance $\mathbf{\Sigma}$ admits an eigendecomposition $\mathbf{\Sigma} = \mathbf{U}\mathbf{\Lambda}\mathbf{U}^\top$, where the columns of $\mathbf{U}$ specify the ellipsoid's principal axes and the square roots of the diagonal entries of $\mathbf{\Lambda}$ give the corresponding axis lengths. Thus, for the low-level distribution $N(\mathbf{0}_\ell, \mathbf{\Sigma}^\ell)$, one obtains an $\ell$-dimensional ellipsoid with axes $\mathbf{U}^\ell$ and lengths $\sqrt{\lambda_i}$, $i \in [\ell]$; similarly, $N(\mathbf{0}_h, \mathbf{\Sigma}^h)$ defines an $h$-dimensional ellipsoid with axes $\mathbf{U}^h$ and lengths $\sqrt{\kappa_j}$, $j \in [h]$.

When projecting the low-level ellipsoid into an $h$-dimensional subspace via an orthonormal map $\mathbf{V} \in \mathrm{St}(\ell, h)$, the resulting covariance $\mathbf{V}^\top \mathbf{\Sigma}^\ell \mathbf{V} = \mathbf{V}^\top \mathbf{U}^\ell \mathbf{\Lambda} \mathbf{U}^{\ell\top} \mathbf{V} = \mathbf{Q}^\top \mathbf{\Lambda} \mathbf{Q}$, with $\mathbf{Q} = \mathbf{U}^{\ell\top} \mathbf{V}$, yields a projected ellipsoid whose axes are linear combinations of those in $\mathbf{U}^\ell$. Because $\mathbf{V}$ is contractive by SEP, no projection can increase variance. Thus, the length of each axis of the projected ellipsoid lies between the minimum and maximum length of those of the $\ell$-dimensional ellipsoid. Specifically, by the Ostrowski's theorem for rectangular $\mathbf{V}$ (cf. Th. 3.2 in Higham & Cheng, 1998), the $i$-th largest axis length of the projected ellipsoid falls between the $i$-th and $(i + \ell - h)$-th largest lengths of the $\ell$-dimensional ellipsoid. Consequently, for the optimal CA case where the projected ellipsoid exactly matches that of $\mathbf{\Sigma}^h$, the projected axis align with $\mathbf{U}^h$ and the previous interlacing inequalities provide necessary spectral conditions for the existence of a linear CA from $N(\mathbf{0}_\ell, \mathbf{\Sigma}^\ell)$ to $N(\mathbf{0}_h, \mathbf{\Sigma}^h)$.

**Theorem 4.3.** *Let $\chi^\ell \sim N(\mathbf{0}_\ell, \mathbf{\Sigma}^\ell)$, $\chi^h \sim N(\mathbf{0}_h, \mathbf{\Sigma}^h)$, where $\mathbf{\Sigma}^\ell \in \mathcal{S}_{++}^\ell$ and $\mathbf{\Sigma}^h \in \mathcal{S}_{++}^h$. Denote by $0 < \lambda_1 \leq \ldots \leq \lambda_\ell$ the eigenvalues of $\mathbf{\Sigma}^\ell$, and by $0 < \kappa_1 \leq \ldots \leq \kappa_h$ those of $\mathbf{\Sigma}^h$. If a linear CA $\mathbf{V} \in \mathrm{St}(\ell, h)$ complying with SEP from $\chi^\ell$ to $\chi^h$ exists, then*

$$\lambda_i \leq \kappa_i \leq \lambda_{i+\ell-h}, \quad \forall i \in [h] \,. \quad (4)$$

*Proof.* See App. G. $\qquad\square$

We now turn to formulating the learning problem. We investigate two approaches for injecting the prior information about $m_{\mathcal{X}}$, encoded in the matrix of *prior knowledge* $\mathbf{B}$, into our problem. Please notice that in case $\mathbf{B}$ is not fully specified, it might not comply with the row and column constraints discussed above. These formulations translate into non-smooth and smooth Riemannian learning problems.

**Nonsmooth problem.** In the nonsmooth problem we introduce $\mathbf{B}$ as a penalty term in the objective function. The rationale is to penalize entries in $\mathbf{V}$ corresponding to zeros in $\mathbf{B}$. Let $\mathbf{D} = (\mathbf{1}_{\ell \times h} - \mathbf{B})$. The problem reads as follows:

**Problem 2.** *Given $\mathbf{\Sigma}^\ell \in \mathcal{S}_{++}^\ell$, $\mathbf{\Sigma}^h \in \mathcal{S}_{++}^h$, $\mathbf{D} \in \{0,1\}^{\ell \times h}$, and $\lambda \in \mathbb{R}_+$, the CA is the transpose of*

$$\mathbf{V}^\star = \underset{\mathbf{V} \in \mathrm{St}(\ell, h)}{\arg\min} \; f(\mathbf{V}) + \lambda \underbrace{\|\mathbf{D} \odot \mathbf{V}\|_1}_{h(\mathbf{V})} \,. \quad (5)$$

*Here, $f(\mathbf{V})$ follows Eq. (3), omitting the constant $C$.*

Please notice that, although appealing in its form, Eq. (5) does not guarantee the constructiveness of the learned CA. Moreover, the penalty term introduces a bias in the learned $\mathbf{V}$ in the case of partial prior knowledge.

**Smooth problem.** In the smooth problem, we introduce $\mathbf{B}$ directly in the objective function $f(\cdot)$. The CA is now defined as the Hadamard product of $\mathbf{V}$ and the support $(\mathbf{B} \odot \mathbf{S})$ integrating prior $\mathbf{B}$ and learned $\mathbf{S}$ knowledge. This formulation is particularly convenient as it enables us to jointly optimize for $\mathbf{V} \in \mathbb{R}^{\ell \times h}$ and matrix $\mathbf{S} \in [0,1]^{\ell \times h}$. However, we also need to introduce three constraints:
*(i)* by SEP, $\mathbf{B} \odot \mathbf{S} \odot \mathbf{V}$ must belong to the Stiefel manifold;
*(ii)* by functionality, the columns of the support $(\mathbf{B} \odot \mathbf{S})^\top$ must sum up to one, meaning that they lie on a sphere, defined as

$$\mathrm{Sp}^\Delta(h, \ell) := \Big\{\mathbf{A} \in \{0,1\}^{h \times \ell} \mid \|\mathbf{a}_j\|_2 = 1 \text{ and}$$
$$\sum_{i=1}^h a_{ij} = 1, \forall j \in [\ell]\Big\}; \quad (6)$$

*(iii)* by surjectivity, the rows of the support $(\mathbf{B} \odot \mathbf{S})^\top$ must contain at least a one. The problem reads as:

**Problem 3.** *Given* $\mathbf{\Sigma}^{\ell} \in \mathcal{S}^{\ell}_{++}$, $\mathbf{\Sigma}^{h} \in \mathcal{S}^{h}_{++}$, *and* $\mathbf{B} \in \{0,1\}^{\ell \times h}$, *the linear constructive CA is given by the transpose of the product* $\mathbf{B} \odot \mathbf{S} \odot \mathbf{V}$, *where*

$$
\mathbf{V}^{\star}, \mathbf{S}^{\star} = \underset{\substack{\mathbf{V} \in \mathbb{R}^{\ell \times h} \\ \mathbf{S} \in [0,1]^{\ell \times h}}}{\arg\min} \quad f(\mathbf{V}, \mathbf{S});
$$

$$
\textit{subject to } (i) \ \mathbf{B} \odot \mathbf{S} \odot \mathbf{V} \in \text{St}(\ell, h), \qquad (7)
$$
$$
(ii) \ (\mathbf{B} \odot \mathbf{S})^{\top} \in \text{Sp}^{\Delta}(h, \ell),
$$
$$
(iii) \ \mathbf{1}_h - (\mathbf{B} \odot \mathbf{S})^{\top} \mathbf{1}_{\ell} \le \mathbf{0}_h;
$$

*and*

$$
f(\mathbf{V}, \mathbf{S}) := \text{Tr}\left\{ \left( (\mathbf{B} \odot \mathbf{S} \odot \mathbf{V})^{\top} \mathbf{\Sigma}^{\ell} (\mathbf{B} \odot \mathbf{S} \odot \mathbf{V}) \right)^{-1} \mathbf{\Sigma}^{h} \right\}
$$
$$
+ \log \det \left\{ (\mathbf{B} \odot \mathbf{S} \odot \mathbf{V})^{\top} \mathbf{\Sigma}^{\ell} (\mathbf{B} \odot \mathbf{S} \odot \mathbf{V}) \right\}.
$$
$$(8)$$

Constraints *(ii)* and *(iii)* further underscore the role of $\mathbf{S}$: it enables learning the support of CA while guaranteeing its constructiveness. Notice that the matrix $\mathbf{S}$ does not need to be a logical matrix; it is the product $\mathbf{B} \odot \mathbf{S}$ which must be logical. Also, if $\mathbf{B}$ provides full prior knowledge about the structure, we have $\mathbf{S} \equiv \mathbf{B}$ and we do not need to learn $\mathbf{S}$. This approach guarantees the ground-truth structure for the learned CA. The full prior problem formulation is provided in App. J.6.

Unfortunately, both the Stiefel manifold in Eq. (2) and $D^{\text{KL}}_{\mathbf{V}}$ in Eq. (3) are nonconvex in $\mathbf{V}$. In the next section we devise methods suitable for this setting.

*Remark* 1. Learning linear CAs remains challenging even with full prior structural knowledge and jointly sampled data (NA5). Although one could decompose the task into $h$ independent linear regressions, enforcing SEP requires each coefficient vector to satisfy a unitary $\ell_2$-norm constraint, rendering the problem nonconvex.

## 5. Problem Solution

To solve the nonsmooth and smooth Riemannian problems in Sec. 4, we leverage the following:

**Proposition 5.1.** *Consider the function*

$$
f(\mathbf{A}) = \text{Tr}\left\{ (\mathbf{A}^{\top} \mathbf{\Sigma}^{\ell} \mathbf{A})^{-1} \mathbf{\Sigma}^{h} \right\} + \log \det \left\{ \mathbf{A}^{\top} \mathbf{\Sigma}^{\ell} \mathbf{A} \right\}. \quad (9)
$$

*Eq. (9) is smooth for* $\mathbf{A} \in \text{St}(\ell, h)$. *Additionally, define* $\widetilde{\mathbf{A}} := (\mathbf{A}^{\top} \mathbf{\Sigma}^{\ell} \mathbf{A})^{-1}$. *The gradient of* $f(\mathbf{A})$ *is*

$$
\nabla_{\mathbf{A}} f = 2 \left( \mathbf{\Sigma}^{\ell} \mathbf{A} \widetilde{\mathbf{A}} \right) \left( \mathbf{I}_h - \mathbf{\Sigma}^{h} \widetilde{\mathbf{A}} \right), \qquad (10)
$$

*Proof.* See App. G. $\qquad\qquad\square$

### 5.1. Solution of the nonsmooth learning problem

Leveraging Proposition 5.1, we have that Eq. (5) is constituted by a smooth yet nonconvex term, $f(\mathbf{V})$, and a nonsmooth one, $h(\mathbf{V})$. Hence we solve Prob. 2 through two different optimization paradigms for nonsmooth Riemannian optimization: MADMM and ManPG. We term the proposed methods *LinSEPAL-ADMM* and *LinSEPAL-PG*, where *LinSEPAL* stands for **Lin**ear **S**emantic **E**mbedding **P**rinciple **A**bstraction **L**earner. Next we provide a sketch of the solution and provide the full mathematical derivation in App. H and App. I.

**LinSEPAL-ADMM.** The MADMM framework appeals to our setting given the objective function separating into smooth and nonsmooth terms. To derive the LinSEPAL-ADMM iterative algorithm, we proceed as follows. First, the nonsmooth term $h(\mathbf{V})$ is associated with a splitting variable $\mathbf{Y}$ to be optimized over $\mathbb{R}^{\ell \times h}$, obtaining an equivalent problem formulation (cf. Eq. (P2)). LinSEPAL-ADMM proceeds by iteratively minimizing the augmented Lagrangian with respect to the primal variables $\mathbf{V}$ and $\mathbf{Y}$, while maximizing w.r.t. the scaled dual variable. Specifically, LinSEPAL-ADMM solves the subproblem for $\mathbf{V}$ (cf. Eq. (36)) through standard techniques for smooth optimization on the Stiefel manifold (e.g., conjugate gradient, Edelman et al., 1998). This is the most complex update in the LinSEPAL-ADMM iterative procedure due to the nonconvex objective and the Stiefel manifold. Next, LinSEPAL-ADMM updates $\mathbf{Y}$ in closed form through the element-wise soft-thresholding operator (cf. Eq. (37)). Finally, the scaled dual variable is updated by adding the primal residual evaluated at the current solution (cf. Eq. (R1)). The stopping criteria for LinSEPAL-ADMM are established according to primal and dual feasibility optimality conditions (Boyd et al., 2011, cf. App. H). To the best of our knowledge, the convergence guarantee for MADMM in the Riemannian space has not been proven. Consequently, the same holds for LinSEPAL-ADMM. Algorithm 1 summarizes the method.

**LinSEPAL-PG.** Our LinSEPAL-PG is an iterative algorithm alternating two updates (cf. Eq. (R2)). The first update is the proximal mapping providing a proximal gradient direction $\mathbf{G}^k$ onto the tangent space to the Stiefel manifold, using the first-order approximation of the objective around the $k$-th estimate. The second is the update for $\mathbf{V}^{k+1}$, which exploits the canonical retraction (cf. Eq. (23)) technique for projecting back $\mathbf{V}^k + \mathbf{G}^k$ from the tangent space to the manifold. The hardest step in the LinSEPAL-PG algorithm is the proximal update (cf. Eq. (42)). We solve it by declining the regularized semi-smooth Newton method (Xiao et al., 2018) to our application (cf. App. I). Following the rationale in (Si et al., 2024), differently from the

original ManPG method which uses the parameterization of the tangent space, we constrain $\mathbf{G}^k$ to the tangent space by exploiting the basis of the normal space to the manifold (cf. Eq. (44)). This way, we ease the mathematical solution, with benefits from the computational perspective (cf. Si et al., 2024). Next, LinSEPAL-PG updates $\mathbf{V}^{k+1}$ in closed form (cf. Eq. (69)) by applying the QR-retraction, employing an Armijo line-search procedure to determine the stepsize. The optimization stops either when a maximum number of iterations is reached, or when $D_{\mathbf{V}^{k+1}}^{\mathrm{KL}}$ is below a threshold $\tau^{\mathrm{KL}} \approx 0$. LinSEPAL-PG inherits the global convergence of the ManPG framework, established in (Chen et al., 2020). Algorithm 2 summarizes the method.

### 5.2. Solution of the smooth learning problem

We provide a sketch of the solution below and the full mathematical derivation in App. J. In this case, we want to jointly optimize $\mathbf{S}$ and $\mathbf{V}$, both being components of the linear CA, viz. $(\mathbf{B} \odot \mathbf{S} \odot \mathbf{V})^{\top}$. Hence, unlike the nonsmooth case, we constrain to the Stiefel manifold the product $(\mathbf{B} \odot \mathbf{S} \odot \mathbf{V})$. To solve Prob. 3, we combine the SOC, ADMM, and SCA methods. According to the rationale behind SOC, we add two splitting variables, namely $\mathbf{Y}_1$ and $\mathbf{Y}_2$ in $\mathrm{St}(\ell, h)$ (cf. Eq. (72)), to separate the nonconvexity of the objective function from that induced by the manifold. The reason why we have two splitting variables is that we need to take into account the bilinear form of the first constraint in Eq. (7). Additionally, to manage the second constraint in Eq. (7), we introduce another splitting variable $\mathbf{X} \in \mathrm{Sp}^{\Delta}(h, \ell)$. Starting from the equivalent problem formulation (cf. Eq. (73)), we write the (nonconvex) scaled augmented Lagrangian (cf. Eq. (74)), thus arriving at the update recursion for our proposed method (cf. Eq. (75)). We term the latter *CLin-SEPAL* (Constructive LinSEPAL) to highlight that it returns constructive support for CA. CLinSEPAL proceeds by iteratively minimizing the augmented Lagrangian w.r.t. the primal variables $\mathbf{V}$, $\mathbf{S}$, $\mathbf{Y}_1$, $\mathbf{Y}_2$, and $\mathbf{X}$; and maximizing it w.r.t. the scaled dual ones. In the subproblems for $\mathbf{V}$ (cf. Eq. (76)) and $\mathbf{S}$ (cf. Eq. (84)), we adopt the SCA paradigm to manage the nonconvexity of $f(\mathbf{V}, \mathbf{S}^k)$ and $f(\mathbf{V}^{k+1}, \mathbf{S})$, respectively. By exploiting the smoothness of $f(\mathbf{V}, \mathbf{S})$ (cf. Corollary J.1), the strongly convex surrogates are derived around the current solution (cf. Eqs. (77) and (85)). CLin-SEPAL solves the strongly convex surrogate subproblems (cf. Eqs. (78) and (88)) exactly. Due to the presence of the inequality constraints, the subproblem for $\mathbf{S}$ is a constrained quadratic programming problem. CLinSEPAL solves it via standard techniques (e.g., Stellato et al., 2020). These two steps in CLinSEPAL can be seen as an instance of the linearized ADMM framework (Alg.1 in Lu et al., 2021) where each internal update is solved exactly. Next, CLinSEPAL solves in closed-form the updates for the three splitting variables. Indeed, the subproblems for $\mathbf{Y}_1$ and $\mathbf{Y}_2$ amount

to the *closest orthogonal approximation problem* (Fan & Hoffman, 1955; Higham, 1986), whose solution is obtained in closed form via polar decomposition. Subsequently, the subproblem for $\mathbf{X}$ is solved in closed-form according to Lemma J.3. Finally, the scaled dual variables are updated with the corresponding primal residuals. Empirical convergence for CLinSEPAL is established when the norms of primal (cf. Eq. (97)) and dual (cf. Eq. (98)) residuals vanish, in accordance with absolute and relative tolerances (cf. Eq. (99)). Algorithm 3 summarizes the method. Additionally, App. J.6 details the solution in the special case of full prior knowledge.

## 6. Empirical Assessment on Synthetic Data

This section provides the empirical assessment of LinSEPAL-ADMM, LinSEPAL-PG and CLinSEPAL with different degrees of prior knowledge, from full (*fp*) to partial (*pp*). We monitor four metrics to evaluate the learned CA $\widehat{\mathbf{V}}^{\top}$: *(i) constructiveness*, as required by Def. 4.2; *(ii)* $D_{\widehat{\mathbf{V}}}^{\mathrm{KL}}$ evaluating the alignment between $\varphi_{\#}^{\widehat{\mathbf{V}}}(\chi^{\ell})$ and $\chi^h$; *(iii)* the Frobenius distance between the absolute value of $\widehat{\mathbf{V}}$ and that of the ground truth $\mathbf{V}^{\star}$, normalized by $\|\mathbf{V}^{\star}\|_{\mathrm{F}}$ to make the settings comparable; *(iv)* the F1 score computed using the support of the learned CAs and that of $\mathbf{V}^{\star}$ to evaluate structural interventional consistency. App. K provides the definition for the above metrics and the hyper-parameters values used in the experiments.

**Full prior knowledge.** In the fp case, we investigate three different settings $(\ell, h) \in \{(12, 2), (12, 4), (12, 6)\}$, corresponding to the cases of *high*, *medium-high*, and *medium* coarse-graining. We do not consider the case where $h > \ell/2$ since the abstraction for $h - \ell/2$ nodes of the low-level model would be fully specified due to the availability of full prior knowledge. For each setting, we instantiate $S = 30$ ground truth abstractions $\mathbf{V}^{\star}$, and for each simulation $s \in [S]$ we run all the methods $R = 50$ times, with different initializations. Then, for each $s$ and method, we retain the $\widehat{\mathbf{V}}$ minimizing the objective $D^{\mathrm{KL}}$.
Fig. 3 shows the performance of the tested methods. All the methods provide constructive CAs $\forall s \in [S]$, and reach a good level of alignment in terms of $D_{\widehat{\mathbf{V}}}^{\mathrm{KL}}$. Recall that, while CLinSEPAL and LinSEPAL-ADMM stop the learning procedure according to primal and dual residuals convergence, LinSEPAL-PG exits when $D_{\widehat{\mathbf{V}}}^{\mathrm{KL}}$ is below a certain threshold $\tau^{\mathrm{KL}}$ (in the experiments $\tau^{\mathrm{KL}} = 10^{-4}$). The Frobenius absolute distance shows comparable performances for the three methods, although CLinSEPAL and LinSEPAL-ADMM outperform in case $(\ell, h) = (12, 4)$. This metric tells us that, as $h$ increases, the learned $\widehat{\mathbf{V}}$ tends (in absolute terms) to the ground truth. Interestingly, when $(\ell, h) = (12, 2)$ we observe a high distance from $\mathbf{V}^{\star}$, although the learned $\widehat{\mathbf{V}}$

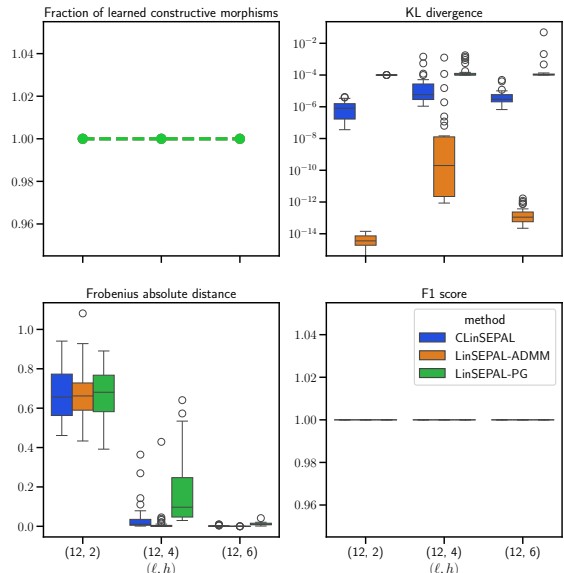

**Figure 3: Synthetic fp results for all settings $(\ell, h)$ and methods:** *(i)* **fraction of learned CAs that are constructive,** *(ii)* $D^{\mathrm{KL}}_{\widehat{\mathbf{V}}}$**,** *(iii)* **normalized absolute Frobenius distance from** $\mathbf{V}^\star$**, and** *(iv)* F1 **score.**

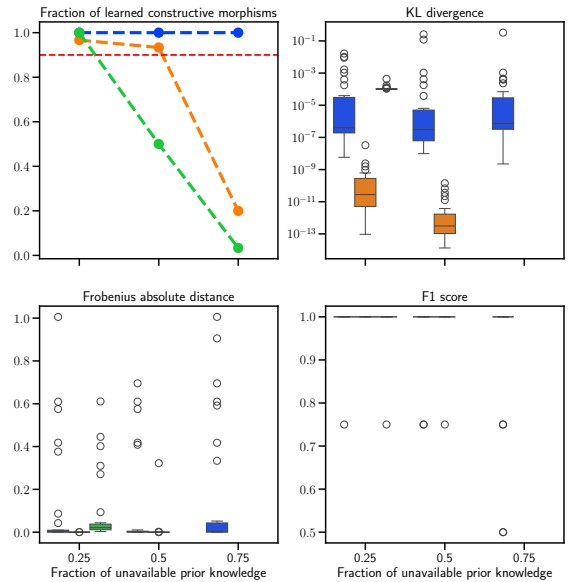

**Figure 4: Synthetic pp results for setting $(\ell, h) = (4, 2)$, all methods, and prior knowledge amounting to the correct structural mapping for** $25\%$**,** $50\%$**, or** $75\%$ **of the nodes. All plots and color legend as in Fig. 3.**

has the correct structure (cf. F1 score). This suggests that under a high coarse-graining, the size of $\ker D^{\mathrm{KL}}$ grows, and it is more difficult for our methods to estimate $\mathbf{V}^\star$ under (NA1)-(NA5). Finally, the F1 score confirms that the methods guarantee the true CA structure of $\widehat{\mathbf{V}}$, for all the settings. To sum up, CLinSEPAL and LinSEPAL-ADMM are slightly better choices than LinSEPAL-PG in case of full prior knowledge in our experimental setting.

**Partial prior knowledge.** In the pp case, we consider the setting $(\ell, h) \in \{(4, 2)\}$ and simulate partial prior knowledge by setting to 1 the entries of the $25\%$, $50\%$, and $75\%$ of the rows in $\mathbf{B}$. For each setting, we instantiate $S = 30$ ground truth abstractions $\mathbf{V}^\star$, and for each simulation $s \in [S]$ we run all the methods $R = 30$ times, with different initializations.

In Fig. 4, the first plot immediately shows that only CLinSEPAL consistently returns a constructive linear CA, as guaranteed by its formulation in Prob. 3. We decided to consider methods performing under a threshold of $90\%$ to be unreliable in returning constructive CAs and not to report their remaining metrics. For completeness, Fig. 7 in App. L provides the results where the threshold on constructiveness is removed. In the case of a limited drop of prior knowledge ($25\%$) all methods perform well, similarly to the fp case, with CLinSEPAL and LinSEPAL-ADMM slightly outperforming LinSEPAL-PG. With a higher drop ($50\%$), LinSEPAL-PG fails to achieve our constructiveness threshold, while CLinSEPAL and LinSEPAL-ADMM still perform well, although LinSEPAL-ADMM provides a lower

fraction of constructive CAs. Finally, with the highest drop ($75\%$) CLinSEPAL succeeds in learning a constructive CA and lowering $D^{\mathrm{KL}}$, even if the Frobenius absolute distance slightly increases. To sum up, for the pp setting only CLinSEPAL guarantees a constructive abstraction.

## 7. Causal Abstraction of Brain Networks

To show the practical relevance of our approach, we apply CLinSEPAL to resting-state functional magnetic resonance imaging (rs-fMRI) data, using the dataset from (D'Acunto et al., 2024) (refer to the paper for details on the dataset). The data, publicly released as part of the *Human Connectome Project* (Smith et al., 2013), comprises recordings from 100 healthy adults with a parcellation scheme that divides the brain into 89 regions of interest (ROIs), $K = 44$ for each hemisphere plus the shared vermis region.

We simulate a first investigating team of neuroscientists taking zero-mean stationary time series for the left hemisphere of the first adult in the dataset. They estimate the data covariance matrix using a Gaussian mixture probability model, viz. $\mathbf{\Sigma}^\ell \in \mathbb{R}^{\ell \times \ell}$, with $\ell = K + 1$, and interpret it as generated by an underlying, unknown, low-level SCM.

In a first fp scenario, we imagine a second investigating team that has collected data according to their causal network specified on a coarser parcellation of the same brain in $h = 14$ macro ROIs. We generate the data for the second team using a ground truth linear CA $\mathbf{B}, \mathbf{V}^\star \in \mathrm{St}(45, 14)$

based on the structural mapping in (D'Acunto et al., 2024), and use the data for estimating the covariance matrix $\mathbf{\Sigma}^h \in \mathbb{R}^{h \times h}$. In this scenario it is realistic to assume knowledge of $\mathbf{B}$ defining how macro ROIa are mapped to ROIs. Then, to align their models, the two groups run CLinSEPAL to recover the abstraction given $\mathbf{\Sigma}^\ell, \mathbf{\Sigma}^h$ and $\mathbf{B}$. Fig. 8 (in App. M) shows that CLinSEPAL recovers $\mathbf{V}^\star$.

In a second pp scenario, we imagine that the second investigating team has collected data according to a causal network aggregating ROI time series into $h = 8$ brain functional networks related to different activities (e.g., motor, visual, default mode). Data is generated again through a ground truth linear CA $\mathbf{B}, \mathbf{V}^\star \in \mathrm{St}(45, 8)$ based on groupings in (D'Acunto et al., 2024) and the covariance matrix $\mathbf{\Sigma}^h \in \mathbb{R}^{h \times h}$ computed. In this scenario, knowledge of $\mathbf{B}$ is debatable as different studies in the literature suggest different relations between ROIs and functions; we then express this partial information via uncertainty over $\mathbf{B}$, meaning that some rows of $\mathbf{B}$ have more than one entry equal to one. Fig. 9 in App. M shows the $\mathbf{B}$ matrix provided as input to CLinSEPAL, as well as the ground truth. The two groups now run CLinSEPAL using $\mathbf{\Sigma}^\ell, \mathbf{\Sigma}^h$ and an uncertain $\mathbf{B}$; partial knowledge compounds on an already challenging learning problem due to the high coarse-graining. Fig. 10 and Fig. 11 show results with different levels of uncertainty. For low uncertainty, CLinSEPAL correctly retrieves the structure of the CA, although we observe some variation in the colors w.r.t. $\mathbf{V}^\star$; additionally, $D_{\widehat{\mathbf{V}}}^{\mathrm{KL}}$ and the Frobenius absolute distance in Fig. 11 show that misalignment is minimized and $\widehat{\mathbf{V}}$ very close to $\mathbf{V}^\star$. For medium and high uncertainty, CLinSEPAL makes some mistakes in terms of structural mapping, but Fig. 11 shows that insights from the method are still valuable.

## 8. Conclusion and Future Works

In this work, we addressed the challenge of CA learning in realistic scenarios, abandoning restrictive assumptions (NA1)-(NA5) that limit the applicability of existing methods. We proposed an alternative category-theoretic framework for SCM and CA, and introduced the *semantic embedding principle* to learn CAs that meaningfully preserve information. We formulated a general CA learning problem grounded in SEP, under a mild assumption of partial prior knowledge about the structure of CA. For the linear CA setting, we showed how SEP links CA to the geometry of the Stiefel manifold; as an application, we tackled the important case of Gaussian measures, with the KL divergence as a measure of alignment between the low- and high-level SCMs. We pursued two different formulations. For the first, a nonsmooth Riemannian learning problem, we devised the LinSEPAL-ADMM and LinSEPAL-PG methods. For the second, a smooth Riemannian learning problem ensuring

the constructiveness of the CA, we developed CLinSEPAL. Our empirical assessment on synthetic data confirmed the effectiveness of our methods, and the application to brain data showcased the potential in real-world problems.

Our work paves the way for several exciting research directions. First, as it emerges from our Gaussian application, *linear CAs with different probability measures* deserve careful investigation. Second, studying the *nonlinear case* is a compelling avenue. We believe that deep and reinforcement learning paradigms, such as encoding-decoding and actor-critic architectures, hold promise for modeling nonlinear CA maps. Lastly, we view our work as a foundational step toward *observational causal abstraction learning*, bridging the gap between *CA learning* and *causal discovery* (Spirtes & Zhang, 2016). Our category-theoretic framework underscores the pivotal role of exogenous variables, drawing a path to translate *SCM identifiability* results into *CA identifiability* results. This suggests that, in some cases, interventional consistency may be achieved without relying on interventional data.

## Impact Statement

Our work is foundational, aiming at advancing the field of causal abstraction. Our proposed methods can be applied to different application domains, such as neuroscience. As demonstrated by our empirical assessment, the information resulting from their application is high-level and useful for a better understanding. Hence, we believe that the risks associated with improper usage of our techniques are low.

## Acknowledgements

The work of Gabriele D'Acunto and Paolo Di Lorenzo was supported by the SNS JU project 6G-GOALS (Strinati et al., 2024) under the EU's Horizon program Grant Agreement No 101139232. The work of Gabriele D'Acunto was also supported by the European Union under the Italian National Recovery and Resilience Plan (NRRP) of NextGenerationEU, partnership on " Telecommunications of the Future" (PE00000001 - program " RESTART"). The work of Yorgos Felekis was supported by the Onassis Foundation - Scholarship ID: F ZR 063-1/2021-2022.

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

## A. Extended Notation for the Appendix

Below is the notation used throughout the appendices. The set of integers from 1 to $n$ is $[n]$. The vectors of zeros and ones of size $n$ are $\mathbf{0}_n$ and $\mathbf{1}_n$. The identity matrix of size $n \times n$ is $\mathbf{I}_n$. The entry indexed by row $i$ and column $j$ is $a_{ij} = [\mathbf{A}]_{ij}$, $\mathrm{diag}(\mathbf{a})$ is the diagonal matrix having as diagonal the vector $\mathbf{a}$, while $\mathrm{diag}(\mathbf{A})$ is the diagonal of the matrix $\mathbf{A}$. The Frobenious norm is $\|\mathbf{A}\|_{\mathrm{F}}$. The set of positive definite matrices over $\mathbb{R}^{n \times n}$ is $\mathcal{S}_{++}^n$. That of symmetric ones as $\mathrm{Sym}(p)$. The column-wise vectorization of a matrix is $\mathrm{vec}\,()$. The Hadamard product is $\odot$. Function composition is $\circ$.

Let $\mathcal{M}(\mathcal{X}^n)$ be the set of Borel measures over $\mathcal{X}^n \subseteq \mathbb{R}^n$. Given a measure $\mu^n \in \mathcal{M}(\mathcal{X}^n)$ and a measurable map $\varphi^{\mathbf{V}}$, $\mathcal{X}^n \ni \mathbf{x} \overset{\varphi^{\mathbf{V}}}{\longmapsto} \mathbf{V}^\top \mathbf{x} \in \mathcal{X}^m$, we denote by $\varphi_{\#}^{\mathbf{V}}(\mu^n) \coloneqq \mu^n(\varphi^{\mathbf{V}^{-1}}(\mathbf{x}))$ the pushforward measure $\mu^m \in \mathcal{M}(\mathcal{X}^m)$. The proximal mapping of $h$ at $\mathbf{A}$ is $\mathrm{prox}_{\lambda\,h(\cdot)}(\mathbf{A}) = \arg\min_{\mathbf{V}} h(\mathbf{V}) + 1/(2\lambda) \|\mathbf{V} - \mathbf{A}\|_{\mathrm{F}}^2$, $\lambda \in \mathbb{R}^+$. The Euclidean gradient of a smooth $f$ is $\nabla f$, while the Riemannian one $\widetilde{\nabla} h$. The Euclidean subgradient of a nonsmooth $h$ is $\partial h$, the Riemannian instead $\widetilde{\partial} h$.

## B. Category Theory Essentials

Below are fundamental definitions and examples that are instrumental in providing the necessary background on category theory to understand our work. For a comprehensive overview of category theory see resources such as Mac Lane (2013); Perrone (2024).

**Definition B.1** (Category). A category $\mathsf{C}$ consists of

- A collection of objects, viz. $X$ in $\mathsf{C}$,
- A collection of morphisms, viz. $f : X \to Y$ in $\mathsf{C}$;

such that:

- Each morphism $f$ has assigned two objects of the category called source and target, respectively,
- Each object $X$ has an identity morphism $\mathrm{id}_X : X \to X$,
- Given $f : X \to Y$ and $g : Y \to Z$, than the composition exists, $g \circ f = h : X \to Z$.

These structures satisfy the following axioms:

- (Unitality) $\forall f : X \to Y$, $f \circ \mathrm{id}_X = f$ and $\mathrm{id}_Y \circ f = f$;
- (Associativity) Given $f$, $g$, and $h$ such that the compositions hold, then $h \circ (g \circ f) = (h \circ g) \circ f$.

*Example* 1. The following are some notable examples of categories:

- Indicate with Poset a partial order set. Poset can be viewed as the category whose objects are the elements $p$ and morphisms are order relations $p \le p'$. Notice that there is at most one morphism between two objects;
- $\mathsf{Vect}_{\mathbb{R}}$ is the category whose objects are real vector spaces and morphisms are linear maps;
- Prob is the category whose objects are probability measure spaces and morphisms measurable maps.

Arrows between categories are called *functors*, defined as follows:

**Definition B.2** (Functor). Consider $\mathsf{C}$ and $\mathsf{D}$ categories. A functor $F : \mathsf{C} \to \mathsf{D}$ consists of the following data:

- For each object $X$ in $\mathsf{C}$, an object $F(X)$ in $\mathsf{D}$;
- For each object morphism $f : X \to Y$ in $\mathsf{C}$, a morphism $F(f) : F(X) \to F(Y)$ in $\mathsf{D}$;

such that the following axioms hold:

- (Unitality) $\forall X$ in $\mathsf{C}$, $F(\mathrm{id}_X) = \mathrm{id}_{F(X)}$. In other words, the identity in $\mathsf{C}$ is mapped into the identity in $\mathsf{D}$.
- (Compositionality) $\forall f$ and $g$ in $\mathsf{C}$ such that the composition is defined, then $F(g \circ f) = F(g) \circ F(f)$. In other words, the composition in $\mathsf{C}$ is mapped into the composition in $\mathsf{D}$.

To ease the notation, in the sequel, we use $F^X$ and $F^f$ to denote $F(X)$ and $F(f)$, respectively. Finally, we can have arrows

between functors as well, called *natural transformations*:

**Definition B.3** (Natural transformation). Consider two categories $\mathsf{C}$ and $\mathsf{D}$, and two functors between them, namely $F : \mathsf{C} \to \mathsf{D}$ and $G : \mathsf{C} \to \mathsf{D}$. A natural transformation $\alpha : F \xrightarrow{\bullet} G$ consists of the following data:

- For each object $X$ in $\mathsf{C}$, a morphism $\alpha_X : F^X \to G^X$ in $\mathcal{D}$ called the component of $\alpha$ at $X$;
- For each morphism $f : X \to X'$ in $\mathsf{C}$, the following diagram commutes:

$$
\begin{array}{ccc}
F^X & \xrightarrow{\;\;F^f\;\;} & F^{X'} \\
\alpha_X \downarrow & & \downarrow \alpha_{X'} \\
G^X & \xrightarrow{\;\;G^f\;\;} & G^{X'}
\end{array}
\tag{11}
$$

A natural transformation can be thought of as a consistent system of arrows between two functors, invariant with respect to maps between the images of two functors.

## C. Causality and Causal Abstraction

This section provides additional definitions and examples related to SCMs and the CA framework.

### C.1. Mixing functions

A set of structural function in a Markovian SCM can be reduced to a set of mixing functions dependent only on the exogenous variables.

Given an SCM $\mathsf{M}^n$, recall that $\mathcal{F}$ is a set of $n$ functional assignments which define the values $X_i = f_i(\mathcal{P}_i, Z_i), \forall\, i \in [n]$, with $\mathcal{P}_i \subseteq \mathcal{X} \setminus \{X_i\}$. Denote by $\mathcal{Z}^{\mathcal{A}_i} \subseteq \mathcal{Z} \setminus \{Z_i\}$ the set of exogenous variables corresponding to the ancestors of $X_i$, where $\mathcal{A}_i \subseteq [n] \setminus \{i\}$. According to $\mathcal{F}$, we can identify a set of mixing functions $\mathcal{M} = \{m_1, \ldots, m_n\}$ such that the values of the endogenous random variables are equivalently expressed as $x_i = m_i\left(\{z_j\}_{j \in \mathcal{A}_i}, z_i\right), \forall\, i \in [n]$.

Further, we can also characterize the product probability measure implied by the SCM purely in terms of the exogenous variables, viz. $\chi^{\mathcal{X}} = \prod_{i \in [n]} P\left(X_i | \mathcal{Z}^{\mathcal{A}_i}, Z_i\right)$.

As an example, consider a causal relation $x_1 \to x_2$. In the linear SCM with additive noise (Bollen, 1989; Shimizu et al., 2006) setting we have

$$
\begin{cases}
x_1 = z_1\,, \\
x_2 = c_{2,1} x_1 + z_2 = c_{2,1} z_1 + z_2\,.
\end{cases}
\tag{12}
$$

Again, for the post-nonlinear model (Zhang & Hyvärinen, 2009), we get

$$
\begin{cases}
x_1 = f_{1,1}(z_1) = m_1(z_1)\,, \\
x_2 = f_{2,2}(f_{2,1}(x_1) + z_2) \\
\quad = f_{2,2}(f_{2,1} \circ f_{1,1}(z_1) + z_2) \\
\quad = m_2(z_1, z_2)\,.
\end{cases}
\tag{13}
$$

### C.2. Interventional consistency

A typical requirement imposed on CA maps is that they act in a consistent way with respect to interventions (Rischel, 2020).

**Definition C.1** (Interventional consistency). Given an $\boldsymbol{\alpha}$-abstraction between $\mathsf{M}^\ell$ and $\mathsf{M}^h$ and a set $\mathcal{I}$ of hard interventions on $\mathcal{X}_{\mathcal{I}}^h \subseteq \mathcal{X}^h$, the abstraction is *interventionally consistent* if, for any intervention in $\mathcal{I}$ and for every set of target variable $\mathcal{Y}_{\mathcal{I}}^h \subseteq \mathcal{X}^h \setminus \mathcal{X}_{\mathcal{I}}^h$, the following diagram commutes:

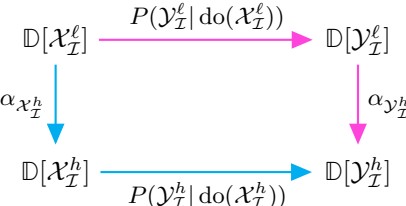

or equivalently,

$$\alpha_{\mathcal{Y}_{\mathcal{I}}^{h}}(P(\mathcal{Y}_{\mathcal{I}}^{\ell}|\,\mathrm{do}(\mathcal{X}_{\mathcal{I}}^{\ell}))) = P(\mathcal{Y}_{\mathcal{I}}^{h}|\alpha_{\mathcal{X}_{\mathcal{I}}^{h}}(\mathrm{do}(\mathcal{X}_{\mathcal{I}}^{h}))), \tag{14}$$

where $\mathcal{X}_{\mathcal{I}}^{\ell} = m^{-1}(\mathcal{X}_{\mathcal{I}}^{h})$ and $\mathcal{Y}_{\mathcal{I}}^{\ell} = m^{-1}(\mathcal{Y}_{\mathcal{I}}^{h})$.

Essentially, commutativity suggests that we obtain equivalent intervention outcomes in two different ways: *(i)* either by intervening on the low-level model and then abstracting or, *(ii)* by abstracting to the high-level model and then intervening in an equivalent fashion.

### C.3. Linear abstraction

The class of abstractions may be restricted by an assumption of the form of the abstraction map (Massidda et al., 2024):

**Definition C.2** (Linear abstraction). Given an $\alpha$-abstraction $\boldsymbol{\alpha} = \langle \mathcal{R}, m, \alpha \rangle$ from $\mathsf{M}^{\ell}$ to $\mathsf{M}^{h}$, the abstraction is linear if $\alpha = \mathbf{V}^{\top} \in \mathbb{R}^{h \times \ell}$.

### C.4. Constructive abstraction

A particularly well-behaved form of abstraction is a constructive abstraction. In the context of the $\tau$-abstraction framework (Beckers & Halpern, 2019), a constructive abstraction is an abstraction such that: *(i)* the variable mapping defines a clustering of the low-level variables (*constructivity*); *(ii)* consistency holds for all high-level interventions (*strongness*); *(iii)* the value map is surjective and it implies a map between exogenous values and between interventions (*$\tau$-abstraction*). In the $\alpha$-framework a few of these properties hold by construction; thus, we define a constructive abstraction as (Schooltink & Zennaro, 2024):

**Definition C.3** (Constructive abstraction). Given an $\alpha$-abstraction $\boldsymbol{\alpha} = \langle \mathcal{R}, m, \alpha \rangle$ from $\mathsf{M}^{\ell}$ to $\mathsf{M}^{h}$, the abstraction is constructive if the abstraction is interventionally consistent and implies the existence of a map $\alpha_U : \mathcal{Z}^{\ell} \to \mathcal{Z}^{h}$ between exogenous variables.

### C.5. Measure-theoretic definition of an SCM

Any SCM can be defined in terms of the probability measure spaces underlying it:

**Definition C.4** (Measure-theoretic SCM). A (Markovian) SCM $\mathsf{M}^{n}$ is a triple $\langle (\mathcal{U}, \Sigma_{\mathcal{U}}, \zeta), (\mathcal{V}, \Sigma_{\mathcal{V}}, \chi), \mathcal{M} \rangle$ where:

- $(\mathcal{U}, \Sigma_{\mathcal{U}}, \zeta)$ is a probability space associated with exogenous variables. Specifically, it consists of the product probability measure $\zeta = \zeta_1 \times \ldots \times \zeta_n$ on the product measurable space $(\mathcal{U}, \Sigma_{\mathcal{U}})$ where $\mathcal{U} = \mathcal{U}_1 \times \ldots \times \mathcal{U}_n$ is a product set and $\Sigma_{\mathcal{U}} = \Sigma_{\mathcal{U}_1} \otimes \ldots \otimes \Sigma_{\mathcal{U}_n}$ is a product $\sigma$-algebra. The probability measure is such that, for each $\mathcal{W}_1 \in \Sigma_{\mathcal{U}_1}, \ldots, \mathcal{W}_n \in \Sigma_{\mathcal{U}_n}$, we have

$$\zeta_1 \times \ldots \times \zeta_n(\mathcal{W}_1 \times \ldots \times \mathcal{W}_n) = \zeta_1(\mathcal{W}_1) \times \ldots \times \zeta_n(\mathcal{W}_n); \tag{15}$$

- $(\mathcal{V}, \Sigma_{\mathcal{V}}, \chi)$ is a probability space associated with endogenous variables consisting of a joint probability measure $\chi$ on the product measurable space $(\mathcal{V}, \Sigma_{\mathcal{V}}) = (\mathcal{V}_1 \times \ldots \times \mathcal{V}_n, \Sigma_{\mathcal{V}_1} \otimes \ldots \otimes \Sigma_{\mathcal{V}_n})$;
- $\mathcal{M}$ is a set of $n$ mixing measurable maps $\varphi^{m_i}$ (cf. Def. 2.1) such that the joint probability measure $\chi$ factorizes as

$$\chi = \bigtimes_{i=1}^{n} \varphi_{\#}^{m_i} \left( \mu_i \left( \mathcal{U}_i \times \mathcal{U}^{\mathcal{A}_i} \right) \right); \tag{16}$$

where $\mathcal{U}^{\mathcal{A}_i} = \bigtimes_{j \in \mathcal{A}_i} \mathcal{U}_j$, and, denoting by $\Sigma_{\mathcal{U}^{\mathcal{A}_i}} = \bigotimes_{j \in \mathcal{A}_i} \Sigma_{\mathcal{U}_j}$, $\mu_i$ is a probability measure on the product measurable space $\left( \mathcal{U}_i \times \mathcal{U}^{\mathcal{A}_i}, \Sigma_{\mathcal{U}_i} \otimes \Sigma_{\mathcal{U}^{\mathcal{A}_i}} \right)$.

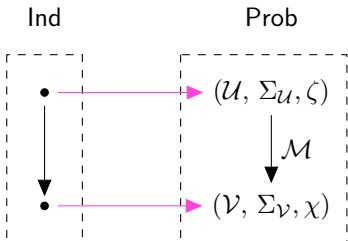

**Figure 5: An SCM is a functor (purple arrows) from** Ind **(left) to** Prob **(right).**

## D. Category-theoretic Formalization

This section extends the category-theoretic formalization introduced in the main paper to intervened models and abstraction.

Recall the category-theoretic definition from the main paper:

**Definition D.1** (Category-theoretic SCM). An SCM is a functor $\mathsf{M}^n : \mathsf{Ind} \to \mathsf{Prob}$, mapping the source node of Ind to the probability space associated with the exogenous variables $(\mathcal{U}, \Sigma_{\mathcal{U}}, \zeta)$, the sink node of Ind to the probability space associated with the endogenous variables $(\mathcal{V}, \Sigma_{\mathcal{V}}, \chi)$, and the only edge of Ind to the measurable map induced by the set $\mathcal{F}$ of functional assignments.

Fig. 5 offers a depiction of an SCM as a functor.

In the same vein, we can have a functorial representation for intervened SCMs as well. However, instead of representing directly the post-interventional model $\mathsf{M}^n_\iota$ as in Def. D.1, we will adopt a representation that is closer to the intervention operator itself. First, notice that, whenever the domains of the variables of an SCM are continuous, we can represent an intervention as a measurable map by relying on the truncation formula (Pearl, 2009):

**Lemma D.2.** *Given a continuous Markovian SCM* $\mathsf{M}^n = \langle (\mathcal{U}, \Sigma_{\mathcal{U}}, \zeta), (\mathcal{V}, \Sigma_{\mathcal{V}}, \chi), \mathcal{M} \rangle$ *and an intervention* $\iota$ *on* $\mathsf{M}^n$, *there exists a measurable map* $\varphi_\iota$ *from the probability space of endogenous variables of the pre-interventional SCM* $(\mathcal{V}, \Sigma_{\mathcal{V}}, \chi)$ *to the probability space of endogenous variables of the post-interventional SCM* $(\mathcal{V}_\iota, \Sigma_{\mathcal{V}_\iota}, \chi_\iota)$.

*Proof.* Given a Markovian SCM $\mathsf{M}^n$, the probability measure $\chi$ over the measure space of endogenous variables $(\mathcal{V}, \Sigma_{\mathcal{V}}, \chi)$ can be expressed by through the factorization over the endogenous variables $\chi = \prod_{i \in [n]} P(X_i | \mathcal{P}_i, Z_i)$, with $\chi_i = P(X_i | \mathcal{P}_i, Z_i)$. Given intervention $\iota = \mathrm{do}(\mathcal{X}^\iota = \mathbf{x}^\iota)$ on $\mathsf{M}^n$, the new post-interventional measure $\chi^\iota$ can be computed through the truncation formula (Pearl, 2009):

$$\chi^\iota = \begin{cases} \prod_{i \in [n], X_i \notin \mathcal{X}^\iota} P(X_i | \mathcal{P}_i, Z_i) & \text{if } \mathcal{X}^\iota = \mathbf{x}^\iota \\ 0 & \text{if } \mathcal{X}^\iota \neq \mathbf{x}^\iota \end{cases} \tag{17}$$

We can now define a measurable map $\varphi^\iota$ connecting $(\mathcal{V}, \Sigma_{\mathcal{V}}, \chi)$ and $(\mathcal{V}, \Sigma_{\mathcal{V}}, \chi^\iota)$ such that $\varphi^\iota_{\#}(\chi) = \chi^\iota$. Specifically, for each $X_i \in \mathcal{X}^\iota$, $\varphi(X_i) = x^\iota_i$, thus guaranteeing the distribution on the second line of Eq. (17); for each $X_i \notin \mathcal{X}^\iota$, denoting by $\chi^\iota_i = P(X_i | \mathcal{P}_i, Z_i)$ evaluated at $\mathcal{X}^\iota = \mathbf{x}^\iota$ as in the first line of Eq. (17), we solve a measure transport problem (Marzouk et al., 2016) from $\chi_i$ to $\chi^\iota_i$ which, in the continuous case, guarantees a transport map over the domains that satisfies the distribution on the first line of Eq. (17). $\qquad\square$

We can then encode an intervened model as follows:

**Definition D.3** (Category-theoretic post-interventional SCM). A post-interventional SCM is a functor $\mathsf{M}^n_\iota : \mathsf{Ind} \to \mathsf{Prob}$, where the functor maps the source node of Ind to the probability space associated with the endogenous variables of the pre-interventional SCM $(\mathcal{V}, \Sigma_{\mathcal{V}}, \chi)$, the sink node of Ind to the probability space associated with the endogenous variables of the post-interventional SCM $(\mathcal{V}_\iota, \Sigma_{\mathcal{V}_\iota}, \chi_\iota)$, and the only edge of Ind to the function $\varphi_\iota$ encoding the intervention $\iota$.

This construction gives rise to the structure in Fig. 6 and an immediate category-theory expression of abstraction equivalent to Def.2.3:

**Lemma D.4.** *An interventionally consistent abstraction is a singular natural transformation* $\boldsymbol{\alpha}$, *that is, a morphism* $\alpha_{\mathcal{V}}$ *in* Prob, *that, for all intervention in* $\mathcal{I}$ *guarantees the commutativity of the diagrams constructed from Fig. 2.*

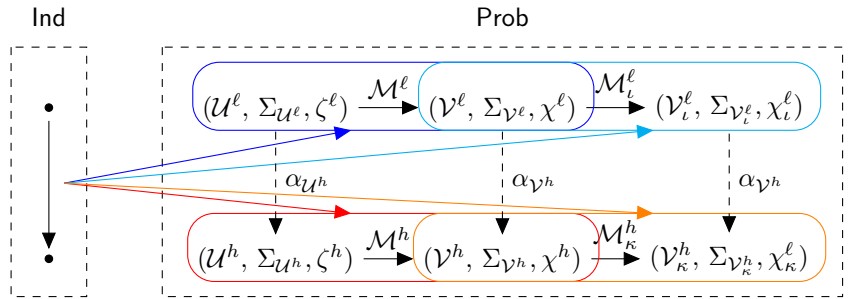

**Figure 6: Representation of** $\mathsf{M}^\ell$ **(blue),** $\mathsf{M}^\ell_\iota$ **(cyan),** $\mathsf{M}^h$ **(red),** $\mathsf{M}^h_\iota$ **(orange) as functors. An abstraction is just a natural transformation, that is, a set of commuting arrows in** Prob **(dashed black). Notice two commuting diagrams in** Prob**: the first observational one rooted on the exogenous variables (**$\mathcal{M}^h \circ \alpha_{\mathcal{U}^h} = \alpha_{\mathcal{V}^h} \circ \mathcal{M}^\ell$**), the second interventional one connecting observational and interventional model (**$\mathcal{M}^h_\kappa \circ \alpha_{\mathcal{V}^h} = \alpha_{\mathcal{V}^h} \circ \mathcal{M}^\ell_\iota$**).**

*Proof.* Recall the definition of interventional consistency in Def. C.1:

$$\alpha_{\mathcal{Y}^h_{\mathcal{I}}}(P(\mathcal{Y}^\ell_{\mathcal{I}}|\operatorname{do}(\mathcal{X}^\ell_{\mathcal{I}}))) = P(\mathcal{Y}^h_{\mathcal{I}}|\alpha_{\mathcal{X}^h_{\mathcal{I}}}(\operatorname{do}(\mathcal{X}^h_{\mathcal{I}}))). \tag{18}$$

Let us relate this definition to our categorical notation. First, $\alpha_{\mathcal{Y}^h_{\mathcal{I}}}$ and $\alpha_{\mathcal{X}^h_{\mathcal{I}}}$ are components of the abstraction map $\alpha$; in the categorical notation, this map correspond to $\alpha_{\mathcal{V}}$. The probability distribution $P(\mathcal{Y}^\ell_{\mathcal{I}}|\operatorname{do}(\mathcal{X}^\ell_{\mathcal{I}}))$ is a distribution in the low-level model; with no loss of generality, assuming $\mathcal{Y}^\ell_{\mathcal{I}}$ to encompass all the non-intervened variables, this distribution correspond to the measure $\chi^\ell_\iota$; furthermore, the interventional measure $\chi^\ell_\iota$ can be obtained through the pushforward of the observational measure $\chi^\ell$ via the interventional mixing functions $\mathcal{M}^\ell_\iota$, as by Lemma D.4. Finally, the probability distribution $P(\mathcal{Y}^h_{\mathcal{I}}|\alpha_{\mathcal{X}^h_{\mathcal{I}}}(\operatorname{do}(\mathcal{X}^h_{\mathcal{I}}))$ is a distribution in the high-level model; again, with no loss of generality, assuming $\mathcal{Y}^h_{\mathcal{I}}$ to encompass all the non-intervened variables, this distribution correspond to the measure $\chi^h_\kappa$, where $\kappa$ is the abstraction of the terms in $\iota$. Also, as before, the interventional measure $\chi^h_\kappa$ can be obtained through the pushforward of the observational measure $\chi^h$ via the interventional mixing functions $\mathcal{M}^h_\kappa$, thanks to Lemma D.4. We then obtain a rewriting of abstraction as:

$$\alpha_{\mathcal{V}} \circ \mathcal{M}^\ell_\iota = \mathcal{M}^h_\kappa \circ \alpha_{\mathcal{V}}. \tag{19}$$

corresponding to the commutativity of the right diagram in Fig. 6, for all interventions. $\qquad\square$

## E. Stiefel Manifold

We now provide a short review of the Stiefel manifold, referring the interested reader to (Absil et al., 2008; Boumal, 2023) for a comprehensive discussion.

Given $\ell, h \in \mathbb{N}$, $h < \ell$, the Stiefel manifold is the set of $\ell \times h$ matrices with orthonormal columns, mathematically

$$\operatorname{St}(\ell, h) \coloneqq \{\mathbf{V} \in \mathbb{R}^{\ell \times h} \mid \mathbf{V}^\top \mathbf{V} = \mathbf{I}_h\}. \tag{20}$$

Consider the function $g : \mathbb{R}^{\ell \times h} \to \operatorname{Sym}(h)$, $g(\mathbf{V}) \coloneqq \mathbf{V}^\top \mathbf{V} - \mathbf{I}^h$. It is well-known that $g$ is a generating function for $\operatorname{St}(\ell, h)$, thus making it an embedded submanifold of $\mathbb{R}^{\ell \times h}$, with dimension $\dim \mathbb{R}^{\ell \times h} - \dim \operatorname{Sym}(h) = \ell h - h(h+1)/2$. Given a point of the manifold $\mathbf{V}$, the tangent space to $\operatorname{St}(\ell, h)$ can be defined implicitly as the kernel of the differential of $g$ at $\mathbf{V}$,

$$T_{\mathbf{V}}\operatorname{St}(\ell, h) \coloneqq \{\mathbf{G} \in \mathbb{R}^{\ell \times h} \mid \mathbf{V}^\top \mathbf{G} + \mathbf{G}^\top \mathbf{V} = 0\}. \tag{21}$$

We consider the Riemannian metric as the restriction of the Euclidean product between two matrices in $\mathbb{R}^{\ell \times h}$ to $\operatorname{St}(\ell, h)$. Accordingly, given $\mathbf{A}, \mathbf{B} \in T_{\mathbf{V}}\operatorname{St}(\ell, h)$, we have $\langle \mathbf{A}, \mathbf{B} \rangle_{\mathbf{V}} = \operatorname{Tr} \mathbf{A}^\top \mathbf{B}$. The tangent space linearizes the manifold around $\mathbf{V}$, then, we can move away from $\mathbf{V}$ along the directions in $T_{\mathbf{V}}\operatorname{St}(\ell, h)$. However, to make such a movement smooth along the manifold, we employ the *retraction map* $\operatorname{R}_{\mathbf{V}}(:) T_{\mathbf{V}}\operatorname{St}(\ell, h) \to \operatorname{St}(\ell, h)$. The retraction has to satisfy the following conditions

$$\textit{(i) } \operatorname{R}_{\mathbf{V}}(\mathbf{0}_{\ell \times h}) = \mathbf{V}, \quad \text{and} \quad \textit{(ii) } \lim_{\mathbf{G} \to \mathbf{0}_{\ell \times h}} \frac{\|\operatorname{R}_{\mathbf{V}}(\mathbf{G}) - (\mathbf{V} + \mathbf{G})\|_{\mathrm{F}}}{\|\mathbf{G}\|_{\mathrm{F}}} = 0. \tag{22}$$

Among the canonical retractions, we have

$$
\begin{aligned}
\mathrm{R}_{\mathbf{V}}^{\mathrm{QR}}(\mathbf{G}) &= \mathrm{qf}(\mathbf{V} + \mathbf{G}), \quad \text{[QR retraction]} \\
\mathrm{R}_{\mathbf{V}}^{\mathrm{Polar}}(\mathbf{G}) &= (\mathbf{V} + \mathbf{G})(\mathbf{I}^h - \mathbf{V}^\top \mathbf{V})^{\frac{1}{2}}, \quad \text{[Polar retraction]} \\
\mathrm{R}_{\mathbf{V}}^{\mathrm{Caley}}(\mathbf{G}) &= (\mathbf{I}^\ell - \tfrac{1}{2}\mathbf{W}(\mathbf{G}))^{-1}(\mathbf{I}^\ell + \tfrac{1}{2}\mathbf{W}(\mathbf{G}))\mathbf{V}; \quad \text{[Caley retraction]}
\end{aligned}
\tag{23}
$$

where qf indicates the $\mathbf{Q}$ factor of the QR decomposition, and $\mathbf{W}(\mathbf{G}) = (\mathbf{I}^\ell - \tfrac{1}{2}\mathbf{V}\mathbf{V}^\top)\mathbf{G}\mathbf{V}^\top - \mathbf{V}\mathbf{G}^\top(\mathbf{I}^\ell - \tfrac{1}{2}\mathbf{V}\mathbf{V}^\top)$.

Finally, the normal space to the manifold at $\mathbf{V}$ has the following explicit form

$$
N_{\mathbf{V}}\mathrm{St}(\ell, h) := \{\mathbf{V}\mathbf{S} \mid \mathbf{S} \in \mathrm{Sym}(h)\}.
\tag{24}
$$

Starting from Eq. (24), the orthogonal projection to $T_{\mathbf{V}}\mathrm{St}(\ell, h)$, namely $\mathrm{Proj}_{\mathbf{V}}$, has to be such that $\mathbf{G} - \mathrm{Proj}_{\mathbf{V}}\mathbf{G}$ lies onto $N_{\mathbf{V}}\mathrm{St}(\ell, h)$, i.e.,

$$
\mathbf{G} - \mathrm{Proj}_{\mathbf{V}}\mathbf{G} = \mathbf{V}\mathbf{S}.
\tag{25}
$$

Plugging Eq. (25) into Eq. (21), it can be derived that

$$
\mathrm{Proj}_{\mathbf{V}}\mathbf{G} = \left(\mathbf{I}^\ell - \mathbf{V}\mathbf{V}^\top\right)\mathbf{G} + \mathbf{V}\frac{\left(\mathbf{V}^\top\mathbf{G} - \mathbf{G}^\top\mathbf{V}\right)}{2}.
\tag{26}
$$

Finally, for $\mathrm{St}(\ell, h)$ (and in general for Riemannian submanifolds) the Riemannian gradient of $f$ at $\mathbf{V}$ is the orthogonal projection of $\nabla_{\mathbf{V}}f$ to $T_{\mathbf{V}}\mathrm{St}(\ell, h)$. Mathematically, starting from Eq. (26), we have

$$
\widetilde{\nabla}_{\mathbf{V}}f = \mathrm{Proj}_{\mathbf{V}}\nabla_{\mathbf{V}}f.
\tag{27}
$$

## F. Information-theoretic Distance on Spaces of Different Dimensionality

Two types of distances can be defined as follows using an affine map $\varphi^{\mathbf{V}, b}$ (Cai & Lim, 2022).

**Definition F.1** (Embedding and projection distances). Let $\ell, h \in \mathbb{N}$ with $h \le \ell$, and let $\varphi^{\mathbf{V}, b} = \mathbf{V}^\top x + b : \mathbb{R}^\ell \to \mathbb{R}^h$ be an affine map with $\mathbf{V} \in \mathrm{St}(\ell, h)$ and $b \in \mathbb{R}^\ell$. For any measures $\chi^h \in \mathcal{M}(\mathbb{R}^h)$ and $\chi^\ell \in \mathcal{M}(\mathbb{R}^\ell)$, the *set of embeddings* of $\chi^h$ into $\mathbb{R}^\ell$ is the set of $\ell$-dimensional measures, defined as follows:

$$
\Phi^+(\chi^h, \ell) := \{\alpha \in \mathcal{M}(\mathbb{R}^\ell) : \varphi_\#^{\mathbf{V}, b}(\alpha) = \chi^h\}
\tag{28}
$$

Similarly, the *set of projections* of $\chi^\ell$ into $\mathbb{R}^h$ is the set of $h$-dimensional measures defined as:

$$
\Phi^-(\chi^\ell, h) := \{\beta \in \mathcal{M}(\mathbb{R}^h) : \varphi_\#^{\mathbf{V}, b}(\chi^\ell) = \beta\}
\tag{29}
$$

Now, for any given distance measure $D(\cdot, \cdot)$ defined in $\mathcal{M}(\mathbb{R}^\ell)$, we can define the *embedding distance* $D^+(\chi^h, \chi^\ell) := \inf_{\alpha \in \Phi^+(\chi^h, \ell)} D(\alpha, \nu)$ and the *projection distance* $D^-(\chi^h, \chi^\ell) := \inf_{\beta \in \Phi^-(\chi^\ell, h)} D(\chi^h, \beta)$.

Embedding and projection distances can measure distances between probability measures of different dimensions. Additionally, Theorem I.2 (Cai & Lim, 2022) states that the former two distances are equivalent, that is, for a number of different distance metrics and $\phi$-divergences, $D^+(\chi^h, \chi^\ell) = D^-(\chi^h, \chi^\ell) = \hat{D}(\chi^h, \chi^\ell)$, implying that computing the embedding distance or the projection distance yields the same result.

## G. Proofs

**Theorem 4.3.** *Let* $\chi^\ell \sim N(\mathbf{0}_\ell, \boldsymbol{\Sigma}^\ell)$, $\chi^h \sim N(\mathbf{0}_h, \boldsymbol{\Sigma}^h)$, *where* $\boldsymbol{\Sigma}^\ell \in \mathcal{S}_{++}^\ell$ *and* $\boldsymbol{\Sigma}^h \in \mathcal{S}_{++}^h$. *Denote by* $0 < \lambda_1 \le \ldots \le \lambda_\ell$ *the eigenvalues of* $\boldsymbol{\Sigma}^\ell$, *and by* $0 < \kappa_1 \le \ldots \le \kappa_h$ *those of* $\boldsymbol{\Sigma}^h$. *If a linear CA* $\mathbf{V} \in \mathrm{St}(\ell, h)$ *complying with SEP from* $\chi^\ell$ *to* $\chi^h$ *exists, then*

$$
\lambda_i \le \kappa_i \le \lambda_{i+\ell-h}, \quad \forall\, i \in [h].
\tag{4}
$$

*Proof.* If a linear CA $\mathbf{V} \in \mathrm{St}(\ell, h)$ exists, then $\chi^h = \varphi_{\#}^{\mathbf{V}^\top}(\chi^\ell)$. This implies that the kernel of any information-theoretic metric or $\phi$-divergence is nonempty, and also $\mathbf{V}^\top \mathbf{\Sigma}^\ell \mathbf{V} = \mathbf{\Sigma}^h$. The latter determines the eigenvalues of $\mathbf{V}^\top \mathbf{\Sigma}^\ell \mathbf{V}$ are those of $\mathbf{\Sigma}^h$. By the Ostrowski's theorem for rectangular $\mathbf{V}$ — cf. Th.3.2 in (Higham & Cheng, 1998) — we have

$$\kappa_i = \vartheta_i \mu_i, \quad \text{with } i \in [h], \tag{30}$$

where

$$\lambda_i \leq \mu_i \leq \lambda_{i+\ell-h}; \tag{31}$$

and

$$\mathrm{eigvls}(\mathbf{V}^\top \mathbf{V})_1 \leq \vartheta_i \leq \mathrm{eigvls}(\mathbf{V}^\top \mathbf{V})_h. \tag{32}$$

Since $\mathbf{V} \in \mathrm{St}(\ell, h)$, by Eq. (32) $\vartheta_i = 1$ for each $i \in [h]$. Substituting the latter into Eq. (30), we get $\kappa_i = \mu_i$, thus obtaining Eq. (4) by Eq. (31). $\qquad\square$

**Proposition 5.1.** *Consider the function*

$$f(\mathbf{A}) = \mathrm{Tr}\left\{\left(\mathbf{A}^\top \mathbf{\Sigma}^\ell \mathbf{A}\right)^{-1} \mathbf{\Sigma}^h\right\} + \log \det\left\{\mathbf{A}^\top \mathbf{\Sigma}^\ell \mathbf{A}\right\}. \tag{9}$$

*Eq. (9) is smooth for $\mathbf{A} \in \mathrm{St}(\ell, h)$. Additionally, define $\widetilde{\mathbf{A}} := \left(\mathbf{A}^\top \mathbf{\Sigma}^\ell \mathbf{A}\right)^{-1}$. The gradient of $f(\mathbf{A})$ is*

$$\nabla_{\mathbf{A}} f = 2\left(\mathbf{\Sigma}^\ell \mathbf{A} \widetilde{\mathbf{A}}\right)\left(\mathbf{I}_h - \mathbf{\Sigma}^h \widetilde{\mathbf{A}}\right), \tag{10}$$

*Proof.* Consider the first term $\mathrm{Tr}\left\{\left(\mathbf{A}^\top \mathbf{\Sigma}^\ell \mathbf{A}\right)^{-1} \mathbf{\Sigma}^h\right\}$ in Eq. (9). We have that $\mathrm{Tr}\left\{\left(\mathbf{A}^\top \mathbf{\Sigma}^\ell \mathbf{A}\right)^{-1} \mathbf{\Sigma}^h\right\}$ is well-defined and smooth in case $\mathbf{A}^\top \mathbf{\Sigma}^\ell \mathbf{A}$ is positive definite (Boyd & Vandenberghe, 2004). If $\mathbf{A}^\top \mathbf{\Sigma}^\ell \mathbf{A} \in \mathcal{S}_{++}^h$, for all $\mathbf{y} \in \mathbb{R}^h$, $\mathbf{y} \neq \mathbf{0}_h$, it holds $\mathbf{y}^\top \mathbf{A}^\top \mathbf{\Sigma}^\ell \mathbf{A} \mathbf{y} > 0$. By defining $\mathbb{R}^\ell \ni \mathbf{z} := \mathbf{A}\mathbf{y}$, this is equivalent to say $\mathbf{z}^\top \mathbf{\Sigma}^\ell \mathbf{z} > 0, \forall \mathbf{y} \neq \mathbf{0}_h$. Since $\mathbf{\Sigma}^\ell \in \mathcal{S}_{++}^\ell$ by assumption, we have to prove that $\mathbf{z} \neq \mathbf{0}_\ell, \forall \mathbf{y} \neq \mathbf{0}_h$. Consider that exists $\tilde{\mathbf{y}} \neq \mathbf{0}_h$ such that $\tilde{\mathbf{z}} = \mathbf{A}\tilde{\mathbf{y}} = \mathbf{0}_\ell$. This means that

$$\mathbf{0}_h = \mathbf{A}^\top \tilde{\mathbf{z}} = \mathbf{A}^\top \mathbf{A} \tilde{\mathbf{y}} = \tilde{\mathbf{y}} \neq \mathbf{0}_h; \tag{33}$$

which is a contradiction. Hence $\mathbf{A}^\top \mathbf{\Sigma}^\ell \mathbf{A} \in \mathcal{S}_{++}^h$ and $\mathrm{Tr}\left\{\left(\mathbf{A}^\top \mathbf{\Sigma}^\ell \mathbf{A}\right)^{-1} \mathbf{\Sigma}^h\right\}$ is smooth over $\mathrm{St}(\ell, h)$. Consider now $\log \det\left\{\mathbf{A}^\top \mathbf{\Sigma}^\ell \mathbf{A}\right\}$ in Eq. (8). Since $\mathbf{A}^\top \mathbf{\Sigma}^\ell \mathbf{A} \in \mathcal{S}_{++}^h$, also this latter term is well-defined and smooth.

Let $\widetilde{\mathbf{A}} := \left(\mathbf{A}^\top \mathbf{\Sigma}^\ell \mathbf{A}\right)^{-1}$. The gradient in Eq. (10) follows from the application of the following rules of matrix calculus (Brookes, 2020),

$$\text{(i)} \quad \nabla \mathrm{Tr}\left\{\left(\mathbf{A}^\top \mathbf{\Sigma}^\ell \mathbf{A}\right)^{-1} \mathbf{\Sigma}^h\right\} = -2\mathbf{\Sigma}^\ell \mathbf{A} \widetilde{\mathbf{A}} \mathbf{\Sigma}^h \widetilde{\mathbf{A}} \quad \text{and} \quad \text{(ii)} \quad \nabla \log \det\left\{\mathbf{A}^\top \mathbf{\Sigma}^\ell \mathbf{A}\right\} = 2\mathbf{\Sigma}^\ell \mathbf{A} \widetilde{\mathbf{A}}, \tag{34}$$

$\qquad\square$

# H. LinSEPAL-ADMM

Let us recall below the nonsmooth Riemannian problem we have to solve.

**Problem 2.** *Given $\mathbf{\Sigma}^\ell \in \mathcal{S}_{++}^\ell$, $\mathbf{\Sigma}^h \in \mathcal{S}_{++}^h$, $\mathbf{D} \in \{0,1\}^{\ell \times h}$, and $\lambda \in \mathbb{R}_+$, the CA is the transpose of*

$$\mathbf{V}^\star = \underset{\mathbf{V} \in \mathrm{St}(\ell, h)}{\arg\min} \ f(\mathbf{V}) + \lambda \underbrace{\|\mathbf{D} \odot \mathbf{V}\|_1}_{h(\mathbf{V})}. \tag{5}$$

*Here, $f(\mathbf{V})$ follows Eq. (3), omitting the constant $C$.*

The structure of the objective in (5), separating into smooth (cf. Proposition 5.1) and nonsmooth terms, makes the *alternating direction method of multipliers* (ADMM, Boyd et al., 2011) an appealing optimization framework for deriving a solution. This is the rationale behind the general framework *manifold ADMM* (Kovnatsky et al., 2016), that we decline to our setting in the following, thus obtaining the LinSEPAL-ADMM algorithm.

Starting from (5), we add a splitting variable $\mathbf{Y} \in \mathbb{R}^{\ell \times h}$ to be optimized over the Euclidean space to handle the non-smooth term $h(\mathbf{V})$:

$$\min_{\mathbf{V} \in \mathrm{St}(\ell,h), \mathbf{Y} \in \mathbb{R}^{\ell \times h}} \quad \mathrm{Tr}\left\{ \left(\mathbf{V}^\top \mathbf{\Sigma}^\ell \mathbf{V}\right)^{-1} \mathbf{\Sigma}^h \right\} + \log \det\left\{\mathbf{V}^\top \mathbf{\Sigma}^\ell \mathbf{V}\right\} + \lambda \|\mathbf{Y}\|_1, \tag{P2}$$
$$\text{subject to} \quad \mathbf{Y} - \mathbf{D} \odot \mathbf{V} = \mathbf{0}_{\ell \times h}.$$

At this point, following (Boyd et al., 2011), by denoting by $\mathbf{U} \in \mathbb{R}^{\ell \times h}$ the scaled dual variable, and by $\rho \in \mathbb{R}^+$ the ADMM stepsize, the scaled augmented Lagrangian reads as

$$L_\rho\left(\mathbf{V}, \mathbf{Y}, \mathbf{U}\right) = \mathrm{Tr}\left\{\left(\mathbf{V}^\top \mathbf{\Sigma}^\ell \mathbf{V}\right)^{-1} \mathbf{\Sigma}^h\right\} + \log \det\left\{\mathbf{V}^\top \mathbf{\Sigma}^\ell \mathbf{V}\right\} + \lambda \|\mathbf{Y}\|_1 + \frac{\rho}{2}\|\mathbf{D} \odot \mathbf{V} - \mathbf{Y} + \mathbf{U}\|_{\mathrm{F}}^2. \tag{35}$$

Starting from Eq. (35), the ADMM updates at the $k$-th iteration are

$$\begin{aligned}
\mathbf{V}^{k+1} &= \underset{\mathbf{V} \in \mathrm{St}(\ell,h)}{\arg\min} \ L_\rho\left(\mathbf{V}, \mathbf{Y}^k, \mathbf{U}^k\right), \\
\mathbf{Y}^{k+1} &= \underset{\mathbf{Y} \in \mathbb{R}^{\ell \times h}}{\arg\min} \ L_\rho\left(\mathbf{V}^{k+1}, \mathbf{Y}, \mathbf{U}^k\right), \\
\mathbf{U}^{k+1} &= \mathbf{U}^k + \mathbf{D} \odot \mathbf{V}^{k+1} - \mathbf{Y}^{k+1}.
\end{aligned} \tag{R1}$$

**Solution for $\mathbf{V}^{k+1}$.**

The update for $\mathbf{V}^{k+1}$ in (R1) reads as

$$\mathbf{V}^{k+1} = \underset{\mathbf{V} \in \mathrm{St}(\ell,h)}{\arg\min} \ \mathrm{Tr}\left\{\left(\mathbf{V}^\top \mathbf{\Sigma}^\ell \mathbf{V}\right)^{-1} \mathbf{\Sigma}^h\right\} + \log \det\left\{\mathbf{V}^\top \mathbf{\Sigma}^\ell \mathbf{V}\right\} + \frac{\rho}{2}\|\mathbf{D} \odot \mathbf{V} - \mathbf{Y}^k + \mathbf{U}^k\|_{\mathrm{F}}^2 \tag{36}$$

Eq. (36) is a standard smooth optimization problem over the Stiefel manifold, and it can be solved by standard techniques such as those in (Boumal, 2023). Newton and conjugate gradient methods for the Stiefel manifold are discussed in (Edelman et al., 1998). In our experiments, we use the conjugate gradient implementation in (Boumal et al., 2014).

**Solution for $\mathbf{Y}^{k+1}$.** The update for $\mathbf{Y}^{k+1}$ in (R1) reads as

$$\begin{aligned}
\mathbf{Y}^{k+1} &= \underset{\mathbf{Y} \in \mathbb{R}^{\ell \times h}}{\arg\min} \ \lambda \|\mathbf{Y}\|_1 + \frac{\rho}{2}\|\mathbf{D} \odot \mathbf{V}^{k+1} - \mathbf{Y} + \mathbf{U}^k\|_{\mathrm{F}}^2 = \\
&= \mathcal{S}_{\lambda/\rho}\left(\mathbf{D} \odot \mathbf{V}^{k+1} + \mathbf{U}^k\right);
\end{aligned} \tag{37}$$

where $\mathcal{S}_\delta(x) = \mathrm{sign}(x) \cdot \max(|x| - \delta, 0)$ is the element-wise soft-thresholding operator (Parikh et al., 2014).

**Stopping criteria.** The empirical convergence of LinSEPAL-ADMM is established according to primal and dual feasibility optimality conditions (Boyd et al., 2011). The primal residual, associated with the equality constraint in Eq. (P2), is

$$\mathbf{R}_p^{k+1} := \mathbf{Y}^{k+1} - \mathbf{D} \odot \mathbf{V}^{k+1}. \tag{38}$$

The dual residual, which can be obtained from the stationarity condition, is

$$\mathbf{R}_d^{k+1} := \rho \, \mathbf{D} \odot \left(\mathbf{Y}^{k+1} - \mathbf{Y}^k\right). \tag{39}$$

As $k \to \infty$, the norm of the primal and dual residuals should vanish. Hence, the stopping criterion can be set in terms of the norms

$$(i) \ d_p^{k+1} = \left\|\mathbf{R}_p^{k+1}\right\|_{\mathrm{F}} \quad \text{and} \quad (ii) \ d_d^{k+1} = \left\|\mathbf{R}_d^{k+1}\right\|_{\mathrm{F}}. \tag{40}$$

Specifically, given absolute and relative tolerance, namely $\tau^a$ and $\tau^r$ in $\mathbb{R}_+$, respectively, convergence in practice is established following Boyd et al. (2011) when

$$(i) \ d_p \le \tau^a \sqrt{\ell h} + \tau^r \max\left(\left\|\mathbf{Y}^{k+1}\right\|_{\mathrm{F}}, \left\|\mathbf{D} \circ \mathbf{V}^{k+1}\right\|_{\mathrm{F}}\right), \quad \text{and} \quad (ii) \ d_d \le \tau^a \sqrt{\ell h} + \tau^r \rho \left\|\mathbf{D} \circ \mathbf{U}^{k+1}\right\|_{\mathrm{F}}. \tag{41}$$

The LinSEPAL-ADMM algorithm is summarized in Algorithm 1.

---

**Algorithm 1** LinSEPAL-ADMM

---

1: **Input:** $\boldsymbol{\Sigma}^\ell$, $\boldsymbol{\Sigma}^h$, $\mathbf{D}$, $\lambda$, $\rho$, $\tau^a$, $\tau^r$
2: Initialize: $\mathbf{V}^0 \in \operatorname{St}(\ell, h)$, $\mathbf{Y}^0 \in \mathbb{R}^{\ell \times h}$, $\mathbf{U}^0 \leftarrow \mathbf{D} \odot \mathbf{V}^0 - \mathbf{Y}^0$
3: **repeat**
4:     $\mathbf{V}^{k+1} \leftarrow$ Solve Eq. (36) via an off-the-shelf method for smooth Riemannian problems
5:     $\mathbf{Y}^{k+1} \leftarrow \mathcal{S}_{\lambda/\rho}\left(\mathbf{D} \odot \mathbf{V}^{k+1} + \mathbf{U}^k\right)$
6:     $\mathbf{U}^{k+1} \leftarrow \mathbf{U}^k + \mathbf{D} \odot \mathbf{V}^{k+1} - \mathbf{Y}^{k+1}$
7: **until** Eq. (41) is satisfied
8: **Output:** $\mathbf{V}$, $\mathbf{Y}$, $\mathbf{U}$

---

## I. LinSEPAL-PG

This method is based upon the manifold proximal gradient (Chen et al., 2020) framework, which generalizes the *proximal gradient* framework defined in the Euclidean space to the Stiefel manifold. Following Chen et al. (2020), denoting by $\mathbf{V}^k$ the iterate at the step $k$, the updates recursion for solving (5) reads as

$$\mathbf{G}^k = \underset{\mathbf{G} \in T_{\mathbf{V}^k} \operatorname{St}(\ell, h)}{\arg\min} \quad \langle \nabla f\left(\mathbf{V}^k\right), \mathbf{G} \rangle + \frac{1}{2\rho}\|\mathbf{G}\|_F^2 + \lambda \|\mathbf{D} \odot \left(\mathbf{V}^k + \mathbf{G}\right)\|_1,$$
$$\mathbf{V}^{k+1} = \mathrm{R}_{\mathbf{V}^k}\left(\mathbf{G}^k\right). \tag{R2}$$

In (R2), the first update is the proximal mapping providing a proximal gradient direction $\mathbf{G}^k$ onto the tangent space to the Stiefel manifold, using the first-order approximation of the objective around the $k$-th estimate. The second is the update for $\mathbf{V}^{k+1}$, which exploits the canonical retraction (cf. Eq. (23)) technique for projecting back $\mathbf{V}^k + \mathbf{G}^k$ from the tangent space to the manifold. Global convergence of the ManPG method has been established in Chen et al. (2020).

**Solution for $\mathbf{G}^k$.** Chen et al. (2020) shows that the first update can be efficiently solved using the regularized semi-smooth Newton method in Xiao et al. (2018). Specifically, according to Eq. (21), the feasible set $T_{\mathbf{V}^k} \operatorname{St}(\ell, h)$ translates into a linear constraint. By defining $\mathcal{A}^k\left(\mathbf{G}\right) := \mathbf{G}^\top \mathbf{V}^k + \mathbf{V}^{k^\top} \mathbf{G}$, the update is

$$\mathbf{G}^k = \underset{\mathbf{G} \in \mathbb{R}^{\ell \times h}}{\arg\min} \quad \langle \nabla f\left(\mathbf{V}^k\right), \mathbf{G} \rangle + \frac{1}{2\rho}\|\mathbf{G}\|_F^2 + \lambda \|\mathbf{D} \odot \left(\mathbf{V}^k + \mathbf{G}\right)\|_1,$$
$$\text{subject to} \quad \mathcal{A}^k\left(\mathbf{G}\right) = \mathbf{0}_{h \times h}. \tag{42}$$

However, following the rationale in Si et al. (2024), we can force $\mathbf{G}^k \in T_{\mathbf{V}^k} \operatorname{St}(\ell, h)$ by exploiting the basis $\mathcal{B}_{\mathbf{V}^k}$ of the normal space to the manifold, namely $N_{\mathbf{V}^k} \operatorname{St}(\ell, h)$. To find such $\mathcal{B}_{\mathbf{V}^k}$, recall the explicit form of $N_{\mathbf{V}} \operatorname{St}(\ell, h)$ in Eq. (24).

The basis of $\operatorname{Sym}(h)$, having dimension $s = h(h+1)/2$, is

$$\mathcal{E} := \{\mathbf{E}_{ij} \in \{0, 1\}^{h \times h} \mid \mathbf{E}_{ij} \text{ has } e_{ij} = e_{ji} = 1, 0 \text{ elsewhere}, 1 \leq i \leq j \leq h\}. \tag{43}$$

It follows from Eqs. (24) and (43) that

$$\mathcal{B}_{\mathbf{V}^k} := \{\mathbf{B}_{ij}^k = \mathbf{V}^k \mathbf{E}_{ij}, 1 \leq i \leq j \leq h\}. \tag{44}$$

At this point, the membership to $T_{\mathbf{V}^k} \operatorname{St}(\ell, h)$ can be expressed as

$$\langle \mathbf{B}_{ij}^k, \mathbf{G} \rangle = 0, \quad \forall 1 \leq i \leq j \leq h. \tag{45}$$

Hence, (42) reads as

$$\mathbf{G}^k = \underset{\mathbf{G} \in \mathbb{R}^{\ell \times h}}{\arg\min} \quad \langle \nabla f\left(\mathbf{V}^k\right), \mathbf{G} \rangle + \frac{1}{2\rho}\|\mathbf{G}\|_F^2 + \lambda \|\mathbf{D} \odot \left(\mathbf{V}^k + \mathbf{G}\right)\|_1,$$
$$\text{subject to} \quad \langle \mathbf{B}_{ij}^k, \mathbf{G} \rangle = 0, \quad \forall 1 \leq i \leq j \leq h. \tag{46}$$

Consider $h\left(\mathbf{V}^k + \mathbf{G}\right) = \left\|\mathbf{D} \odot \left(\mathbf{V}^k + \mathbf{G}\right)\right\|_1$ and $\mathbb{R}^s \ni \boldsymbol{\mu} = [\mu_{11}, \mu_{12}, \ldots, \mu_{ij}, \ldots, \mu_{hh}]$, with $1 \leq i \leq j \leq h$. The Lagrangian for (46) is

$$L_\rho\left(\mathbf{G}, \boldsymbol{\mu}\right) = \left\langle \nabla f\left(\mathbf{V}^k\right), \mathbf{G} \right\rangle + \frac{1}{2\rho} \|\mathbf{G}\|_F^2 + \lambda h\left(\mathbf{V}^k + \mathbf{G}\right) - \sum_{1 \leq i \leq j \leq h} \mu_{ij} \langle \mathbf{B}_{ij}^k, \mathbf{G} \rangle. \tag{47}$$

Let us define now the matrix $\mathbb{R}^{s \times \ell h} \ni \mathbf{B}^k := [\text{vec}\left(\mathbf{B}_{11}^k\right), \text{vec}\left(\mathbf{B}_{12}^k\right), \ldots, \text{vec}\left(\mathbf{B}_{hh}^k\right)]^\top$, where $\text{vec}\left(\mathbf{B}_{ij}^k\right) \in \mathbb{R}^{\ell h}$. We can compactly express the $s$ equality constraints as

$$\mathbf{B}^k \text{vec}\left(\mathbf{G}\right) = \mathbf{0}_s. \tag{48}$$

Thus, the Karush-Kuhn-Tacker (KKT) conditions of Eq. (42) reads as

$$(i)\ \mathbf{0}_{l \times h} \in \partial_{\mathbf{G}} L_\rho\left(\mathbf{G}, \boldsymbol{\mu}\right), \quad \text{and} \quad (ii)\ \mathbf{B}^k \text{vec}\left(\mathbf{G}\right) = \mathbf{0}_s. \tag{49}$$

From the stationarity condition we get

$$\mathbf{0}_{l \times h} \in \mathbf{G} + \rho\left(\nabla f\left(\mathbf{V}^k\right) - \sum_{1 \leq i \leq j \leq h} \mu_{ij} \mathbf{B}_{ij}^k\right) + \lambda \rho\, \partial_{\mathbf{G}} h\left(\mathbf{V}^k + \mathbf{G}\right). \tag{50}$$

At this point, recalling the inclusion property of proximal operators, viz. $\mathbf{P} = \text{prox}_g(\mathbf{B}) \iff \mathbf{B} - \mathbf{P} \in \partial g(\mathbf{P})$, we have

$$\mathbf{0}_{l \times h} \in \underbrace{\mathbf{V}^k + \mathbf{G}}_{\mathbf{P}} - \underbrace{\left(\mathbf{V}^k - \rho\left(\nabla f\left(\mathbf{V}^k\right) - \sum_{1 \leq i \leq j \leq h} \mu_{ij} \mathbf{B}_{ij}^k\right)\right)}_{\mathbf{B}(\boldsymbol{\mu})} + \lambda \rho\, \partial_{\mathbf{G}} h \underbrace{\left(\mathbf{V}^k + \mathbf{G}\right)}_{\mathbf{P}}; \tag{51}$$

from which we get

$$\mathbf{G}(\boldsymbol{\mu}) = \text{prox}_{\lambda \rho\, h(\cdot)}\left(\mathbf{B}\left(\boldsymbol{\mu}\right)\right) - \mathbf{V}^k. \tag{52}$$

At this point, $\text{prox}_{\lambda \rho\, h(\cdot)}$ can be computed element-wise as

$$\text{prox}_{\lambda \rho\, h(\cdot)}\left(b_{ij}\left(\boldsymbol{\mu}\right)\right) = \begin{cases} b_{ij}(\boldsymbol{\mu}), & \text{if } d_{ij} = 0, \\ \mathcal{S}_{\lambda \rho}(b_{ij}(\boldsymbol{\mu})), & \text{otherwise}. \end{cases} \tag{53}$$

Substituting Eq. (52) into Eq. (49), we have

$$\mathbf{B}^k \text{vec}\left(\mathbf{G}(\boldsymbol{\mu})\right) = \mathbf{0}_s. \tag{54}$$

Here the $r$-th entry of $\text{vec}\left(\mathbf{G}(\boldsymbol{\mu})\right)$ corresponds to the entry of $\mathbf{G}(\boldsymbol{\mu})$ at row $u = (r-1) \bmod \ell + 1$, and column $v = \lfloor (r-1)/\ell \rfloor + 1$, $r \in [\ell h]$.

At this point, we can use the regularized semi-smooth Newton method (Xiao et al., 2018) to solve Eq. (54). Our target function is

$$F(\boldsymbol{\mu}) = \mathbf{B}^k \text{vec}\left(\mathbf{G}(\boldsymbol{\mu})\right) : \mathbb{R}^s \to \mathbb{R}^s. \tag{55}$$

By the chain rule of calculus, using Eq. (52), the generalized Jacobian matrix is

$$\begin{aligned} \mathbb{R}^{s \times s} \ni \mathbf{J} &= \frac{\partial F(\boldsymbol{\mu})}{\partial \text{vec}\left(\mathbf{G}(\boldsymbol{\mu})\right)} \cdot \frac{\partial \text{vec}\left(\mathbf{G}(\boldsymbol{\mu})\right)}{\partial \boldsymbol{\mu}} \\ &= \mathbf{B}^k \frac{\partial \text{prox}_{\lambda \rho\, h(\cdot)}\left(\text{vec}\left(\mathbf{B}(\boldsymbol{\mu})\right)\right)}{\partial \text{vec}\left(\mathbf{B}(\boldsymbol{\mu})\right)} \cdot \frac{\partial \text{vec}\left(\mathbf{B}(\boldsymbol{\mu})\right)}{\partial \boldsymbol{\mu}}. \end{aligned} \tag{56}$$

The proximal-related term is a diagonal matrix $\mathbf{M} \in \mathbb{R}^{\ell h \times \ell h}$, where

$$m_{rr} = \begin{cases} 1, & \text{if } \text{vec}\left(\mathbf{D}\right)_r = 0 \text{ or } \left(\text{vec}\left(\mathbf{D}\right)_r = 1 \text{ and } |b_r| - \lambda\rho > 0\right). \\ 0, & \text{otherwise.} \end{cases} \tag{57}$$

Additionally, starting from

$$
\begin{aligned}
b_r(\boldsymbol{\mu}) &= \text{vec}\left(\mathbf{V}^k - \rho\left(\nabla f(\mathbf{V}^k) - \sum_{1 \leq i \leq j \leq h} \mu_{ij}\left[\mathbf{B}_{ij}^k\right]_{uv}\right)\right) \\
&= \text{vec}\left(\mathbf{V}^k - \rho\left(\nabla f(\mathbf{V}^k) - \mathbf{b}_{uv}^{k\top}\boldsymbol{\mu}\right)\right).
\end{aligned}
\tag{58}
$$

Hence, we get

$$\mathbb{R}^s \ni \frac{\partial b_r(\boldsymbol{\mu})}{\partial\boldsymbol{\mu}} = \mathbf{b}_{uv}^k. \tag{59}$$

Consequently, starting from Eq. (56), using Eqs. (57) and (59), we finally have

$$\mathbf{J} = \mathbf{B}^k \mathbf{M} \mathbf{C}, \quad \text{with } \mathbb{R}^{\ell h \times s} \ni \mathbf{C} = \begin{pmatrix} \mathbf{b}_{11}^{k\top} \\ \mathbf{b}_{21}^{k\top} \\ \vdots \\ \mathbf{b}_{\ell h}^{k\top} \end{pmatrix}. \tag{60}$$

Following Xiao et al. (2018), denoting with $\nu^k = \alpha^k\|F^k\|_2$, $\alpha^k \in \mathbb{R}^+$, we define

$$r^k := \left(\mathbf{J}^{k-1} + \nu^{k-1}\mathbf{I}_s\right)\mathbf{d}^k + F^{k-1}. \tag{61}$$

At each iteration we want to find the step $\mathbf{d}^k$ by solving Eq. (61) inexactly, such that

$$\left\|r^k\right\|_2 \leq \tau \min\left(1, \alpha^{k-1}\left\|F^{k-1}\right\|_2\left\|\mathbf{d}^k\right\|_2\right), \quad \tau \in (0,1); \tag{62}$$

obtaining a trial point

$$\mathbf{u}^k = \boldsymbol{\mu}^{k-1} + \mathbf{d}^k. \tag{63}$$

Let $\beta^0 = \left\|F(\boldsymbol{\mu}^0)\right\|_2$ and $\gamma \in (0,1)$. If $\left\|F(\mathbf{u}^k)\right\|_2 \leq \gamma\beta^{k-1}$ then we set

$$\boldsymbol{\mu}^k = \mathbf{u}^k, \ \beta^k = \left\|F(\mathbf{u}^k)\right\|_2, \ \text{ and } \alpha^k = \alpha^{k-1}. \quad \text{[Newton step]} \tag{64}$$

Otherwise, let

$$\xi^k = \frac{-F(\mathbf{u}^k)^\top \mathbf{d}^k}{\|\mathbf{d}^k\|_2^2}. \tag{65}$$

Select $0 < \phi_1 \leq \phi_2 < 1$ and $1 < \psi_1 < \psi_2$. Hence, we make a safeguard step as follows

$$\boldsymbol{\mu}^k = \begin{cases} \mathbf{v}^k, & \text{if } \xi^k \geq \phi_1 \text{ and } \left\|F(\mathbf{v}^k)\right\|_2 \leq \left\|F(\boldsymbol{\mu}^{k-1})\right\|_2, \text{ [projection step]} \\ \mathbf{w}^k, & \text{if } \xi^k \geq \phi_1 \text{ and } \left\|F(\mathbf{v}^k)\right\|_2 > \left\|F(\boldsymbol{\mu}^{k-1})\right\|_2, \text{ [fixed-point step]} \\ \boldsymbol{\mu}^{k-1}, & \text{if } \xi^k < \phi_1, \text{ unsuccessful step} \end{cases} \tag{66}$$

where

$$\mathbf{v}^k = \boldsymbol{\mu}^{k-1} - \frac{F(\mathbf{u}^k)^\top(\boldsymbol{\mu}^{k-1} - \mathbf{u}^k)}{\|F(\mathbf{u}^k)\|_2}F(\mathbf{u}^k), \ \mathbf{w}^k = \boldsymbol{\mu}^{k-1} - \delta F(\boldsymbol{\mu}^k), \ \delta \in \left(0, \frac{1}{\omega}\right); \tag{67}$$

where $\omega \in (0, 1]$. Finally, denoting $\mathbb{R}^+ \ni \bar{\alpha} \approx 0$, the parameters $\beta^{k+1}$ and $\alpha^{k+1}$ are updated as

$$
\beta^k = \beta^{k-1}, \quad \alpha^k \in \begin{cases} (\bar{\alpha}, \alpha^{k-1}), & \text{if } \xi^k \geq \phi_2, \\ [\alpha^{k-1}, \psi_1 \alpha^{k-1}], & \text{if } \phi_1 \leq \xi^k < \phi_2, \\ (\psi_1 \alpha^{k-1}, \psi_2 \alpha^{k-1}], & \text{otherwise.} \end{cases} \tag{68}
$$

At this point, we set $\mathbf{G}^k = \mathbf{G}^k(\boldsymbol{\mu}^k)$ according to Eq. (52).

**Solution for $\mathbf{V}^{k+1}$.** Given $\mathbf{V}^k + \mathbf{G}^k \in T_{\mathbf{V}^k}\mathrm{St}(\ell, h)$, we have to project the point onto the manifold. This can be accomplished via the canonical retractions in Eq. (23). However, as suggested by Chen et al. (2020), our LinSEPAL-PG implementation performs an Armijo line-search procedure to determine the stepsize $a$. Hence, the update is

$$
\mathbf{V}^{k+1} = \mathrm{R}_{\mathbf{V}^k}^{\mathrm{QR}}\left(a\mathbf{G}^k\right). \tag{69}
$$

**Stopping criteria.** Empirical convergence of the LinSEPAL-PG algorithm is established either when a maximum number of iterations $K$ is reached, or when the $D_{\mathbf{V}^{k+1}}^{\mathrm{KL}}$ is below a certain threshold $\tau^{\mathrm{KL}} \approx 0$. The LinSEPAL-PG algorithm is summarized in Algorithm 2.

---

**Algorithm 2** LinSEPAL-PG

1: **Input:** $\boldsymbol{\Sigma}^\ell$, $\boldsymbol{\Sigma}^h$, $\mathbf{D}$, $\lambda$, $\rho$, $\gamma \in (0, 1)$, $\tau^{\mathrm{KL}}$, $K$
2: Initialize: $\mathbf{V}^0 \in \mathrm{St}(\ell, h)$, $\mathbf{Y}^0 \in \mathbb{R}^{\ell \times h}$, $\mathbf{U}^0 \in \mathbb{R}^{\ell \times h}$
3: **repeat**
4:    $\mathbf{G}^k \leftarrow$ Solve Eq. (46) via the regularized semi-smooth Newton method
5:    $a \leftarrow 1$
6:   **repeat**
7:      $a = \gamma a$
8:      $\bar{\mathbf{V}} = \mathrm{R}_{\mathbf{V}^k}^{\mathrm{QR}}\left(a\,\mathbf{G}^k\right)$
9:   **until** $D_{\bar{\mathbf{V}}}^{\mathrm{KL}} > D_{\mathbf{V}^k}^{\mathrm{KL}} - \frac{a\|\mathbf{G}^k\|_{\mathrm{F}}^2}{2\rho}$
10:   $\mathbf{V}^{k+1} \leftarrow \bar{\mathbf{V}}$
11: **until** $k > K$ or $D_{\mathbf{V}^{k+1}}^{\mathrm{KL}} < \tau^{\mathrm{KL}}$
12: **Output: V**

---

## J. CLinSEPAL

The problem we want to solve is:

**Problem 3.** *Given $\boldsymbol{\Sigma}^\ell \in \mathcal{S}_{++}^\ell$, $\boldsymbol{\Sigma}^h \in \mathcal{S}_{++}^h$, and $\mathbf{B} \in \{0,1\}^{\ell \times h}$, the linear constructive CA is given by the transpose of the product $\mathbf{B} \odot \mathbf{S} \odot \mathbf{V}$, where*

$$
\mathbf{V}^\star, \mathbf{S}^\star = \underset{\substack{\mathbf{V} \in \mathbb{R}^{\ell \times h} \\ \mathbf{S} \in [0,1]^{\ell \times h}}}{\arg\min} \quad f(\mathbf{V}, \mathbf{S});
$$

$$
\text{subject to } (i)\ \mathbf{B} \odot \mathbf{S} \odot \mathbf{V} \in \mathrm{St}(\ell, h), \tag{7}
$$

$$
(ii)\ (\mathbf{B} \odot \mathbf{S})^\top \in \mathrm{Sp}^\Delta(h, \ell),
$$

$$
(iii)\ \mathbf{1}_h - (\mathbf{B} \odot \mathbf{S})^\top \mathbf{1}_\ell \leq \mathbf{0}_h;
$$

*and*

$$
f(\mathbf{V}, \mathbf{S}) \coloneqq \mathrm{Tr}\left\{\left((\mathbf{B} \odot \mathbf{S} \odot \mathbf{V})^\top \boldsymbol{\Sigma}^\ell (\mathbf{B} \odot \mathbf{S} \odot \mathbf{V})\right)^{-1} \boldsymbol{\Sigma}^h\right\}
$$
$$
+ \log\det\left\{(\mathbf{B} \odot \mathbf{S} \odot \mathbf{V})^\top \boldsymbol{\Sigma}^\ell (\mathbf{B} \odot \mathbf{S} \odot \mathbf{V})\right\}. \tag{8}
$$

Prob. 3 makes it explicit that the abstraction morphism is given by three key ingredients: *(i)* the given partial, structural prior information represented by $\mathbf{B}$; *(ii)* the structural component $\mathbf{S}$ to be learned, such that the resulting causal abstraction is

constructive; and *(iii)* the abstraction coefficients in $\mathbf{V}$ determining the linear functional forms of the causal abstraction, which have to be learned as well. Specifically, "partial" means that some rows of $\mathbf{B}$ have more than one entry equal to one.

Unfortunately, Prob. 3 is nonconvex because of the objective function and the Stiefel manifold. Additionally, in this case, the CA results in a bilinear form $\mathbf{B} \odot \mathbf{S} \odot \mathbf{V}$, which is not jointly convex in $\mathbf{S}$ and $\mathbf{V}$. Consequently, the constraint $\mathbf{B} \odot \mathbf{S} \odot \mathbf{V} \in \mathrm{St}(\ell, h)$ has to be carefully handled.

Regarding the nonconvexity of the objective in Eq. (8), we proceed by leveraging its smoothness. Specifically, we have the following result.

**Corollary J.1.** *The function $f(\mathbf{V}, \mathbf{S})$ in Eq. (8) is smooth. Additionally, define $\mathbf{A} := (\mathbf{B} \odot \mathbf{S} \odot \mathbf{V})$ and $\widetilde{\mathbf{A}} := \left(\mathbf{A}^\top \mathbf{\Sigma}^\ell \mathbf{A}\right)^{-1}$. The partial derivatives w.r.t. $\mathbf{V}$ and $\mathbf{S}$ are*

$$\nabla_{\mathbf{V}} f = 2 \left(\mathbf{B} \odot \mathbf{S}\right) \odot \left(\left(\mathbf{\Sigma}^\ell \mathbf{A} \widetilde{\mathbf{A}}\right) \left(\mathbf{I}_h - \mathbf{\Sigma}^h \widetilde{\mathbf{A}}\right)\right), \tag{70}$$

*and*

$$\nabla_{\mathbf{S}} f = 2 \left(\mathbf{B} \odot \mathbf{V}\right) \odot \left(\left(\mathbf{\Sigma}^\ell \mathbf{A} \widetilde{\mathbf{A}}\right) \left(\mathbf{I}_h - \mathbf{\Sigma}^h \widetilde{\mathbf{A}}\right)\right). \tag{71}$$

*Proof.* Smoothness directly follows from Proposition 5.1 by defining $\mathbf{A} = (\mathbf{B} \odot \mathbf{S} \odot \mathbf{V})$, which is constrained to $\mathrm{St}(\ell, h)$ as given in Eq. (7). The partial derivatives in Eqs. (70) and (71) follow from the application of Eq. (34), together with the chain rule for derivatives. $\qquad \square$

At this point, we leverage Corollary J.1 to provide a solution which combines ADMM (Boyd et al., 2011) and SCA (Nedić et al., 2018). Specifically, ADMM is suitable to isolate and consequently tackle the nonconvexity in different subproblems. To manage the bilinear form within the first constraint in Eq. (7), we introduce two splitting variables, namely $\mathbf{Y}_1$ and $\mathbf{Y}_2$ in $\mathrm{St}(\ell, h)$, and the corresponding equality constraints

$$\mathbf{Y}_1 - \mathbf{B} \odot \mathbf{S}^k \odot \mathbf{V} = 0_{\ell \times h} \quad \text{and} \quad \mathbf{Y}_2 - \mathbf{B} \odot \mathbf{V}^{k+1} \odot \mathbf{S} = 0_{\ell \times h}, \text{ respectively.} \tag{72}$$

In this way, given the solution at iteration $k$ within the ADMM framework, we optimize separately over $\mathbf{V}$ and $\mathbf{S}$ while always tracking $\mathrm{St}(\ell, h)$. Please notice that we use $\mathbf{V}^{k+1}$ since when optimizing over $\mathbf{S}$, $\mathbf{V}$ has already been updated. The rationale behind the usage of the splitting variable for handling the Stiefel manifold is the same as the *splitting of orthogonality constraints* method (SOC, Lai & Osher, 2014). Additionally, to handle $(\mathbf{B} \odot \mathbf{S})^\top \in \mathrm{Sp}^\Delta(h, \ell)$, we introduce another splitting variable $\mathbf{X} \in \mathrm{Sp}^\Delta(h, \ell)$, and the corresponding equality constraint $\mathbf{X} - (\mathbf{B} \odot \mathbf{S})^\top = \mathbf{0}_{h \times \ell}$. Thus, starting from Eq. (7), we get the following equivalent minimization problem

$$\mathbf{V}^\star, \mathbf{S}^\star, \mathbf{Y}_1^\star, \mathbf{Y}_2^\star, \mathbf{X}^\star = \underset{\substack{\mathbf{V} \in \mathbb{R}^{\ell \times h} \\ \mathbf{S} \in [0,1]^{\ell \times h} \\ \mathbf{Y}_1 \in \mathrm{St}(\ell, h) \\ \mathbf{Y}_2 \in \mathrm{St}(\ell, h) \\ \mathbf{X} \in \mathrm{Sp}^\Delta(h, \ell)}}{\arg\min} \quad f(\mathbf{V}, \mathbf{S});$$

$$\text{subject to} \quad \mathbf{Y}_1 - \mathbf{B} \odot \mathbf{S}^k \odot \mathbf{V} = \mathbf{0}_{\ell \times h},$$
$$\mathbf{Y}_2 - \mathbf{B} \odot \mathbf{V}^{k+1} \odot \mathbf{S} = \mathbf{0}_{\ell \times h}, \tag{73}$$
$$\mathbf{X} - (\mathbf{B} \odot \mathbf{S})^\top = \mathbf{0}_{h \times \ell},$$
$$\mathbf{1}_h - (\mathbf{B} \odot \mathbf{S})^\top \mathbf{1}_\ell \leq \mathbf{0}_h.$$

Starting from Eq. (73), considering the penalty $\rho \in \mathbb{R}_+$, we introduce the scaled dual variables $\mathbf{U}_1$ and $\mathbf{U}_2$ in $\mathbb{R}^{\ell \times h}$; and $\mathbf{W} \in \mathbb{R}^{h \times \ell}$, and write the scaled augmented Lagrangian

$$L_\rho \left(\mathbf{V}, \mathbf{S}, \mathbf{Y}_1, \mathbf{Y}_2, \mathbf{X}, \mathbf{U}_1, \mathbf{U}_2, \mathbf{W}\right) = f(\mathbf{V}, \mathbf{S}) + \frac{\rho}{2} \left\|\mathbf{B} \odot \mathbf{S}^k \odot \mathbf{V} - \mathbf{Y}_1 + \mathbf{U}_1\right\|_{\mathrm{F}}^2 +$$
$$+ \frac{\rho}{2} \left\|\mathbf{B} \odot \mathbf{V}^{k+1} \odot \mathbf{S} - \mathbf{Y}_2 + \mathbf{U}_2\right\|_{\mathrm{F}}^2 + \frac{\rho}{2} \left\|(\mathbf{B} \odot \mathbf{S})^\top - \mathbf{X} + \mathbf{W}\right\|_{\mathrm{F}}^2. \tag{74}$$

Now, we can apply ADMM iterative procedure, getting the recursion for updating the primal and scaled dual variables. In detail, denote by $k \in \mathbb{N}$ the current iteration. We have

$$
\begin{aligned}
\mathbf{V}^{k+1} &= \underset{\mathbf{V} \in \mathbb{R}^{\ell \times h}}{\arg\min} \, L_\rho \left( \mathbf{V}, \mathbf{S}^k, \mathbf{Y}_1^k, \mathbf{Y}_2^k, \mathbf{X}^k, \mathbf{U}_1^k, \mathbf{U}_2^k, \mathbf{W}^k \right) \, ; \\
\mathbf{S}^{k+1} &= \underset{\mathbf{S} \in [0,1]^{\ell \times h}}{\arg\min} \, L_\rho \left( \mathbf{V}^{k+1}, \mathbf{S}, \mathbf{Y}_1^k, \mathbf{Y}_2^k, \mathbf{X}^k, \mathbf{U}_1^k, \mathbf{U}_2^k, \mathbf{W}^k \right) \, , \\
&\quad \text{subject to} \quad \mathbf{1}_h - \left( \mathbf{B} \odot \mathbf{S} \right)^\top \mathbf{1}_\ell \leq \mathbf{0}_h \, ; \\
\mathbf{Y}_1^{k+1} &= \underset{\mathbf{Y}_1 \in \mathrm{St}(\ell,h)}{\arg\min} \, L_\rho \left( \mathbf{V}^{k+1}, \mathbf{S}^{k+1}, \mathbf{Y}_1, \mathbf{Y}_2^k, \mathbf{X}^k, \mathbf{U}_1^k, \mathbf{U}_2^k, \mathbf{W}^k \right) \, ; \\
\mathbf{Y}_2^{k+1} &= \underset{\mathbf{Y}_2 \in \mathrm{St}(\ell,h)}{\arg\min} \, L_\rho \left( \mathbf{V}^{k+1}, \mathbf{S}^{k+1}, \mathbf{Y}_1^{k+1}, \mathbf{Y}_2, \mathbf{X}^k, \mathbf{U}_1^k, \mathbf{U}_2^k, \mathbf{W}^k \right) \, ; \\
\mathbf{X}^{k+1} &= \underset{\mathbf{X} \in \mathrm{Sp}^\Delta(h,\ell)}{\arg\min} \, L_\rho \left( \mathbf{V}^{k+1}, \mathbf{S}^{k+1}, \mathbf{Y}_1^{k+1}, \mathbf{Y}_2^{k+1}, \mathbf{X}, \mathbf{U}_1^k, \mathbf{U}_2^k, \mathbf{W}^k \right) \, ; \\
\mathbf{U}_1^{k+1} &= \mathbf{U}_1^k + \left( \mathbf{B} \odot \mathbf{S}^k \odot \mathbf{V}^{k+1} - \mathbf{Y}_1^{k+1} \right) \, ; \\
\mathbf{U}_2^{k+1} &= \mathbf{U}_2^k + \left( \mathbf{B} \odot \mathbf{V}^{k+1} \odot \mathbf{S}^{k+1} - \mathbf{Y}_2^{k+1} \right) \, ; \\
\mathbf{W}^{k+1} &= \mathbf{W}^k + \left( \mathbf{B} \odot \mathbf{S}^{k+1} \right)^\top - \mathbf{X}^{k+1} \, .
\end{aligned}
\tag{75}
$$

Similarly to SOC, we isolate the objective nonconvexity into the first and second (nonconvex) subproblems; and the nonconvexity of the manifold into the third and fourth (nonconvex) ones. Notably, the first and second subproblems can be managed through SCA. Additionally, the third and fourth nonconvex subproblems admit closed-form solutions since they boil down to the *closest orthogonal approximation problems* (Fan & Hoffman, 1955; Higham, 1986). Thus, the latter nonconvexity is somehow resolved. Finally, we solve the subproblem for $\mathbf{X}^{k+1}$ in closed form as well.

### J.1. Update for $\mathbf{V}^{k+1}$

Starting from Eqs. (74) and (75), the subproblem we have to solve is

$$
\mathbf{V}^{k+1} = \underset{\mathbf{V} \in \mathbb{R}^{\ell \times h}}{\arg\min} \quad f(\mathbf{V}, \mathbf{S}^k) + \frac{\rho}{2} \left\| \mathbf{B} \odot \mathbf{S}^k \odot \mathbf{V} - \mathbf{Y}_1^k + \mathbf{U}_1^k \right\|_\mathrm{F}^2 \, .
\tag{76}
$$

Eq. (76) is nonconvex due to the inherent nonconvexity of $f(\mathbf{V}, \mathbf{S}^k)$. However, the latter function is smooth and differentiable w.r.t. $\mathbf{V}$, as given in Corollary J.1. Hence, we apply the SCA framework. In detail, denote by $q$ the SCA iteration and set $\mathbf{V}^0 = \mathbf{V}^k$ for $q = 0$. We derive a strongly convex surrogate $\widetilde{f}(\mathbf{V}; \mathbf{V}^q, \mathbf{S}^k)$ around the point $\mathbf{V}^q$ – i.e., the solution at the iterate $q$ – exploiting Eq. (70):

$$
\widetilde{f}(\mathbf{V}; \mathbf{V}^q, \mathbf{S}^k) \coloneqq \mathrm{Tr}\left\{ \nabla_\mathbf{V} f|_{(\mathbf{V}^q, \mathbf{S}^k)}^\top \left( \mathbf{V} - \mathbf{V}^q \right) \right\} + \frac{\tau}{2} \left\| \mathbf{V} - \mathbf{V}^q \right\|_\mathrm{F}^2 \, .
\tag{77}
$$

It is immediate to check that Eq. (77) is a proper surrogate satisfying the stationarity condition $\nabla_\mathbf{V} f|_{\mathbf{V}^q} = \nabla_\mathbf{V} \widetilde{f}|_{\mathbf{V}^q}$.

Therefore, at each SCA iteration $q$, we solve a strongly convex problem in closed-form and then apply the usual smoothing operation by using a diminishing stepsize $\gamma^q \in \mathbb{R}_+$. Specifically,

$$
\begin{aligned}
\mathbf{V}^{q+1} &= \underset{\mathbf{V} \in \mathbb{R}^{\ell \times h}}{\arg\min} \quad \widetilde{f}(\mathbf{V}; \mathbf{V}^q, \mathbf{S}^k) + \frac{\rho}{2} \left\| \mathbf{B} \odot \mathbf{S}^k \odot \mathbf{V} - \mathbf{Y}_1^k + \mathbf{U}_1^k \right\|_\mathrm{F}^2 \, , \quad \text{(Strongly convex problem)} \\
\mathbf{V}^{q+1} &= \mathbf{V}^q + \gamma^k \left( \mathbf{V}^{q+1} - \mathbf{V}^q \right) \, . \quad \text{(Smoothing)}
\end{aligned}
\tag{78}
$$

The solution of the strongly-convex problem is given element-wise in Lemma J.2.

**Lemma J.2.** *The update for* $\mathbf{V}^{q+1}$ *can be computed element-wise as*

$$
v_{ij}^{q+1} = \frac{1}{\tau + b_{ij} s_{ij}^{k2}} \left( \rho \, b_{ij} s_{ij}^k y_{1_{ij}}^k - \rho \, b_{ij} s_{ij}^k u_{1_{ij}}^k + \tau \, v_{ij}^q - \left[ \nabla_\mathbf{V} f|_{(\mathbf{V}^q, \mathbf{S}^k)} \right]_{ij} \right) .
\tag{79}
$$

*Proof.* The proof follows by imposing the stationarity condition

$$\mathbf{0}_{\ell \times h} = \nabla_{\mathbf{V}} f|_{(\mathbf{V}^q, \mathbf{S}^k)} + \tau \left( \mathbf{V} - \mathbf{V}^q \right) + \rho \, \mathbf{B} \odot \mathbf{S}^k \odot \left( \mathbf{B} \odot \mathbf{S}^k \odot \mathbf{V} - \mathbf{Y}_1^k + \mathbf{U}_1^k \right), \tag{80}$$

and solving for $\mathbf{V}$. □

Additionally, the diminishing stepsize $\gamma^k$ has to satisfy the classical stochastic approximation conditions (Nedić et al., 2018),

$$(i) \sum_{q=1}^{\infty} \gamma^q = \infty \quad \text{and} \quad (ii) \sum_{q=1}^{\infty} (\gamma^q)^2 < \infty. \tag{81}$$

In our experiments, we use the decaying rule

$$\gamma^{q+1} = \gamma^q \left( 1 - \varepsilon \, \gamma^q \right), \quad \varepsilon \in (0, 1). \tag{82}$$

The SCA framework is guaranteed to converge to stationary points of the original nonconvex problem in Eq. (76) (Nedić et al., 2018). Accordingly, we establish convergence for the update when

$$\left\| \mathbf{V}^{q+1} - \mathbf{V}^q \right\|_{\mathrm{F}} \leq \tau^c, \quad \tau^c \approx 0; \tag{83}$$

and set $\mathbf{V}^{k+1} = \mathbf{V}^{q+1}$.

## J.2. Update for $\mathbf{S}^{k+1}$

Starting from Eqs. (74) and (75), the subproblem we have to solve is

$$\mathbf{S}^{k+1} = \underset{\mathbf{S} \in [0,1]^{\ell \times h}}{\arg\min} \, f(\mathbf{V}^k, \mathbf{S}) + \frac{\rho}{2} \left\| \mathbf{B} \odot \mathbf{V}^{k+1} \odot \mathbf{S} - \mathbf{Y}_2^k + \mathbf{U}_2^k \right\|_{\mathrm{F}}^2 + \frac{\rho}{2} \left\| (\mathbf{B} \odot \mathbf{S})^\top - \mathbf{X}^k + \mathbf{W}^k \right\|_{\mathrm{F}}^2,$$

$$\text{subject to} \quad \mathbf{1}_h - (\mathbf{B} \odot \mathbf{S})^\top \mathbf{1}_\ell \leq \mathbf{0}_h. \tag{84}$$

The subproblem above is nonconvex and constrained. Similarly to App. J.1, we apply the SCA framework. Denote by $q$ the SCA iteration and set $\mathbf{S}^0 = \mathbf{S}^k$ for $q = 0$. Here, the strongly convex surrogate of $f(\mathbf{V}^{k+1}, \mathbf{S})$ reads as

$$\widetilde{f}(\mathbf{S}; \mathbf{V}^{k+1}, \mathbf{S}^q) \coloneqq \mathrm{Tr}\left\{ \nabla_{\mathbf{S}} f|_{(\mathbf{V}^{k+1}, \mathbf{S}^q)}^\top (\mathbf{S} - \mathbf{S}^q) \right\} + \frac{\tau}{2} \|\mathbf{S} - \mathbf{S}^q\|_{\mathrm{F}}^2, \tag{85}$$

which satisfies $\nabla_{\mathbf{S}} f|_{(\mathbf{V}^{k+1}, \mathbf{S}^q)} = \nabla_{\mathbf{S}} \widetilde{f}|_{(\mathbf{V}^{k+1}, \mathbf{S}^q)}$. At each SCA iteration $q$, we solve a constrained quadratic programming (QP) problem and apply the smoothing step by using the stepsize $\gamma^q \in \mathbb{R}_+$ complying with the conditions in Eq. (81). In detail, let $\mathrm{vec}\,(\mathbf{A})$ be the column-wise vectorization of a given matrix $\mathbf{A}$ and define

$$\mathbf{Q} = \tau \mathbf{I}_{\ell h} + \rho \, \mathrm{diag}\left( \left( \mathrm{vec}\,(\mathbf{B}) \odot \mathrm{vec}\,(\mathbf{V}^{k+1}) \right) \odot \left( \mathrm{vec}\,(\mathbf{B}) \odot \mathrm{vec}\,(\mathbf{V}^{k+1}) \right) \right) + \rho \, \mathrm{diag}\,(\mathrm{vec}\,(\mathbf{B}) \odot \mathrm{vec}\,(\mathbf{B})),$$

$$\mathbf{c} = \mathrm{vec}\,(\nabla_{\mathbf{S}} f|_{(\mathbf{V}^{k+1}, \mathbf{S}^q)}) - \tau \, \mathrm{vec}\,(\mathbf{S}^q) - \rho \, \mathrm{vec}\,(\mathbf{Y}_2^k - \mathbf{U}_2^k) \odot \mathrm{vec}\,(\mathbf{B}) \odot \mathrm{vec}\,(\mathbf{V}^{k+1}) - \rho \, \mathrm{vec}\,(\mathbf{B}) \odot \mathrm{vec}\left( \left( \mathbf{X}^k - \mathbf{W}^k \right)^\top \right).$$

$$\tag{86}$$

Additionally, recall that $\mathrm{vec}\,(\mathbf{A}\mathbf{C}) = \left( \mathbf{C}^\top \otimes \mathbf{I}_h \right) \mathrm{vec}\,(\mathbf{A})$, with $\mathbf{A} \in \mathbb{R}^{h \times \ell}$ and $\mathbf{C} \in \mathbb{R}^{\ell \times m}$. Hence, denoting with $\mathbf{K}^{\ell,h}$ the commutation matrix, the inequality constraint can be rewritten as

$$\mathbf{1}_h - \mathrm{vec}\left( (\mathbf{B} \odot \mathbf{S})^\top \mathbf{1}_\ell \right) = \mathbf{1}_h - \left( \mathbf{1}_\ell^\top \otimes \mathbf{I}_h \right) \mathrm{vec}\left( (\mathbf{B} \odot \mathbf{S})^\top \right)$$

$$= \mathbf{1}_h - \left( \mathbf{1}_\ell^\top \otimes \mathbf{I}_h \right) \mathbf{K}^{\ell,h} \mathrm{vec}\,(\mathbf{B} \odot \mathbf{S})$$

$$= \mathbf{1}_h - \left( \mathbf{1}_\ell^\top \otimes \mathbf{I}_h \right) \mathbf{K}^{\ell,h} \mathrm{vec}\,(\mathrm{diag}\,(\mathrm{vec}\,(\mathbf{B}))\,\mathrm{vec}\,(\mathbf{S})) \tag{87}$$

$$= \mathbf{1}_h - \underbrace{\left( \mathbf{1}_\ell^\top \otimes \mathbf{I}_h \right) \mathbf{K}^{\ell,h} \mathrm{diag}\,(\mathrm{vec}\,(\mathbf{B}))}_{\mathbf{G}} \mathrm{vec}\,(\mathbf{S}) \leq \mathbf{0}_h.$$

At this point, starting from Eq. (84) and exploiting Eqs. (86) and (87), we can pose the SCA recursion:

$$\mathrm{vec}\,(\mathbf{S})^{q+1} = \underset{\mathbf{S} \in [0,1]^{\ell \times h}}{\arg\min} \quad \frac{1}{2} \mathrm{vec}\,(\mathbf{S})^\top \mathbf{Q}\,\mathrm{vec}\,(\mathbf{S}) + \mathbf{c}^\top \mathrm{vec}\,(\mathbf{S}), \quad \text{(QP problem)}$$

$$\text{subject to} \quad \mathbf{1}_h - \mathbf{G}\,\mathrm{vec}\,(\mathbf{S}) \leq \mathbf{0}_h. \tag{88}$$

$$\mathrm{vec}\,(\mathbf{S})^{q+1} = \mathrm{vec}\,(\mathbf{S})^q + \gamma^k \left( \mathrm{vec}\,(\mathbf{S})^{q+1} - \mathrm{vec}\,(\mathbf{S})^q \right). \quad \text{(Smoothing)}$$

The QP problem in Eq. (88) can be solved through off-the-shelf quadratic programming solvers. In our experiments, we use the OSQP (Stellato et al., 2020) implementation available in `cvxpy` (Diamond & Boyd, 2016). Since the quadratic form involves a diagonal, positive definite matrix $\mathbf{Q}$, in case a solution exists in the feasible set determined by the inequality constraint, it is also unique. Regarding the smoothing step, $\gamma^q$ follows Eq. (82). Similarly to App. J.1, we determine convergence when

$$\left\| \operatorname{vec}(\mathbf{S})^{q+1} - \operatorname{vec}(\mathbf{S})^q \right\|_{\mathrm{F}} \leq \tau^c, \quad \tau^c \approx 0 \,; \tag{89}$$

and set $\mathbf{S}^{k+1} = \mathbf{S}^{q+1}$, where $\mathbf{S}^{q+1}$ is the reshaping of $\operatorname{vec}(\mathbf{S})^{q+1}$ in matrix form.

## J.3. Update for $\mathbf{Y}_1^{k+1}$ and $\mathbf{Y}_2^{k+1}$

Starting from Eqs. (74) and (75), the subproblem to solve is

$$\begin{aligned}
\mathbf{Y}_1^{k+1} &= \underset{\mathbf{Y}_1 \in \mathrm{St}(\ell, h)}{\arg\min} \quad \frac{\rho}{2} \left\| \mathbf{B} \odot \mathbf{S}^{k+1} \odot \mathbf{V}^{k+1} - \mathbf{Y}_1 + \mathbf{U}_1^k \right\|_{\mathrm{F}}^2 \\
&= \operatorname{prox}_{\mathrm{St}(\ell,h)} \left( \widetilde{\mathbf{Y}}_1 \right), \quad \text{with } \widetilde{\mathbf{Y}}_1 := \mathbf{B} \odot \mathbf{S}^{k+1} \odot \mathbf{V}^{k+1} + \mathbf{U}_1^k.
\end{aligned} \tag{90}$$

The evaluation of $\operatorname{prox}_{\mathrm{St}(\ell,h)}(\widetilde{\mathbf{Y}}_1)$ in Eq. (90) is equivalent to the (unique) solution of the closest orthogonal approximation problem (Fan & Hoffman, 1955; Higham, 1986). Specifically, it is equal to the $\mathbf{U}_{p_1}$ factor of the polar decomposition of the matrix $\widetilde{\mathbf{Y}}_1 = \mathbf{U}_{p_1} \mathbf{P}_{p_1}$, namely

$$\mathbf{Y}_1^{k+1} = \mathbf{U}_{p_1}. \tag{91}$$

Similarly, defining $\widetilde{\mathbf{Y}}_2 := \mathbf{B} \odot \mathbf{S}^{k+1} \odot \mathbf{V}^{k+1} + \mathbf{U}_2^k$ and considering the polar decomposition $\widetilde{\mathbf{Y}}_2 = \mathbf{U}_{p_2} \mathbf{P}_{p_2}$, we have

$$\mathbf{Y}_2^{k+1} = \mathbf{U}_{p_2}. \tag{92}$$

## J.4. Update for $\mathbf{X}^{k+1}$

Starting from Eqs. (74) and (75), the subproblem reads as

$$\begin{aligned}
\mathbf{X}^{k+1} &= \underset{\mathbf{X} \in \mathrm{Sp}^\Delta(h,\ell)}{\arg\min} \frac{\rho}{2} \left\| \left( \mathbf{B} \odot \mathbf{S}^{k+1} \right)^\top - \mathbf{X} + \mathbf{W}^k \right\|_{\mathrm{F}}^2 \\
&= \operatorname{prox}_{\mathrm{Sp}^\Delta(h,\ell)} \left( \left( \mathbf{B} \odot \mathbf{S}^{k+1} \right)^\top + \mathbf{W}^k \right).
\end{aligned} \tag{93}$$

The following result gives the solution.

**Lemma J.3.** *Consider*

$$\mathrm{Sp}^\Delta(h, \ell) := \left\{ \mathbf{A} \in \{0,1\}^{h \times \ell} \mid \|\mathbf{a}_j\|_2 = 1 \text{ and } \sum_{i=1}^h a_{ij} = 1, \ \forall j \in [\ell] \right\}; \tag{94}$$

*and* $\mathbf{A} \in \mathbb{R}^{h \times \ell}$. *The proximal operator*

$$\operatorname{prox}_{\mathrm{Sp}^\Delta(h,\ell)}(\mathbf{A}) := \underset{\mathbf{X} \in \mathbb{R}^{h \times \ell}}{\arg\min} \|\mathbf{A} - \mathbf{X}\|_{\mathrm{F}}, \tag{95}$$

*is the matrix* $\mathbf{X}^\star$ *such that*

$$\forall j \in [\ell], \ x_{ij}^\star = \begin{cases} 1, & \text{if } a_{ij} = \arg\min_i |a_{ij} - 1|, \\ 0, & \text{otherwise.} \end{cases} \tag{96}$$

*Proof.* To belong to $\mathrm{Sp}^\Delta(h, \ell)$, $\mathbf{X}^\star$ must have only a single nonzero entry equal to one for each column $j \in [\ell]$. Consequently, the objective in Eq. (95) is minimized by setting, for each column $j \in [\ell]$, $x_{ij}^\star = 1$ in correspondence of the element $a_{ij}$ whose absolute distance from one is minimum. □

## J.5. Stopping criteria

The empirical convergence of the proposed method is established according to primal and dual feasibility optimality conditions. In this case, the primal residuals associated with the equality constraints in Eq. (73) are

$$
\begin{aligned}
\mathbf{R}_{p,1}^{k+1} &:= \mathbf{Y}_1^{k+1} - \mathbf{B} \odot \mathbf{S}^k \odot \mathbf{V}^{k+1} \,; \\
\mathbf{R}_{p,2}^{k+1} &:= \mathbf{Y}_2^{k+1} - \mathbf{B} \odot \mathbf{V}^{k+1} \odot \mathbf{S}^{k+1} \,; \\
\mathbf{R}_{p,3}^{k+1} &:= \mathbf{X}^{k+1} - \left(\mathbf{B} \odot \mathbf{S}^{k+1}\right)^{\top} \,.
\end{aligned}
\tag{97}
$$

Additionally, the dual residuals obtained from the stationarity condition are

$$
\begin{aligned}
\mathbf{R}_{d,1}^{k+1} &:= \rho\, \mathbf{B} \odot \mathbf{S}^k \odot \left(\mathbf{Y}_1^{k+1} - \mathbf{Y}_1^k\right) \,; \\
\mathbf{R}_{d,2}^{k+1} &:= \rho\, \mathbf{B} \odot \mathbf{V}^{k+1} \odot \left(\mathbf{Y}_2^{k+1} - \mathbf{Y}_2^k\right) \,; \\
\mathbf{R}_{d,3}^{k+1} &:= \rho\, \mathbf{B} \odot \left(\mathbf{X}^{k+1} - \mathbf{X}^k\right)^{\top} \,.
\end{aligned}
\tag{98}
$$

Following (Boyd et al., 2011), denoting with $\tau^a$ and $\tau^r$ in $\mathbb{R}_+$ the absolute and relative tolerances, respectively, the stopping criteria to be satisfied for empirical convergence are

$$
\begin{aligned}
\left\|\mathbf{R}_{p,1}^{k+1}\right\|_{\mathrm{F}} &= d_{p,1}^{k+1} \le \tau^a \sqrt{\ell h} + \tau^r \max\left(\left\|\mathbf{Y}_1^{k+1}\right\|_{\mathrm{F}}, \left\|\mathbf{B} \odot \mathbf{S}^k \odot \mathbf{V}^{k+1}\right\|_{\mathrm{F}}\right), \\
\left\|\mathbf{R}_{p,2}^{k+1}\right\|_{\mathrm{F}} &= d_{p,2}^{k+1} \le \tau^a \sqrt{\ell h} + \tau^r \max\left(\left\|\mathbf{Y}_2^{k+1}\right\|_{\mathrm{F}}, \left\|\mathbf{B} \odot \mathbf{V}^{k+1} \odot \mathbf{S}^{k+1}\right\|_{\mathrm{F}}\right), \\
\left\|\mathbf{R}_{p,3}^{k+1}\right\|_{\mathrm{F}} &= d_{p,3}^{k+1} \le \tau^a \sqrt{\ell h} + \tau^r \max\left(\left\|\mathbf{X}^{k+1}\right\|_{\mathrm{F}}, \left\|\mathbf{B} \odot \mathbf{S}^{k+1}\right\|_{\mathrm{F}}\right), \\
\left\|\mathbf{R}_{d,1}^{k+1}\right\|_{\mathrm{F}} &= d_{d,1}^{k+1} \le \tau^a \sqrt{\ell h} + \tau^r \rho \left\|\mathbf{B} \odot \mathbf{S}^k \odot \mathbf{U}_1^{k+1}\right\|_{\mathrm{F}}, \\
\left\|\mathbf{R}_{d,2}^{k+1}\right\|_{\mathrm{F}} &= d_{d,2}^{k+1} \le \tau^a \sqrt{\ell h} + \tau^r \rho \left\|\mathbf{B} \odot \mathbf{V}^{k+1} \odot \mathbf{U}_2^{k+1}\right\|_{\mathrm{F}}, \\
\left\|\mathbf{R}_{d,3}^{k+1}\right\|_{\mathrm{F}} &= d_{d,3}^{k+1} \le \tau^a \sqrt{\ell h} + \tau^r \rho \left\|\mathbf{B}^{\top} \odot \mathbf{W}^{k+1}\right\|_{\mathrm{F}}.
\end{aligned}
\tag{99}
$$

The CLinSEPAL method is summarized in Algorithm 3

---

**Algorithm 3** CLinSEPAL

---

1: **Input:** $\boldsymbol{\Sigma}^{\ell}$, $\boldsymbol{\Sigma}^h$, $\mathbf{B}$, $\rho$, $\tau$, $\varepsilon$, $\tau^c$, $\tau^a$, $\tau^r$
2: Initialize: $\mathbf{V}^0 \in \mathbb{R}^{\ell \times h}$, $\mathbf{S}^0 = \mathbf{B}$, $\mathbf{Y}_1^0 \in \mathrm{St}(\ell, h)$, $\mathbf{Y}_2^0 \in \mathrm{St}(\ell, h)$, $\mathbf{X}^0 = \mathbf{B}^{\top}$, $\mathbf{U}_1^0 \leftarrow \mathbf{B} \odot \mathbf{S}^0 \odot \mathbf{V}^0 - \mathbf{Y}_1^0$, $\mathbf{U}_2^0 \leftarrow \mathbf{B} \odot \mathbf{S}^0 \odot \mathbf{V}^0 - \mathbf{Y}_2^0$, $\mathbf{W}^0 \leftarrow \left(\mathbf{B} \odot \mathbf{S}^0\right)^{\top} - \mathbf{X}^0$
3: **repeat**
4:     $\mathbf{V}^{k+1} \leftarrow$ Apply Eq. (78)
5:     $\mathbf{S}^{k+1} \leftarrow$ Apply Eq. (88)
6:     $\mathbf{Y}_1^{k+1} \leftarrow$ Eq. (91)
7:     $\mathbf{Y}_2^{k+1} \leftarrow$ Eq. (92)
8:     $\mathbf{U}_1^{k+1} \leftarrow \mathbf{U}_1^k + \mathbf{B} \odot \mathbf{S}^k \odot \mathbf{V}^{k+1} - \mathbf{Y}_1^{k+1}$
9:     $\mathbf{U}_2^{k+1} \leftarrow \mathbf{U}_2^k + \mathbf{B} \odot \mathbf{V}^{k+1} \odot \mathbf{S}^{k+1} - \mathbf{Y}_2^{k+1}$
10:    $\mathbf{W}^{k+1} \leftarrow \mathbf{W}^k + \left(\mathbf{B} \odot \mathbf{S}^{k+1}\right)^{\top} - \mathbf{X}^{k+1}$
11: **until** Eq. (99) is satisfied
12: **Output:** $\mathbf{V}$, $\mathbf{S}$, $\mathbf{Y}_1$, $\mathbf{Y}_2$, $\mathbf{X}$, $\mathbf{U}_1$, $\mathbf{U}_2$, $\mathbf{W}$

---

## J.6. Full prior case

Prob. 3 simplifies in case of full prior knowledge of $\mathbf{B}$. Indeed, it is not needed to learn $\mathbf{S}$ since $\mathbf{S} \equiv \mathbf{B}$. Accordingly, we get the following.

**Problem 4.** *Given $\boldsymbol{\Sigma}^\ell \in \mathcal{S}_{++}^\ell$, $\boldsymbol{\Sigma}^h \in \mathcal{S}_{++}^h$, and $\mathbf{B} \in \{0,1\}^{\ell \times h}$, the linear constructive CA is given by the transpose of the product $\mathbf{B} \odot \mathbf{V}$, where*

$$\mathbf{V}^\star = \underset{\mathbf{V} \in \mathbb{R}^{\ell \times h}}{\arg\min} \quad f(\mathbf{V}) \, ;$$
$$\text{subject to} \quad \mathbf{B} \odot \mathbf{V} \in \mathrm{St}(\ell, h) \, ; \tag{100}$$

*and*

$$f(\mathbf{V}) \coloneqq \mathrm{Tr}\left\{ \left( (\mathbf{B} \odot \mathbf{V})^\top \boldsymbol{\Sigma}^\ell (\mathbf{B} \odot \mathbf{V}) \right)^{-1} \boldsymbol{\Sigma}^h \right\} + \log\det\left\{ (\mathbf{B} \odot \mathbf{V})^\top \boldsymbol{\Sigma}^\ell (\mathbf{B} \odot \mathbf{V}) \right\}. \tag{101}$$

The solution can be obtained in a similar manner as for the partial prior knowledge case. Below, we report the mathematical derivation for completeness without further comments.

**Corollary J.4.** *The function $f(\mathbf{V})$ in Eq. (101) is smooth. Additionally, define $\mathbf{A} \coloneqq (\mathbf{B} \odot \mathbf{V})$ and $\widetilde{\mathbf{A}} \coloneqq \left( \mathbf{A}^\top \boldsymbol{\Sigma}^\ell \mathbf{A} \right)^{-1}$. The gradient is*

$$\nabla_{\mathbf{V}} f = 2 \mathbf{B} \odot \left( \left( \boldsymbol{\Sigma}^\ell \mathbf{A} \widetilde{\mathbf{A}} \right) \left( \mathbf{I}_h - \boldsymbol{\Sigma}^h \widetilde{\mathbf{A}} \right) \right) , \tag{102}$$

*Proof.* Smoothness directly follows from Proposition 5.1 by defining $\mathbf{A} \coloneqq (\mathbf{B} \odot \mathbf{V})$, which is constrained to $\mathrm{St}(\ell, h)$ as given in Eq. (100). The gradient in Eq. (102) follows from the application of Eq. (34), together with the chain rule for derivatives. $\square$

Starting from Eq. (100), we get the following equivalent minimization problem

$$\mathbf{V}^\star, \mathbf{Y}^\star = \underset{\substack{\mathbf{V} \in \mathbb{R}^{\ell \times h} \\ \mathbf{Y} \in \mathrm{St}(\ell, h)}}{\arg\min} \quad f(\mathbf{V}) \, ;$$
$$\text{subject to} \quad \mathbf{Y} - \mathbf{B} \odot \mathbf{V} = \mathbf{0}_{\ell \times h} \, . \tag{103}$$

Considering the scaled dual variable $\mathbf{U} \in \mathbb{R}^{\ell \times h}$, the scaled augmented Lagrangian is

$$L_\rho(\mathbf{V}, \mathbf{Y}, \mathbf{U}) = f(\mathbf{V}) + \frac{\rho}{2} \|\mathbf{B} \odot \mathbf{V} - \mathbf{Y} + \mathbf{U}\|_{\mathrm{F}}^2 . \tag{104}$$

The ADMM recursion is

$$\mathbf{V}^{k+1} = \underset{\mathbf{V} \in \mathbb{R}^{\ell \times h}}{\arg\min} L_\rho(\mathbf{V}, \mathbf{Y}^k, \mathbf{U}^k) \, ;$$
$$\mathbf{Y}^{k+1} = \underset{\mathbf{Y} \in \mathrm{St}(\ell, h)}{\arg\min} L_\rho(\mathbf{V}^{k+1}, \mathbf{Y}, \mathbf{U}^k) \, ; \tag{105}$$
$$\mathbf{U}^{k+1} = \mathbf{U}^k + \left( \mathbf{B} \odot \mathbf{V}^{k+1} - \mathbf{Y}^{k+1} \right) \, .$$

### J.6.1. Update for $\mathbf{V}^{k+1}$

Starting from Eqs. (104) and (105), the subproblem we have to solve is

$$\mathbf{V}^{k+1} = \underset{\mathbf{V} \in \mathbb{R}^{\ell \times h}}{\arg\min} \quad f(\mathbf{V}) + \frac{\rho}{2} \|\mathbf{B} \odot \mathbf{V} - \mathbf{Y}^k + \mathbf{U}^k\|_{\mathrm{F}}^2 . \tag{106}$$

Eq. (106) is nonconvex due to the inherent nonconvexity of $f(\mathbf{V})$. However, the latter function is smooth and differentiable w.r.t. $\mathbf{V}$, as given in Corollary J.4. Hence, we apply the SCA framework. In detail, denote by $q$ the SCA iteration and set $\mathbf{V}^0 = \mathbf{V}^k$ for $q = 0$. We derive a strongly convex surrogate $\widetilde{f}(\mathbf{V}; \mathbf{V}^q)$ around the point $\mathbf{V}^q$ – i.e., the solution at the iterate $q$ – exploiting Eq. (102),

$$\widetilde{f}(\mathbf{V}; \mathbf{V}^q) \coloneqq \mathrm{Tr}\left\{ \nabla_{\mathbf{V}} f|_{\mathbf{V}^q}^\top (\mathbf{V} - \mathbf{V}^q) \right\} + \frac{\tau}{2} \|\mathbf{V} - \mathbf{V}^q\|_{\mathrm{F}}^2 . \tag{107}$$

Therefore, at each SCA iteration $q$, we solve a strongly convex problem in closed-form and then apply the usual smoothing operation by using a diminishing stepsize $\gamma^q \in \mathbb{R}_+$ following Eq. (82) and satisfying Eq. (81). Specifically,

$$\mathbf{V}^{q+1} = \underset{\mathbf{V} \in \mathbb{R}^{\ell \times h}}{\arg\min} \quad \widetilde{f}(\mathbf{V}; \mathbf{V}^q) + \frac{\rho}{2} \|\mathbf{B} \odot \mathbf{V} - \mathbf{Y}^k + \mathbf{U}^k\|_{\mathrm{F}}^2 , \quad \text{(Strongly convex problem)}$$
$$\mathbf{V}^{q+1} = \mathbf{V}^q + \gamma^k \left( \mathbf{V}^{q+1} - \mathbf{V}^q \right) . \quad \text{(Smoothing)} \tag{108}$$

The solution of the strongly-convex problem is given element-wise in Lemma J.5.

**Lemma J.5.** *The update for $\mathbf{V}^{q+1}$ can be computed element-wise as*

$$v_{ij}^{q+1} = \frac{1}{\tau + b_{ij}}\left(\rho\, b_{ij} y_{ij}^k - \rho\, b_{ij} u_{ij}^k + \tau\, v_{ij}^q - \left[\nabla_{\mathbf{V}} f|_{\mathbf{V}^q}\right]_{ij}\right). \tag{109}$$

*Proof.* The proof follows by imposing the stationarity condition

$$\mathbf{0}_{\ell \times h} = \nabla_{\mathbf{V}} f|_{\mathbf{V}^q} + \tau\left(\mathbf{V} - \mathbf{V}^q\right) + \rho\, \mathbf{B} \odot \left(\mathbf{B} \odot \mathbf{V} - \mathbf{Y}^k + \mathbf{U}^k\right), \tag{110}$$

and solving for $\mathbf{V}$. $\qquad\square$

We establish convergence for the update when

$$\left\|\mathbf{V}^{q+1} - \mathbf{V}^q\right\|_{\mathrm{F}} \leq \tau^c, \quad \tau^c \approx 0\,; \tag{111}$$

and set $\mathbf{V}^{k+1} = \mathbf{V}^{q+1}$.

### J.6.2. UPDATE FOR $\mathbf{Y}^{k+1}$

Starting from Eqs. (104) and (105), the subproblem to solve is

$$\begin{aligned}
\mathbf{Y}^{k+1} &= \underset{\mathbf{Y} \in \mathrm{St}(\ell,h)}{\arg\min} \quad \frac{\rho}{2}\left\|\mathbf{B} \odot \mathbf{V}^{k+1} - \mathbf{Y} + \mathbf{U}^k\right\|_{\mathrm{F}}^2 \\
&= \mathrm{prox}_{\mathrm{St}(\ell,h)}(\widetilde{\mathbf{Y}}), \quad \text{with } \widetilde{\mathbf{Y}} := \mathbf{B} \odot \mathbf{V}^{k+1} + \mathbf{U}^k.
\end{aligned} \tag{112}$$

Denoting by $\mathbf{U}_p \mathbf{P}_p$ the polar decomposition of the matrix $\widetilde{\mathbf{Y}}$, the update is

$$\mathbf{Y}^{k+1} = \mathbf{U}_p. \tag{113}$$

### J.6.3. STOPPING CRITERIA

The empirical convergence of the proposed method is established according to primal and dual feasibility optimality conditions (Boyd et al., 2011). The primal residual, associated with the equality constraint in Eq. (103), is

$$\mathbf{R}_p^{k+1} := \mathbf{Y}^{k+1} - \mathbf{B} \odot \mathbf{V}^{k+1}. \tag{114}$$

The dual residual, which can be obtained from the stationarity condition, is

$$\mathbf{R}_d^{k+1} := \rho\, \mathbf{B} \odot \left(\mathbf{Y}^{k+1} - \mathbf{Y}^k\right). \tag{115}$$

As $k \to \infty$, the norm of the primal and dual residuals should vanish. Hence, the stopping criterion can be set in terms of the norms

$$(i)\ d_p^{k+1} = \left\|\mathbf{R}_p^{k+1}\right\|_{\mathrm{F}} \quad \text{and} \quad (ii)\ d_d^{k+1} = \left\|\mathbf{R}_d^{k+1}\right\|_{\mathrm{F}}. \tag{116}$$

Specifically, given absolute and relative tolerance, namely $\tau^a$ and $\tau^r$ in $\mathbb{R}_+$, respectively, convergence in practice is established following Boyd et al. (2011), when

$$(i)\ d_p^{k+1} \leq \tau^a \sqrt{\ell h} + \tau^r \max\left(\left\|\mathbf{Y}^{k+1}\right\|_{\mathrm{F}}, \left\|\mathbf{B} \odot \mathbf{V}^{k+1}\right\|_{\mathrm{F}}\right), \quad \text{and} \quad (ii)\ d_d^{k+1} \leq \tau^a \sqrt{\ell h} + \tau^r \rho \left\|\mathbf{B} \odot \mathbf{U}^{k+1}\right\|_{\mathrm{F}}. \tag{117}$$

The full prior version of CLinSEPAL is summarized in Algorithm 4.

---

**Algorithm 4** CLinSEPAL (full prior case)

---

1: **Input:** $\boldsymbol{\Sigma}^\ell, \boldsymbol{\Sigma}^h, \mathbf{B}, \rho, \tau, \varepsilon, \tau^c, \tau^a, \tau^r$
2: Initialize: $\mathbf{V}^0 \in \mathbb{R}^{\ell \times h}, \mathbf{Y}^0 \in \mathrm{St}(\ell, h), \mathbf{U}^0 \leftarrow \mathbf{B} \odot \mathbf{V}^0 - \mathbf{Y}^0$
3: **repeat**
4: $\quad \mathbf{V}^{k+1} \leftarrow$ Apply Eq. (108)
5: $\quad \mathbf{Y}^{k+1} \leftarrow$ Eq. (113)
6: $\quad \mathbf{U}^{k+1} \leftarrow \mathbf{U}^k + \mathbf{B} \odot \mathbf{V}^{k+1} - \mathbf{Y}^{k+1}$
7: **until** Eq. (117) is satisfied
8: **Output:** $\mathbf{V}, \mathbf{Y}, \mathbf{U}$

---

# K. Metrics and Hyper-parameters

This section provides the definition of the metrics monitored in our empirical assessment in Secs. 6 and 7. Additionally, we report the hyper-parameters configuration for Algorithms 1 to 3 used in the experiments.

**Metrics.** Denote by $\mathbf{V}^\star$ and $\widehat{\mathbf{V}}$ the ground-truth and the learned (transpose of the) linear CA, both being matrices in $\mathbb{R}^{\ell \times h}$. The metrics are defined as follows.

- Fraction of learned constructive morphisms: We define constructiveness as

  $$\text{constr} = (\text{number of rows with one nonzero entry})/\ell + (\text{number of columns with at least one nonzero entry})/h \,. \tag{118}$$

  Then, indicating by $S$ the number of experiments, the metric is given by the number of $\widehat{\mathbf{V}}$ leading to $\text{constr} = 1$ divided by $S$.

- KL divergence: Eq. (3) evaluated at $\widehat{\mathbf{V}}$;

- Frobenious absolute distance:

  $$\frac{\left\| |\mathbf{V}^\star| - \left| \widehat{\mathbf{V}} \right| \right\|_{\text{F}}}{\left\| |\mathbf{V}^\star| \right\|_{\text{F}}} \,; \tag{119}$$

- F1 score: Given

  - True positive rate $\text{tpr}$: (true positive, $\text{tp}$: number of predicted nonzero entries in $\widehat{\mathbf{V}}$ existing in $\mathbf{V}^\star$)/(number of nonzero entries in $\mathbf{V}^\star$),

  - False discovery rate $\text{fdr}$: (false positive, $\text{fp}$: number of predicted nonzero entries in $\widehat{\mathbf{V}}$ that do not exist in $\mathbf{V}^\star$)/($\text{tp} + \text{fp}$);

  the F1 results in the harmonic mean of $\text{tpr}$ and $(1 - \text{fdr})$.

**Hyper-parameters.**

- CLinSEPAL: $\rho = 1$, $\tau = 10^{-3}$, $\varepsilon = 0.1$ for the full prior case and $\varepsilon = 0.01$ for the partial prior case, $\tau^c = 10^{-3}$, $\tau^a = 10^{-4}$, $\tau^r = 10^{-4}$ . The same hyper-parameters were used in the experiments in Sec. 7;

- LinSEPAL-ADMM: $\rho = 1$, $\lambda = 1$, $\tau^a = 10^{-4}$, $\tau^r = 10^{-4}$;

- LinSEPAL-PG: $\lambda = 1$, $\rho = 1/\left(2\left\|\mathbf{\Sigma}^\ell\right\|_{\text{F}}^2\right)$, $\gamma = 0.5$, $\tau^{\text{KL}} = 10^{-4}$, $K = 1000$ .

## L. Additional Results on Synthetic Data

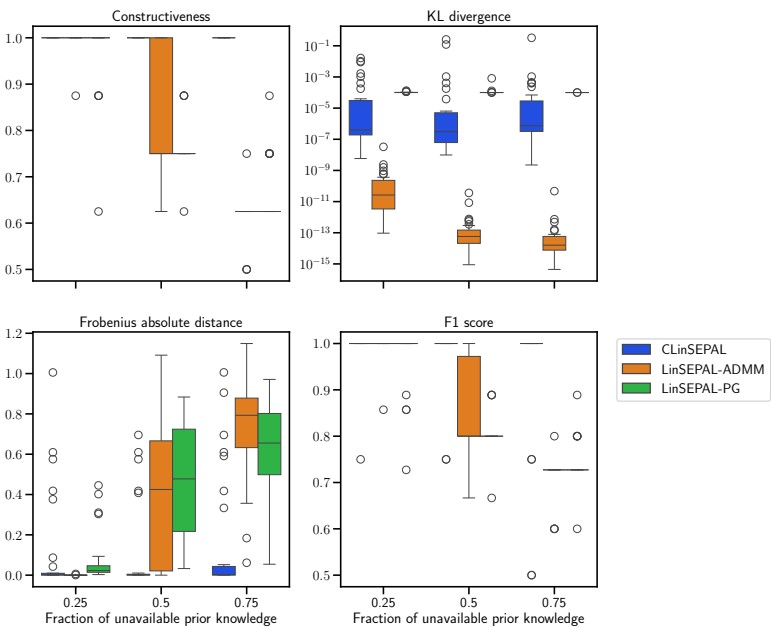

Figure 7: **The figure shows the results in the pp setting where the** $90\%$ **threshold on constructiveness was removed. Differently from Fig. 4, top left boxplots are for constructiveness values. The remaining plots are as in Fig. 4. From the figure we see that although LinSEPAL-ADMM and LinSEPAL-PG minimize the misalignment between the (abstracted) low- and high-level probability measures (top right), the quality of learned CAs is poor compared to that of learned CAs from CLinSEPAL (bottom plots).**

## M. Additional Material for the Causal Abstraction of Brain Networks

This section provides additional material about the full and partial prior applications of CLinSEPAL to brain data, given in Sec. 7. Specifically, Fig. 8 depicts the ground truth linear CA and the learned linear CA by CLinSEPAL for the full prior setting; whereas Fig. 10 the results for the partial prior setting. Regarding the partial prior setting, we also report in Fig. 9 the partial prior received as an input by CLinSEPAL for all the settings, and in Fig. 11 the monitored metrics to better understand the performance of CLinSEPAL with varying degree of uncertainty (low, medium, high), as discussed in Sec. 7. The color coding for the partial prior setting refers to the following classification, reported unaltered from (D'Acunto et al., 2024):

- Red for ROIs corresponding to cognitive functions, attention, emotion, and decision-making;
- Orange for those related to auditory processing, speech and language processing, and memory;
- Blue for those concerning memory formation and memory retrieval;
- Pink for those associated with sensory integration and somatosensory;
- Purple for the ROIs within the visual network and related to the visual memory;
- Green for those within the motor network;
- Yellow for those regarding the motor control and the posture.

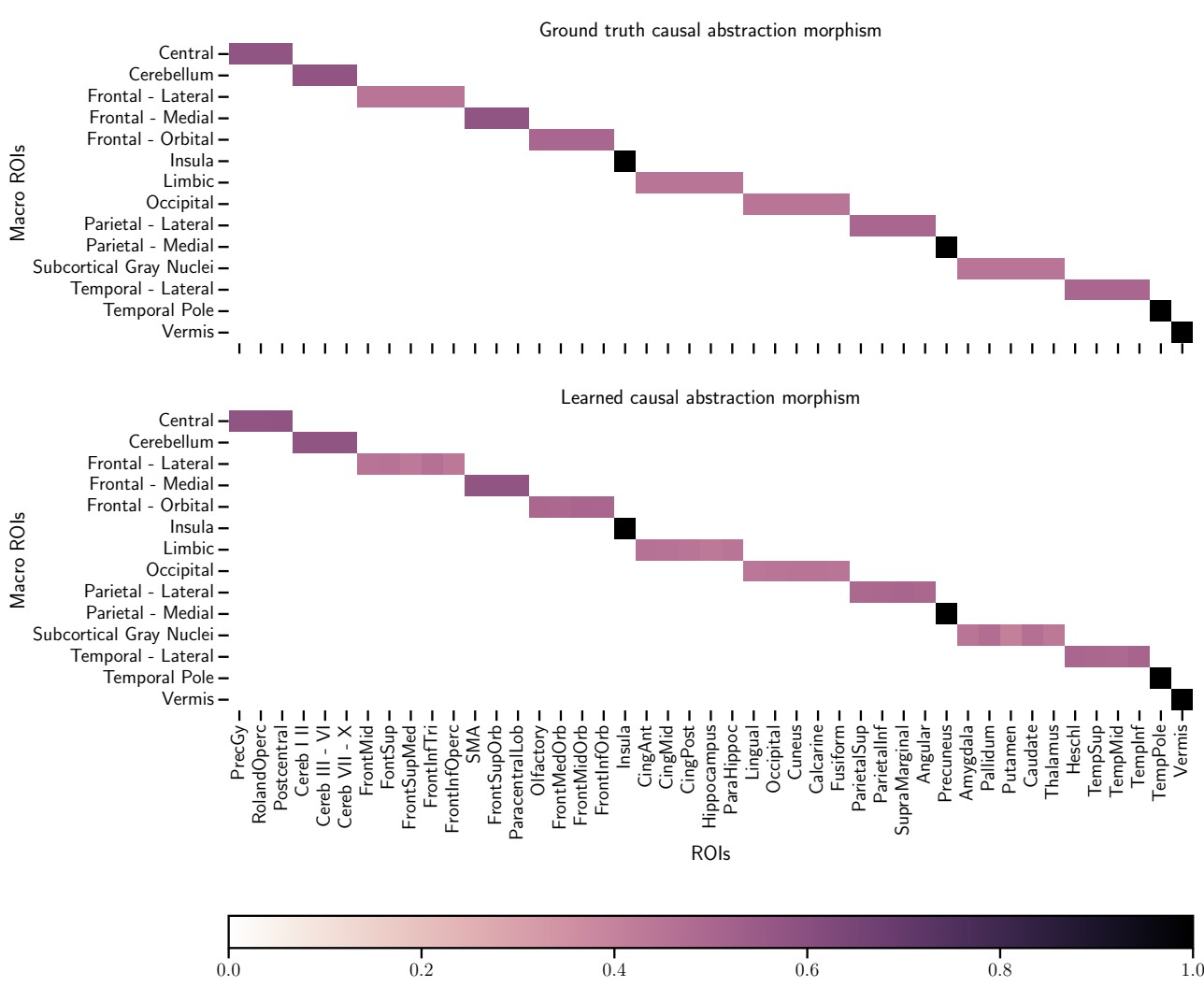

Figure 8: The figure shows (top) the ground truth linear CA and (bottom) the learned linear CA for the simulated full prior setting in Sec. 7.

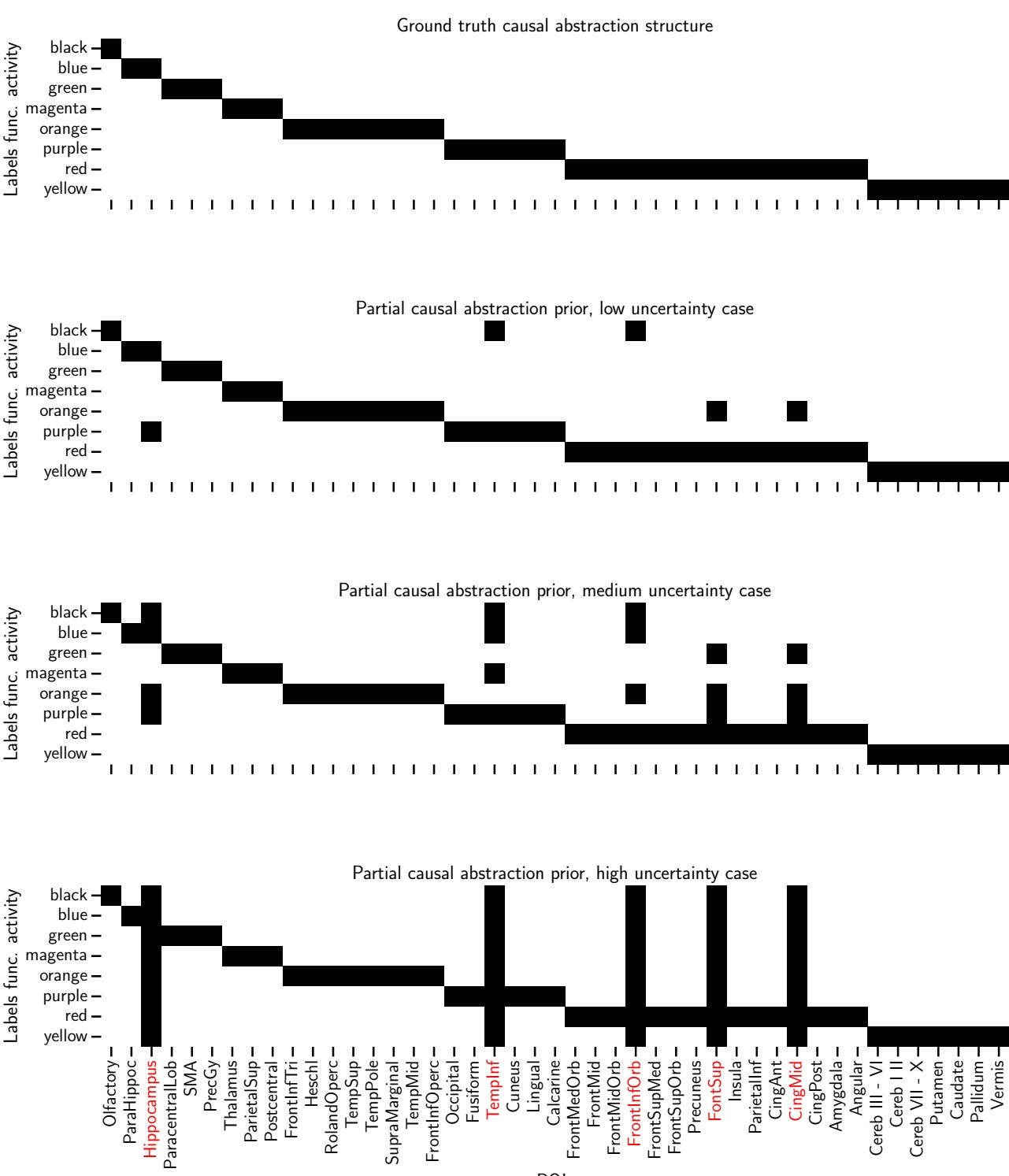

**Figure 9: Starting from the top, the figure shows *(i)* the ground truth structure for linear CA, and the partial prior provided as input to CLinSEPALlearned linear CA for the simulated partial prior in the *(ii)* low, *(iii)* medium, and *(iv)* high uncertainty settings discussed in Sec. 7.**

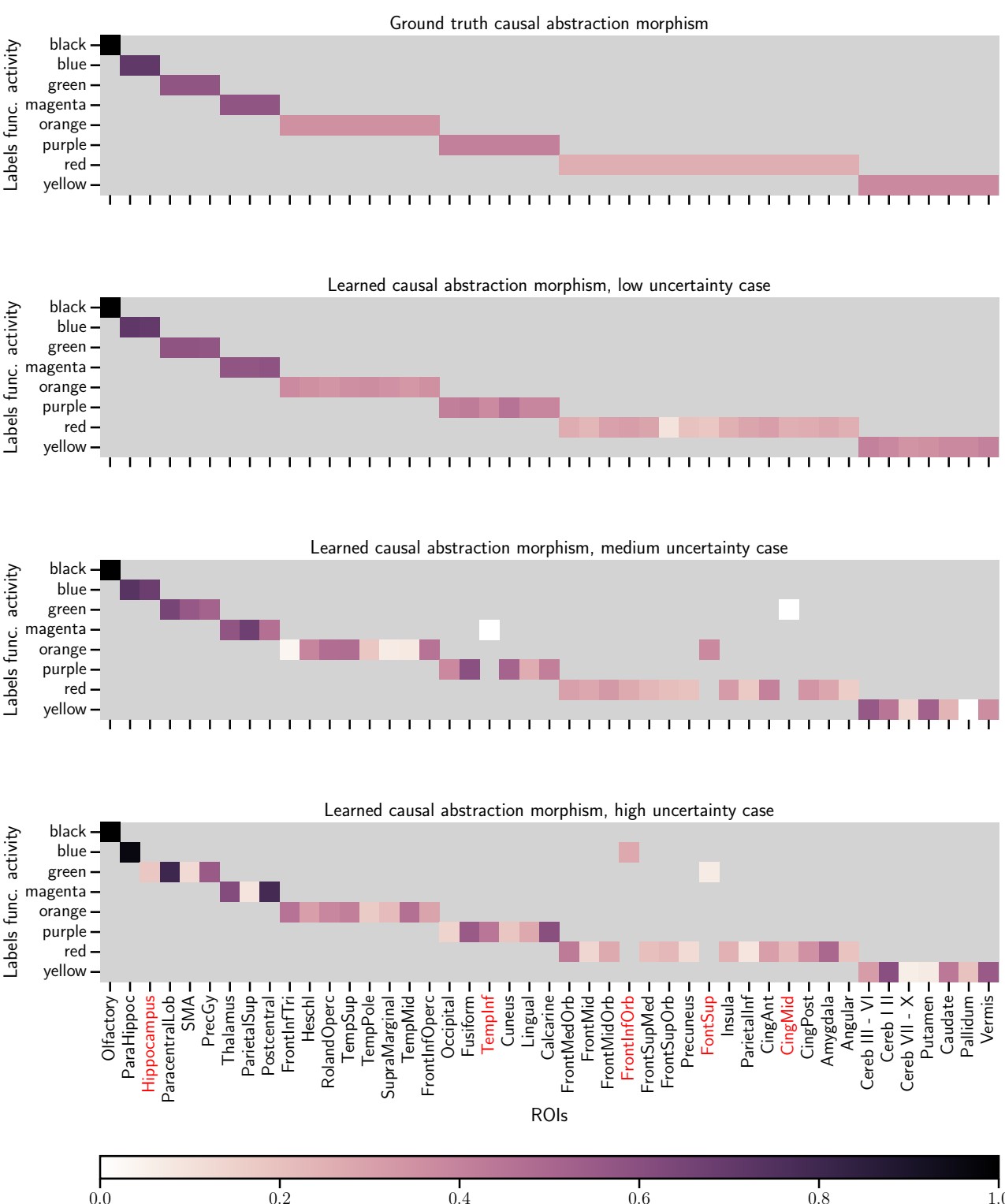

**Figure 10: Starting from the top, the figure shows** *(i)* **the ground truth linear CA, and the learned linear CA for the simulated partial prior setting with** *(ii)* **low,** *(iii)* **medium, and** *(iv)* **high uncertainty in Sec. 7.**

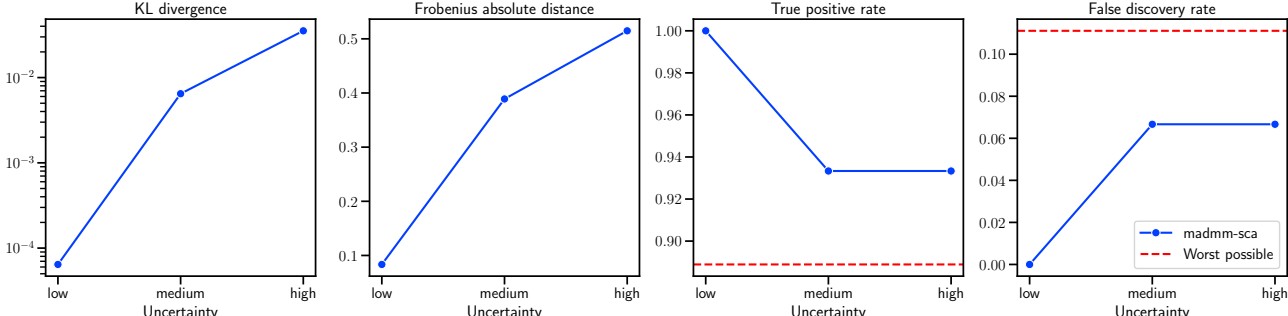

**Figure 11: Starting from the left, the figure provides the** *(i)* **the KL divergence evaluated at the learned** $\widehat{V}$, *(ii)* **the Frobenious absolute distance,** *(iii)* **the true positive rate, and** *(iv)* **the false discovery rate for the simulated partial prior setting with low, medium, and high uncertainty in Sec. 7.**

