# OpenReview forum: "Causal Abstraction Learning based on the Semantic Embedding Principle"
_ICML.cc/2025/Conference — ICML 2025 poster_

### Official Review · Reviewer_p62B · 2025-02-28

**Overall Recommendation:** 4

**Summary:**

The authors use a category theoretical formalization SCM and causal abstraction to derive and optimize similarity measures over the measurable spaces of observed distribution of the corresponding low- and high-level representations of data.

Towards this existing notions of $\alpha$-abstractions and constructive causal abstractions are leveraged to formulate a semantic embedding principle (SEP) that implies a right-inversive (=existence of a consistent high- to low-level map) abstraction which allows for the formalization of a distance measure between low- and high-level data. They formulate a "SEP-based CA Learning"-Problem and employ several optimizations to minimize distances between the probability measures of the high- and low-level distributions. To obtain unique solutions from observational data, prior knowledge about the functional form of the abstraction is induced to enforce the constructiveness of the abstraction. The authors reformulate the learning problem under the prior knowledge constraint as an Riemannian optimization problem within a Stiefel manifold.

The authors propose and derive several variants of a "Linear Semantic Embedding Principle Abstraction Learner" (LinSEPAL-ADMM, LinSEPAL-PG, CLinSEPAL) over smooth and non-smooth setups utilizing different optimization methods. Ablation experiments on synthetic data under the presence of full and partial prior knowledge are presented, showing robust performance of the CLinSEPAL method. The best performing LinSEPAL method is furthermore applied the real-world fMRI brain data, where the authors simulate the coarsening of brain regions of interest (ROI) between a raw data model and combination several ROI into macro ROI. In a second experiment, uncertainty about the grouping of some ROI is simulated such that the method has to decide the assignment of ROI to several macro ROI, resulting in moderate errors.

**Update after rebuttal.** The rebuttal and following answers were able to fully address my concerns regarding the role of $\mathbf{B}$, partial prior knowledge. The subsequent explanation and presented proof on bounded eigenvalues and constructive CAs were able to further strengthen the contributions of the paper. Considering this and the other reviewers discussion I have raised my score to an accept.

**Claims And Evidence:**

The authors argue that previous assumptions about availability of interventional data, structural knowledge of the SCM or functional form seem to be unrealistic or are infeasible to obtain. To overcome this problem, the authors work under the assumption of partial prior knowledge on the structure of factor assignments between the causal abstractions being available. To my understanding, the assumed prior knowledge is embedded in the form of a assignment matrix which indicates the relation between low- and high-level features, such that the CA learning problem transforms into a parameter regression problem under the presence of full prior knowledge. Even though the authors present a convincing case where full prior knowledge might available, learning the exact parameter assignment is arguably the core problematic of CA learning. In that regard, the authors present a final experiment on brain data where uncertainty about factor assignments is induced to the prior information. However, the property to recover correct factor assignments is only analyzed marginally. (See 'Methods And Evaluation Criteria' below).

Under the assumption of given prior information however, the presented algorithms seem yield convincing results, with LinSEPAL-ADMM and LinSEPAL-PG declining under partial knowledge (with missing factor assignments) and CLinSEPAL even obtaining robust performance in the latter case.

**Essential References Not Discussed:**

To the best of my knowledge, the authors cite and discuss relevant literature of causal abstraction learning and related category theoretical perspectives. While the authors utilize on a measure-theoretic view on acyclic SCM which is induced via recursive applications of push-forward measures, a more general view (that, e.g., allows for cyclic causal relations) might be taken with the use of transition probability kernels as formalized by Park et al. [1].



[1] Park, J., Buchholz, S., Schölkopf, B., & Muandet, K. (2023). A measure-theoretic axiomatisation of causality. *Advances in Neural Information Processing Systems*, *36*, 28510-28540.

**Experimental Designs Or Analyses:**

A first experiment is conducted over synthetically generated data. The data generation and chosen dimensions of the experimental setups are reasonable to demonstrate the correct workings of the algorithms. Algorithms are evaluated under full and partial evidence, meaning the omission of variable assignments in the prior knowledge. As discussed in 'Claims And Evidence', the algorithm should be tested to recover CAs in the light of uncertainty/multiple possible factor assignments / morphisms.

The experiment on coarsened brain regions of interest with full prior knowledge seems to follow domain specific knowledge and is sound and well conducted. With regard to the evaluation of partial knowledge the authors might indicate the specific prior knowledge matrix $\mathbf{B}$ that is used for the respective low, medium and high setups. Particularly, to indicate the number of assignments the algorithm can choose from in each setting.



Finally, I would like to recommend to still report results of all methods for all metrics in Fig. 4. Worse performing methods might still yield reasonable results --even if, for the wrong reasons-- and might give insights on whether those metrics can be used to make validate the validity of the prior knowledge.

**Methods And Evaluation Criteria:**

The authors report results for distribution distance in terms of KL-divergence, F1 score and Frobenious absolute distance of the learned map and analyze the correctness of the learned factor assignments ('learned morphisms'). The reported metrics are suited to asses the performance of the respective algorithms.

**Other Comments Or Suggestions:**

* The caption of figure 5 ("from Ind (right) to Prob (left)") does not match the labels in the figure.
* $\mathbf{V}^{\star}$ is referred to in line 203 before its actual definition in eq. 4.

**Other Strengths And Weaknesses:**

**Strengths**

The problem setup is well setup and derived. Intermediate steps over category theoretical are laid out clearly and follow naturally. Albeit the strong use of prior concepts and formalisms, all concepts are well described and the derivations in the paper are mostly self-contained.

The embedding of the problem into Stiefel Manifolds is an interesting insight and allows for the application of well-known Riemannian optimization methods.

The presented synthetic and real-world experiments confirm the robust application of the derived methods. Interestingly, the particular CLinSEPAL approach is demonstrated to perform well on real-world brain data, even under the presence of only partial knowledge.



**Weaknesses**

The main weaknesses have mainly been discussed in the prior sections and mostly regard the identification of morphisms in the case of partial prior knowledge and specifically concern the following points:

1) The presence of full prior knowledge might be an even stronger constraint than the knowledge about underlying DAG and possibly transforms the task into a mere parameter regression problem(?). The authors might elaborate (or compare) to the specific differences to a naive regression approach for the full prior information setting that simply regresses the non-zero entries of $\mathbf{B}$.
2) The arguably interesting case of identifying the right morphisms in the presence of only uncertain partial prior knowledge is not demonstrated over synthetic data. The paper might be improved by adding analysis on the effects of uncertain partial prior knowledge over synthetic data.
3) In regard to the former point and evaluation is presented on real-world brain data. However, details on the extend of uncertainty are not specified. The authors might add the utilized  $\mathbf{B}$ matrices or specify the number of non-zero entries per row per setting.

**Questions For Authors:**

I would like to ask the authors to reply to the points listed in the weaknesses. Furthermore, I have the following questions on (possibly minor details of) the paper:

1) Could the authors elaborate on the derivation of Eq. 3? Specifically, the log determinant term seems to only involve a single determinant, while KL divergence it is usually formulated as a quotient involving both of the arguments.
2) Could the authors, elaborate on the condition of constructive CAs being bound by the range of eigenvalues? The mentioned relation was not immediately obvious and was not explained further in the paper.
3) What is the reason for inducing $\mathbf{S}$ for the smooth problem setup? How is it different from learning $V$ directly (and why is this distinction not necessary in the nonsmooth approach)?



Minor:

* Possible misunderstanding or typo [l165 (right column); 'zeroing distance function']: Wouldn't $\chi_{\#}^{\alpha^*_{\mathcal{X}}}$ map $\chi^{\mathcal{l}}$ onto $\chi^{\mathcal{h}}$. Why can $\chi^{\mathcal{l}}$ be on both sides of the equation?

**Relation To Broader Scientific Literature:**

Being able to relating findings across different levels of causal abstractions is an important goal as it allows allows for the transformation of results between different approaches and yields models that allow to communicate low-level findings on a higher-level. Within the past few years, the automated learning of causal representation, therefore, has attracted increasing interest. The authors cite and discuss relevant approaches of the field which come --do to the inherent unidentifiability of factors from observational data-- with different particular assumptions. The proposed method(s) require prior knowledge on the function structure of abstraction, motivated on a measure theoretic and category theory theoretic formalization of SCM. To the best of my knowledge, the presented approach mark a novel contribution in terms of motivation and optimization.

**Theoretical Claims:**

The authors present semantic embedding principle (SEP) that implies the existence of a right-inversive causal abstraction (CA) between the high- and low-level data measures. Furthermore, the problem of SEP-based CA learning is formalized, which states as a goal, the learning of a CA which minimizes the distance between data representations and complies with the SEP. Both, the definition of SEP and the SEP-based CA learning, follow naturally from the category theoretic and measure theoretic formalizations. The problem formulation within a Stiefel manifold and the consequent formalization as a Riemannian optimization problem is laid out clearly and seems to be correct. While I am not an expert in the proposed optimization algorithm I followed the derivations of the CLinSEPAL (Sec. 5.2) method in Appendix J, which, to the best of my knowledge, seems to be without immediate errors.

---

> ### Author Rebuttal · Authors · 2025-04-01
>
> Thank you to the Reviewer for their effort, valuable comments, and appreciation of our work. We address below all the Reviewer’s concerns, also providing an additional theoretical contribution in [Q2]. We are happy to further discuss any additional concerns.
>
> _Claims_
>
> See [W1-W2].
>
> _Experimental Designs_
>
> [Uncertainty] See [W2].
>
> [Brain application] See [W3].
>
> [Fig.4] Agreed, we will report the results for our proposed methods that do not guarantee constructiveness in the Appendix.
>
> _Weaknesses_
>
> [W1] We do not assume the availability of aligned (i.e., jointly sampled) data from the SCMs. Hence, we cannot pose any regression problem.
>
> Assuming jointly sampled data are available (but again, that’s a different work  from ours), denoting by $h$ the number of nodes in the high-level model and considering full-prior knowledge, then one could solve separately $h$ linear regression problems subject to unitary $\ell_2$-norm constraints for the vectors of coefficients for complying with SEP. Thus, the individual problems are not mere regressions, rather nonconvex problems due to the constraints $\beta_i^\top \beta_i=1$, $i \in [h]$ and $\beta_i$ being the $i$-th _vector_ of coefficients.
>
> Thanks, we will add these details as a _Remark_.
>
> [W2] We believe there is a misunderstanding, probably due to lines 369-370 where we say _“by forgetting the mapping for 25%, 50%, and 75% of the variables”_. This passage in the text may be ambiguous to the reader and we’ll modify it. We clarify below this point that raised a major concern.
>
> In lines 433-435 we say: _“We then express this partial information via uncertainty over B, meaning that some rows of B have more than one entry equal to one”_. This is exactly what we do in the partial prior setting of the synthetic experiments in Sec.6.
>
> Indeed, by _“forgetting the mapping for 25%, 50%, and 75% of the variables”_ we mean that  25%, 50%, and 75% of the rows in $B$ have all entries equal to one. Thus, for each case, a specific fraction of nodes in the low-level considers all nodes in the high-level as plausible abstractions and our algorithms have to identify the correct high-level node for each low-level one.
>
> Hence, Fig.4 already shows the results for partial prior knowledge on synthetic data.
>
> [W3] Agreed, we will add the $B$ used in the experiments to App. L.
>
> _Comments_
>
> [C1] Thank you, we will correct the typo in Fig.5.
>
> [C2] Thanks, we will introduce $V^\star$ before.
>
> _Questions_
>
> [Q1] In Eq.(3) we look at KL as a function of $V$. The constant term is exactly $-\log{ \det{\Sigma^h}} - h$. We will specify this in line 202 2nd col.
>
> [Q2] Below we translate Remark 1 into a rigorous additional result we will add to the paper. It establishes a spectral characterization for linear CA and Gaussian measures, valid for any information-theoretic metric and $\phi$-divergence.
>
> _Theorem._ Let $\chi^\ell \sim N(0_\ell, \Sigma^\ell)$, $\chi^h \sim N(0_h, \Sigma^h)$, where $\Sigma^\ell$ and $\Sigma^h $ are positive definite and $\ell>h$. Denote by $0<\lambda\_1\leq…\leq \lambda\_\ell$ the eigenvalues of $\Sigma^\ell$, and by $0<k\_1\leq…\leq k\_h$ those of $\Sigma^h$. If a linear CA $V \in \text{St}(\ell,h)$ complying with SEP from $\chi^\ell$ to $\chi^h$ exists, then
> $$
> \lambda\_i \leq k\_i \leq \lambda\_{i+\ell-h}, \forall  i \in [h]. \tag{1}
> $$
> _Proof._ If a linear CA $\mathbf{V} \in \mathrm{St}(\ell,h)$ exists, then $\chi^h=\varphi\_\text{push}^V(\chi^\ell)$. Thus the kernel of any information-theoretic metric or $\phi$-divergence is nonempty, and $V^\top \Sigma V=\Sigma^h$ implying the eigenvalues of $V^\top \Sigma V$ are those of $\Sigma^h$. By the Ostrowski’s theorem for rectangular $V$ (cf. Th. 3.2 in Higham & Cheng, 1998) we have
> $$
> k_i = \theta_i \mu_i, \quad i \in [h]; \tag{2}
> $$
> where
> $$
> \lambda\_i \leq \mu\_i \leq \lambda\_{i+\ell-h}, \tag{3}
> $$
> and
> $$
> \text{eigvls}(V^\top V)\_1 \leq \theta_i \leq \text{eigvls}(V^\top V)\_h . \tag{4}
> $$
> Since $V \in \text{St}(\ell,h)$, by (4) $\theta_i=1$ for each $i \in [h]$. Substituting the latter into (2), we get $k_i=\mu_i$, thus obtaining (1) by (3).
>
> _Ref.:_ Higham, N. J., & Cheng, S. H. (1998). Modifying the inertia of matrices arising in optimization. Linear Algebra and its Applications, 275, 261-279.
>
> [Q3] The matrix $S$ guarantees a constructive CA. Consider the partial prior knowledge case where $B$ has more than a single 1 per row. By disentangling the support $S$ and the coefficients $V$ we can learn and enforce the support of CA to be constructive through the second and third constraints in Eq. (5).
>
> In the nonsmooth case we propose a simpler unconstrained formulation at the price of losing constructiveness guarantees for CA. In fact, we simply penalize entries in $V$ corresponding to zeros in $B$. Clearly, this does not guarantee the constructiveness of $V$, especially in the case of partial prior knowledge.
>
> [Minor] Thanks, on the LHS should be $\chi^h$.

---

> > ### Comment · Reviewer_p62B · 2025-04-02
> >
> > Thank you for the detailed answers regarding my questions. I believe that all comments on the relation to regression/non-convex optimization, the comment on the KL term in Eq. 3 and the clarifications on the role of $S$ will strengthen the paper.
> >
> > **W2:** I was indeed under the assumption that the statement in line 369 was implying additional restrictions on the already given partial knowledge. Thank you for clearing up this point.
> >
> > **Q2:** While I have to admit that I have not yet fully understood the theorem, I appreciate the effort. I would like to kindly ask the authors whether there exists an intuitive interpretation of the theorem. Particularly, regarding the condition under which for a linear constructive CA (might) exist, and particularly rather under which conditions it is guaranteed to not exist?
> >
> > Considering, that causal abstraction learning is an inherently difficult problem to solve in general, I do not share the view of the other reviewers on the linearity of the abstractions being a downside of the approach. In the light of the other ongoing discussions I have raised my score to a weak accept for now.

---

> > > ### Author Response · Authors · 2025-04-04
> > >
> > > We are pleased that our response has addressed the Reviewer's concerns, and we sincerely appreciate their acknowledgment that learning a causal abstraction is a complex challenge, even in the linear case. We greatly value the Reviewer's constructive approach and believe that the final version will be an improvement over our initial submission. \
> > > Below is a geometric intuition for the theorem, leading to the derivation of a necessary condition for the existence of a causal abstraction between $N(0, \Sigma^\ell)$ and $N(0, \Sigma^h)$. We hope it will aid in the understanding and assessment of our results.
> > >
> > > __[Covariance as an ellipsoid.]__ In the theorem we consider w.l.o.g. zero-mean Gaussian distributions. Therefore, the causal information lies in the variance of the distributions. As is well known, we can imagine the covariance matrices $\Sigma^\ell$ and $\Sigma^h$ as $\ell$-dimensional and $h$-dimensional ellipsoids, respectively.\
> > > __[Role of eigenvectors and eigenvalues.]__ The eigenvectors identify the axis of the ellipsoids, and the square root of the eigenvalues the length of each axis. Thus, the eigenvectors form two bases of $\mathbb{R}^\ell$ and $\mathbb{R}^h$, viz. $U^\ell$ and $U^h$, respectively. In particular, the eigenvectors are the columns of these matrices.\
> > > __[Projection of the low-level ellipsoid.]__ Consider now $V \in St(\ell, h)$. The columns of $V$ are also orthonormal and therefore define the basis of an $h$-dimensional subspace of $\mathbb{R}^\ell$. When $V$ is applied to $\Sigma^\ell$, the eigenvectors in $U^\ell$ are projected onto the latter basis. At this point, there are two aspects to notice:
> > > * **[Eigenvectors (axis) of the projected ellipsoid as a combination of those in $U^\ell$.]** First, the eigenvectors for the projected abstract measure $N(0, V^\top \Sigma^\ell V)$ will not simply be the projections of the eigenvectors of $\Sigma^\ell$ onto the subspace, but can still be written in terms of the eigenvectors in $U^\ell$. This can be seen simply by considering the eigendecomposition. We have $\Sigma^\ell = U^\ell \Lambda U^{\ell^\top}$ implying $V^\top \Sigma^\ell V = V^\top U^\ell \Lambda U^{\ell^\top} V = Q^\top \Lambda Q$, where we defined $Q = U^{\ell^\top} V$. Specifically, $Q$ expresses the basis identified by $V$ in terms of linear combinations of the eigenvectors in $U^\ell$. Consequently, the new eigenvectors of the subspace, which are written in the basis $V$, can also be written as linear combinations of those in $U^\ell$.
> > > * __[Variance cannot be increased by the projection.]__ Second, these new eigenvectors identify an $h$-dimensional ellipsoid that is, in a geometric sense, “contained” in the original $\ell$-dimensional ellipsoid. Since the projection $V \in St(\ell,h)$, it defines a contractive projection. Thus it cannot create additional variance but only combine (redistribute) existing one. This means that the variance along any direction in the subspace - determined by an eigenvalue of $V^\top \Sigma^\ell V$ - is a weighted combination of the variances (eigenvalues) in the original space, with the weights given by the entries of $V$ (which sum up to one in a squared sense). Consequently, each axis (direction) of the projected ellipsoid has a length (variance) that is bounded both below and above by the lengths of the axes of the original ellipsoid. Since the variance relates to the eigenvalues as discussed above, this means the eigenvalues of $V^\top \Sigma^\ell V$ must lie within an interval determined by the minimum and maximum eigenvalues of $\Sigma^\ell$.
> > >
> > > **[Projected ellipsoid coincides with the high-level one for optimal V.]** Let us now consider $V$ to be the optimal causal abstraction that we assume exists. From the spectral point of view, $V$ aligns the eigenvectors of the projected ellipsoid with those of $\Sigma^h$. Therefore, it is possible for us to derive a necessary condition for the existence of the optimal abstraction by looking at the spectral decomposition of $\Sigma^h$ as if it were that of the optimal projection $V^\top \Sigma^\ell V$.\
> > > **[Additional contribution of the theorem and answer to Reviewer’s question.]** Our theorem does not simply state that the eigenvalues of $\Sigma^h$ lie within the range determined by those of $\Sigma^\ell$ (as indicated by Remark 1, directly stemming from the second point above). Indeed, the theorem characterizes the length of each of the new axes of the projected ellipsoid (which for optimality we can interpret as the ellipsoid of $\Sigma^h$) in terms of the length of the old axes (eigenvectors in $U^\ell$), providing a more precise necessary condition for the existence of a causal abstraction between $N(0, \Sigma^\ell)$ and $N(0, \Sigma^h)$. Specifically, sorting the axis by their lengths (square root of eigenvalues), the length of the $i$-th new axis must lie between the lengths of the $i$-th and $(i+\ell-h)$-th old axis identified by the columns of $U^\ell$ (cf. Eq.(1) in the theorem).

---

### Official Review · Reviewer_rKnp · 2025-03-09

**Overall Recommendation:** 3

**Summary:**

This paper introduces a framework for learning causal abstractions (CA) when structural causal models (SCMs) are unknown, interventional data is unavailable, and observational data is misaligned. The authors proposed the Semantic Embedding Principle (SEP), which helps to reconstruct the relationship between the low-level and high-level causal variables. In particular, this paper focuses on linear CA problem. The linear CA problem can be formulated as a Riemannian optimization problem on the Stiefel. Various optimization algorithms are proposed to solve this problem.

**Claims And Evidence:**

Claims are supported by evidence.

**Essential References Not Discussed:**

I think the author should at least discuss the following two papers [1,2]. These papers study one application of causal abstraction in the literature.


[1]Geiger, Atticus, et al. "Causal abstractions of neural networks." Advances in Neural Information Processing Systems 34 (2021): 9574-9586.
[2]Geiger, Atticus, et al. "Finding alignments between interpretable causal variables and distributed neural representations." Causal Learning and Reasoning. PMLR, 2024.

**Experimental Designs Or Analyses:**

I checked the design of the experiments.

**Methods And Evaluation Criteria:**

The proposed method and evaluation criteria make sense.

**Other Comments Or Suggestions:**

1. It would be helpful if the author could provide some background knowledge on category theory, which can help the readers understand this paper better. Most readers may not be familiar with category theory.


2. Line 257: the paper states that rows of the support $(B\odot S)^T$ must sum up to one, but later in (iii) "the columns of the support $B\odot S$" must contain at least a one. Wouldn't the columns of $B\odot S$ be the rows of $(B\odot S)^T$ ? It seems to me (ii) already implies (iii).


3. The hyperlink (NA1)-(NA5) and (A1) do not work. I am kind of confused what is non-assumption.

**Other Strengths And Weaknesses:**

Strength: The paper introduces the Semantic Embedding Principle (SEP), a novel and well-motivated approach to ensuring that high-level causal knowledge is faithfully embedded in low-level models. Unlike prior works that rely on full knowledge of SCMs, known DAG structures, or interventional data, this method operates when only partial prior knowledge is available. The methods are empirically validated on synthetic data as well as resting-state fMRI data, showcasing their effectiveness in neuroscience applications.

Weakness: This paper focuses on the linear CA problems. Since many real-world causal systems are inherently nonlinear, this limits the immediate applicability of the approach. In the experiment part, there is no direct comparison with base line methods.

**Questions For Authors:**

I appreciate it if the author could clarify the following problems.

1.  It seems to me that I do not need category theory to understand the SEP. Could the author explain how the category theory formulation lead to this principle? Or what is beneficial using the category theory language?

2. In the experiment part, the authors compare the performance of three proposed methods. Can the authors compare their methods with existing methods [1,2]? I understand the setting may be a bit different. It is also beneficial to see without prior knowledge how good can the proposed method perform.

3. If I understand this paper correctly, this paper is doing CA based on observational data. In practice, what we care about most is what would happen if we do interventions, so one key question here is after learning the abstraction map using the proposed method, can the abstraction map stay consistent under interventions?

**Relation To Broader Scientific Literature:**

This paper studies causal abstraction without interventional data and prior knowledge about the SCM. This is an important problem because many current causal abstract algorithm requires some kind of conjecture about the underlying SCM [1,2]. Without prior knowledge, these methods fail.

**Theoretical Claims:**

I did not check the soundness of all the proof.

---

> ### Author Rebuttal · Authors · 2025-04-01
>
> We thank the Reviewer for their effort, valuable comments, and appreciation of our work. We address below the Reviewer’s concerns in a concise manner due to text limit. We are happy to further discuss any additional concerns.
>
> __Weaknesses__
>
> __[W1]__
> The Reviewer is right, in real-world applications, systems often display nonlinear interactions. However, the weakness they point out does not apply to our work since _we do not assume any linearity of the low- and high-level SCMs_.
>
> Indeed, from both the theoretical and learning perspectives, (i) the category-theoretic treatment of SCM, (ii) SEP in Def. 4.1, and (iii) the SEP-based CA learning problem in Prob.1 are general and do not make any assumptions on the functional forms of the involved SCMs.
>
> It’s only from the application perspective that we decline Prob.1 to the case of linear CA. However, the linearity of the CA does not imply the linearity of the causal modules of the SCMs (cf. “Counterexample” in reply to Reviewer 232e).
>
> __[W2]__ We agree with the Reviewer on the importance of the comparison with baselines. Unfortunately, we were not able to find any baseline suitable for a fair comparison as we dropped many assumptions of existing methods, viz. (NA1)-(NA5), as the Reviewer acknowledges. Requirements:
> * Zennaro et al. (2023): (NA1), (NA2)
> * Felekis et al. (2024), $(\tau,\omega)$ CA setting: (NA2),(NA3)
> * Massida et al. (2024): (NA4),(NA5)
> * Kekic et al. (2024) perform _targeted reduction of an SCM_ in the $(\tau,\omega)$ abstraction setting, a different task than ours. Require (NA3) (NA4)
> * Dyer et al. (2024) consider a different problem within the setting of $(\tau,\omega)$-abstraction. Require (NA3)
> * Geiger et al. (2021) and (2024) leverage CA formalism to rigorously analyze explainability of neural networks, a different task from ours. \
> Given a neural network as a low-level SCM, the assumption is that there exists a high-level interpretable SCM. The goal is to evaluate the alignment between the two. Here, “low-level” means “black-box”; “high-level” refers to “human-understandable” SCMs built from theoretical and empirical modeling work. Differently, in our work “low-” and “high-level” mean “micro”(fine-grained) and “macro”(coarse-grained). It’s not a mere difference of terminology. Consider the SCMs $\mathcal{B}$ and $\mathcal{N}$ in Sec.3 of Geiger et al. (2024): $\mathcal{B}$ is a high-level model for $\mathcal{N}$ although both have the same structure. In our work, since there is no difference in interpretability between SCMs, the previous setting would lead to a contradiction, since it would mean considering as a (macro) abstraction of an SCM a rotation of it.  \
> Further, Geiger et al. (2021) and (2024) use _interchange intervention training_ (IIT) objectives, developed for neural networks and requiring the possibility to perform interventions over both the black-box and human-interpretable models. Citing from Geiger et al. (2024): _“interchange intervention (also known as activation patching), in which a neural network is provided a ‘base’ input, and sets of neurons are forced to take on the values they would have if different ‘source’ inputs were processed”_. Due to (NA1)-(NA3), it is not possible to adapt IIT to our setting.
>
> __Comments__
>
> __[S1]__ We devoted App.B to “Category theory essentials”. If the paper will be accepted, we will add a concise background to the main using the extra page.
>
> __[S2]__ Thanks, there is a typo in the text. The corrected version is:
> _“... the columns of the support $(B \odot S)^\top$ must sum up to one, [...] the rows of the support  $(B \odot S)^\top$ must contain at least a one.”_
>
> __[S3]__ Thanks, we will fix the link. A non-assumption is an assumption made by existing methods that we do not make. We will specify it better in the paper.
>
> __Questions__
>
> __[Q1]__ We work purely at the semantic (distributional) level dropping (NA1)-(NA5). Category theory (CT) is applied to isolate the distributional layer of the SCMs. This has an impact on SEP in Def. 4.1 since CT requires the involved mappings to be measurable. Finally, CT is used to generalize the existing CA $\alpha$-framework – which is posed in category-theoretic terms – into our setting.
>
> __[Q2]__ Please refer to [W2]. We can add the reported discussion in our manuscript to motivate the absence of a comparison with baselines. Additionally, among future works we will add the investigation of SEP in the setting of Geiger (2021, 2024). It is an intriguing research question that could lead to jointly aligning and compressing human-understandable models to AI ones in a principled manner.
>
> __[Q3]__ As stated in lines 194-195, _“Only if we identify the true constructive abstraction, we are guaranteed interventional consistency.“_. We plan to investigate in which cases interventional consistency can be guaranteed without additional assumptions (cf. Sec. 8).
>
> _References_: Already in the paper. Geiger (2021) and (2024) are [1,2] of the review.

---

> > ### Comment · Reviewer_rKnp · 2025-04-04
> >
> > Thanks for clarifying my questions. I have raised the score to 3. I have some comments for the authors.
> >
> > - From definition 2.3 and my understanding, causal abstraction aims to find the triple $<\mathcal{R}, m, \alpha>$. In Problem 1, the authors assume that the mapping $m$ is known, which may be restrictive. In many cases, the most difficult part is to find the corresponding relationship $m$.
> >
> > - I also want to point out that the setting in Geiger et al. (2021) and (2024) is actually similar. In their setting, they consider a neural network as a low-level model and a high-level causal model. What they try to do is to find an alignment (can be formulated as an $\alpha$-abstraction in your terminology). While I understand that their approach is different than yours, I would say the problem is similar.

---

> > > ### Author Response · Authors · 2025-04-04
> > >
> > > We thank the Reviewer again for their appreciation and constructive comments leading to an improvement over our initial submission. We believe there is still room to clarify some key aspects of our work, and we hope the discussion below will aid in the understanding and assessment of our results.
> > > 1. Point 1 was raised by Reviewer p62B as well (see point 1 in their reported weaknesses), who, after our response, recognized CA learning as a difficult challenge even in the linear case with full-prior knowledge and jointly sampled data (a requirement we drop, viz. NA5). Basically, the reason is that CA learning results in a nonconvex learning problem even in the latter case that seems trivial only at first sight. We refer the Reviewer to our discussion with Reviewer p62B. Having said that, there are two points to be highlighted here:
> > >     * First, our approach only assumes partial prior knowledge of $m_{\mathcal{X}}$, allowing for the realistic scenario in which users may have no prior information about certain structural maps. We evaluated our methods in this challenging setting using both synthetic and real-world brain data, and they demonstrated robust performance across these contexts (see Sec.6 and Sec.7).
> > >     * Second, we believe assumptions should not be judged in absolute terms, by stripping them away from the context. Let us consider the works of Geiger et al. dealing with DNNs. In the case of DNNs, it is reasonable – and smart – to exploit the full-knowledge of and access to the SCMs since they are inherently provided by the application setting. There is also no issue related to feasibility, cost and ethics regarding interventions, which can be performed on those known SCMs essentially for free. Conversely, since DNNs are black-boxes, it would be unreasonable for them to grant our assumption (A1) connecting some nodes in the low-level to the high level. Indeed, the discovery of that connection is in their aim in the interpretability setting. In other words, Geiger et al. aim at explaining DNNs implementing the causal abstraction analysis. \
> > > By contrast, in domains such as neuroscience and finance, the situation is markedly different. We do not have full-knowledge of and access to the SCMs, and we can rarely make assumptions on them. Thus it is important to drop (NA1), (NA2), and (NA4). Since we do not know the SCMs, we cannot generate data from them as for DNNs. Moreover, obtaining jointly sampled data can be hard (NA5) due to, for instance, feasibility and privacy issues. Think to traders operating in financial markets, or neuroscience teams acquiring data from patients. Also, obtaining interventional data can be problematic for ethical, feasibility, and cost reasons (NA3). This is a well-known problem within the causality community, and motivated the development of causal discovery methods over the years. Conversely, what is reasonable and often feasible in these application areas is leveraging domain-specific knowledge that can be translated into partial information about $m_{\mathcal{X}}$. As shown in the paper, in neuroscience we can leverage the way brain atlases are built. Similarly, in finance, one can use the knowledge that broad industry portfolios are constructed from finer-grained indexes (as formalized in the Global Industry Classification Standard). This is the rationale behind assumption (A1), making it relevant and justifiable across a range of domains.
> > >
> > > 2. Concerning point 2 , we agree with the Reviewer that the tasks are related, although there are some fundamental differences discussed in our previous rebuttal and point 1 above. We thank the Reviewer again for their constructive comments about the work of Geiger et al. Indeed, the addition of these references as well as a discussion on the potential interplay between our and their work will enrich the paper and broaden its relevance for the ML community. We will add this material to the final version as already agreed. Nevertheless, we wish to reiterate that, in its current form, there is a fundamental mismatch in the required inputs that prevents a direct and fair empirical comparison between our methods and those of Geiger et al.

---

### Official Review · Reviewer_Mguo · 2025-03-13

**Overall Recommendation:** 3

**Summary:**

This paper addresses the challenge of learning causal abstractions (CAs) between structural causal models (SCMs) at different resolutions, a critical task for bridging causal evidence across scales (e.g., molecular vs. organism-level processes). The authors propose the Semantic Embedding Principle (SEP), which enforces that embedding a high-level SCM into a low-level one and abstracting back should reconstruct the original high-level model. They formalize SEP using a category-theoretic framework, decoupling structural and functional components of CAs. For learning, they focus on linear CAs under partial prior knowledge (assumption A1), framing the problem as Riemannian optimization over the Stiefel manifold. They develop optimization methods (LinSEPAL-ADMM, LinSEPAL-PG, CLinSEPAL) tailored to smooth/nonsmooth objectives in Gaussian settings, validated on synthetic and neuroimaging data.

**Claims And Evidence:**

The claims are generally supported by providing some definition. However, the paper still lacks clarity when introducing those new terminologies, e.g., the introduction of causal abstraction.

**Essential References Not Discussed:**

Yes, the paper would benefit from a more thorough discussion of causal representation learning methods, such as CausalVAE [1] and SCM-VAE [2], which also aim to learn causal structures from data but focus on disentangled representations rather than multi-resolution abstractions.

**Experimental Designs Or Analyses:**

The experimental design is generally sound but the experiments lacks discussion with the related causal representation methods.

**Methods And Evaluation Criteria:**

Yes, the (LinSEPAL-ADMM, LinSEPAL-PG, CLinSEPAL) are well-suited for the problem.

**Other Comments Or Suggestions:**

N/A

**Other Strengths And Weaknesses:**

Strengths
- A causal abstraction learning method is proposed with the integration of Riemannian optimization.

- Experiments on synthetic and real-world brain data showcase the feasibility of the approach under varying levels of prior knowledge.

Weaknesses:

- As for clarity, I think in the start of paper, it should first clarify what is causal abstraction and how is it related and difference to the similar concept of causal representation learning.
- Moreover, this paper fails to properly motivate why we need the causal abstraction and how it can be used in terms of application and how other methods (e.g., the causal representation learning methods [1-2]) fails on those tasks.
- As for the causal abstraction, can you show the identifiability of the causal abstraction? Moreover, is the the structure of the SCM need to be priorly specified for causal abstraction?

[1] Yang, Mengyue, et al. "Causalvae: Disentangled representation learning via neural structural causal models." Proceedings of the IEEE/CVF conference on computer vision and pattern recognition. 2021.
[2] Komanduri, Aneesh, et al. "Scm-vae: Learning identifiable causal representations via structural knowledge." 2022 IEEE International Conference on Big Data (Big Data). IEEE, 2022.

**Questions For Authors:**

See the weankesses above.

**Relation To Broader Scientific Literature:**

The key contribution is somewhat limited in the broader literature on causal reasoning and representation learning. While the introduction of SEP and its formalization using category theory builds on prior work in causal abstraction (Rubenstein et al., 2017; Beckers & Halpern, 2019) and extends it to a learning framework, the paper could better situate itself relative to causal representation learning methods, which address similar challenges but from a different perspective.

**Theoretical Claims:**

The theoretical claims seems sound. However, the paper does not provide detailed proofs for the identifiability of causal abstractions.

---

> ### Author Rebuttal · Authors · 2025-04-01
>
> We thank the Reviewer for their effort and valuable  comments. We address below the Reviewer’s major concerns on the relationship between CA learning and CRL, and we are happy to further discuss to clarify any additional points.
>
> __Claims And Evidence__
> Causal abstraction is intuitively introduced at the beginning of Sec.1, from line 46 1st col to line 49 2nd col; and formally in Def. 2.3. However, we are happy to improve our manuscript as per Reviewer suggestion by adding the proposed text in italic reported below, within our reply to “Weaknesses”.
>
> __Theorethical Claims__
> We do not make any claim about the _identifiability_ of the causal abstraction for the reasons reported below in [W3.2].
>
> __Experimental Designs Or Analyses__
> Please refer to [W1-W2] below. CA and CRL tackle different learning tasks, although a comparison between them is valuable to further improve the paper.
>
> __Relation To Broader Scientific Literature__
> Please refer to [W1-W2] below. We believe that our work is correctly placed within the literature, although considering the comparison with CRL is valuable and planning to add it in the revised version of the manuscript.
>
> __Essential References Not Discussed__
> Please refer to [W1-W2] below.
>
> __Weaknesses__
>
> __[W1-W2]__ We agree with the Reviewer that an additional comparison between CA learning and CRL within Sec.1 would improve the paper, specifically aiding the reader in setting apart CA learning from CRL. We propose to add the following discussion in the revised version:
>
> _Causal abstraction (CA) learning aims at learning a mapping between two different SCMs, for instance, the architecture of a neural network and a human-interpretable causal model [2], or an SCM of brain regions of interest (ROIs) and one of brain functional activity (see Sec.7)._
>
> _Within CA literature, it is usual to distinguish between low- and high-level variables (or SCMs). Although the same adjectives are usually employed also in causal representation learning (CRL, [3]) literature, they convey different meanings in the two research fields._
>
> _In CA, both low- and high-level refer to endogenous variables being causal and observed. Specifically, the latter are said to be causal since they relate to each other within the SCM, and are the relevant variables for interventions and counterfactual reasoning.
> Instead, in CRL, the low-level variables are observed but not causal, that is, mere mathematical functions of high-level causal but unobserved variables, where causal has the same meaning as above. As an example, the low-level variables could be the pixels of an image, whereas the high-level ones are concepts related by an SCM [4, 5]. Additionally, the high-level variables could also not be labeled [3]._
>
> _Consequently, also the goal of CRL is deeply different from that of CA: Given the low-level variables, CRL algorithms aim at learning (i) the high-level variables and (ii) the causal structure underlying these variables. In brief, while CRL extracts a meaningful causal representation from non-causal data to improve model performance and interpretability [4, 5]; CA learns mappings between already meaningful representations to enable causal knowledge transfer and communication between SCMs working at different levels of abstractions._
>
> __[W3.1]__ Identifiability of CA is not key as identifiability of causal structures in CRL and causal discovery. In CA learning, we mainly care about (structural and distributional) interventional consistency. It is well known that there might exist multiple causal abstractions between low- and high-level SCMs (see Example 5.2 in [1] where symmetry allows for multiple interventionally consistent causal abstractions).
>
> __[W3.2]__ As we drop (NA1), (NA2), and (NA4), our work shows that it is possible to learn causal abstractions without assuming any knowledge of the underlying low- and high-level SCMs.
>
> _References_
>
> [1] Zennaro, F. M., Bishop, N., Dyer, J., Felekis, Y., Calinescu, A., Wooldridge, M., & Damoulas, T. (2024, September). Causally Abstracted Multi-armed Bandits. In Uncertainty in Artificial Intelligence (pp. 4109-4139). PMLR.
>
> [2] Geiger, A., Lu, H., Icard, T., & Potts, C. (2021). Causal abstractions of neural networks. Advances in Neural Information Processing Systems, 34, 9574-9586.
>
> [3] Schölkopf, B., Locatello, F., Bauer, S., Ke, N. R., Kalchbrenner, N., Goyal, A., & Bengio, Y. (2021). Toward causal representation learning. Proceedings of the IEEE, 109(5), 612-634.
>
> [4] Yang, M., Liu, F., Chen, Z., Shen, X., Hao, J., & Wang, J. (2021). Causalvae: Disentangled representation learning via neural structural causal models. In Proceedings of the IEEE/CVF conference on computer vision and pattern recognition (pp. 9593-9602).
>
> [5] Komanduri, A., Wu, Y., Huang, W., Chen, F., & Wu, X. (2022, December). Scm-vae: Learning identifiable causal representations via structural knowledge. In 2022 IEEE International Conference on Big Data (Big Data) (pp. 1014-1023). IEEE.

---

> > ### Comment · Reviewer_Mguo · 2025-04-05
> >
> > Thanks for the clarification and I would like to raise the score to 3.

---

> > > ### Author Response · Authors · 2025-04-08
> > >
> > > We thank the Reviewer for their appreciation. We are glad that our rebuttal effectively addressed the Reviewer’s concerns.
> > >
> > >
> > > As the rebuttal period is nearing its end - and considering the new conference rules, this may potentially be our last message visible to the Reviewers - we would like to take this opportunity to provide a brief summary. Our goal is to encourage discussion among reviewers and support a fair and thorough evaluation of our work. **We thank again all the Reviewers for their constructive approach**.
> > >
> > >
> > > **All concerns raised were factually addressed in the rebuttal**, leading to score increases from Reviewers Mguo, rKnp, and p62B, whose initial assessments were negative. Reviewer 232e was already positive.
> > >
> > >
> > > In their initial reviews, the following strengths were noted:
> > > * **Novelty** of the Semantic Embedding Principle (SEP) for CA learning (Reviewer rKnp)
> > > * Its **clear motivation** and **presentation** (Reviewers rKnp and p62B),
> > > * The formulation of **Riemannian CA learning problems** on the Stiefel manifold (Reviewers 232e, Mguo, and p62B)
> > > * And the methods' **practical applicability**, made possible by relaxing five restrictive assumptions common to prior work across various domains
> > >
> > >
> > > The main concerns were addressed as follows:
> > > * Provided a **factual clarification** that our application setting does **not imply linearity** of the underlying SCMs (Reviewers 232e and rKnp)
> > > * **Integrated discussions** - outlined in our rebuttal - on:
> > >   - The distinction between CA learning and causal representation learning (Reviewer Mguo)
> > >   - The lack of suitable baselines in the existing literature (Reviewer rKnp)
> > >   - The comparison and interplay with the work of Geiger et al. (Reviewer rKnp)
> > > * **Resolved a misunderstanding** concerning the synthetic experiments (Reviewer p62B)
> > > * **Clarified** that nonconvexity arises in regression problems even under full prior knowledge and joint sampling from the SCMs (Reviewer p62B). This clarification led the Reviewer to recognize the inherent complexity of CA learning.
> > >
> > >
> > > Additionally, we proved a **new theorem** establishing a necessary (spectral) condition for the existence of a linear CA between Gaussian measures, and provided its **geometrical interpretation** (Reviewer p62B).

---

### Official Review · Reviewer_232e · 2025-03-14

**Overall Recommendation:** 3

**Summary:**

This paper formalises the problem of causal abstraction in category theory language, then it introduces the Semantic Embedding Principle (SEP). Intuitively, SEP states that if we go from a high level model to a low level and then abstracting back, one should get the initial high level model back, the way I understand it is that high level models should contain all information of the low level model that produced it. The authors make some general assumptions in causal abstraction into non-assumptions and using SEP they and certain assumptions they establish the problem as a Riemannian Optimisation problem which then they go on to test on synthetic data and then real data from brain networks.

**Claims And Evidence:**

From what I understood of the paper (and I have to admit I don’t think I fully understood it), I would say the claims are only partly supported. Let me elaborate. The authors claim that most of the previous work makes assumptions that are unacceptable in the causal abstraction setting and thus are not so applicable whereas their assumptions are based on information of the structure of CA. For example, the assumptions they use for the experiments are linear causal abstractions, constructability of the abstraction and Gaussianity of the errors. I don’t know how the authors feel about this, but I would say that these assumptions are as strong as assuming the functional form of the SCM.

In the end the optimisation ends up being similar to other research, namely the KL divergence between the pushforward of the low level distribution against the high level model.

**Essential References Not Discussed:**

The authors do a good job at referencing related literature. There is nothing very obvious that I believe they missed.

**Experimental Designs Or Analyses:**

I checked what is on the main document. They seem reasonable. Furthermore, I appreciate the application to brain networks as a real world scenario.

**Methods And Evaluation Criteria:**

Yes, their methods seem reasonable to me. Although as mentioned above, they seem very similar to what we already see in the abstraction literature with the backing of the Categorical formalisation and the exception of the need for interventional data, which they don’t assume have access to.

**Other Comments Or Suggestions:**

Please see the weaknesses for some potential discussion. Additionally, I would like to know what do the authors think is needed to change the assumption of having a linear abstraction to something more complex. That is, how easy is it to find a space that satisfies SEP that we can also optimise on?

**Other Strengths And Weaknesses:**

Strengths:
- I think the use of the categorical language to go beyond interventional consistency is interesting and has the potential to allow the application of causal abstraction in other areas like they show with the brain networks.
- The optimisation procedure and its description looks very good as well although admittedly the details are beyond my optimisation knowledge in relation also to the time I can invest into reviewing the paper.

Weaknesses:
- Unless I’m misunderstanding something in the paper the excessive claims, as discussed above seem to be the greatest weakness.

**Questions For Authors:**

None

**Relation To Broader Scientific Literature:**

The contribution seems relevant to the literature. It gives the categorical perspective from Rischel (2020) applicability.

**Theoretical Claims:**

Not in detail.

---

> ### Author Rebuttal · Authors · 2025-04-01
>
> We thank the Reviewer for their effort, valuable comments, and appreciation of our work. We address below the Reviewer’s concerns. We are happy to further discuss to dispel any additional concerns.
>
> __Claims And Evidence__
>
> __[Our claims]__ Our claims are highlighted in the “Contributions” paragraph of Sec.1. We support them both theoretically and empirically. We are open to reducing any eventual excessive claims as per Reviewer suggestion. At the moment, it is not clear to us which claim is problematic. We kindly ask the Reviewer to specify not supported or justified claims.
>
> __[Assumptions]__ We believe the Reviewer’s concern is centered on the assumptions behind our work. We clarify this point below.
>
> Regarding the SOTA, we do not believe – and do not state in the manuscript – that the assumptions of existing works are _unacceptable_. We say they are _restrictive_ to tackle CA learning in real-world applications (cf. line 60 1st col, lines 398-399 2nd col).
> Accordingly, we make only one assumption, viz. (A1), supported by empirical evidence (cf. Sec.7).
>
> Also leveraging (A1), we pose the general SEP-based CA learning problem (cf. Prob1). This problem does not assume any specific (i) functional form for CA, (ii) probability measures for the involved SCMs, and (iii) distance function for quantifying the misalignment (cf. lines 167 1st col - 170 2nd col). Prob.1 should be read as a learning paradigm for CA rather than a single learning problem.
>
> As an application, we decline Prob.1 to the case of (i) linear CA, (ii) Gaussian measures for the endogenous, and (iii) KL divergence, arriving at Prob.s 2 and 3. We remark that (i)-(iii) are not assumptions within our work, rather a particular case of Prob 1.
>
> Concerning the Reviewer’s statement, it’s unclear to us what they mean by “Gaussianity of the errors”. If error stands for exogenous variables, then we remark that we do not make any such assumption in our work. If they mean Gaussianity of the endogenous probability measures, then we provide below a simple counterexample showing that (nonlinear) functional forms other than linear for causal modules are compatible with the (i)-(iii). We will consider lognormal distributed variables as those are relevant in application domains such as quantitative finance: stock prices are considered to be log-normally distributed (Black and Scholes, 1973; Fama, 1965).
>
> __Counterexample.__ Denote by $U_i$ and $L_i$ the exogenous and endogenous variables of the low-level SCM. Denote by $V_j$ and $H_j$ the exogenous and endogenous of the  high-level SCM. Let $T:[0,1] \rightarrow [0,1]$ be a measure preserving map (e.g., $T(x)=1- |2x-1|$), $f$ and $g$ be the quantile function and CDF of two Gaussians,  $N(0, c_1^2+c_2^2+c_3^2)$ and $N(0, c_4^2)$, respectively.
>
> _Causal Abstraction complying with SEP_: $H_1=1/\sqrt{2} (L_1 + L_2)$, $H_2=L_3$.
>
> * Low-level SCM:
>   * Exogenous: $U_1, U_2, U_3$; each following $\log{N(0,1)}$
>   * Endogenous: $L_1=\log{U_1}; L_2=\log{U_2}; L_3= (f \circ T \circ g) (c_1 L_1 + c_2 L_2 + c_3 \log{U_3})$
>   * Observational distributions for the endogenous: $L_1 \sim N(0,1), L_2 \sim N(0,1), L_3 \sim N(0,c_4^2)$.
>
> * High-level observational distributions entailed by CA: $H_1 \sim N(0,1)$, $H_2 \sim N(0, c_4^2)$
> * High-level SCM:
>   * Exogenous: $V_1, V_2$; each following $\log{N(0,1)}$
>   * Endogenous: $H_1 = \log{V_1}, H_2 = (f \circ T \circ g)(d_1 H_1 + d_2 \log{V_2})$; with $d_1=\sqrt{ c_1^2 + c_2^2 }$ and $d_2=c_3$.
>   * Observational distributions for the endogenous: $H_1 \sim N(0,1)$, $H_2 \sim N(0, c_4^2)$
>
> Finally, the KL divergence is used as an objective function in Prob. 2 and 3, and not limited to any specific probability measure and functional form for the SCM.
>
> __[Relations to existing methods]__ We deliberately selected KL divergence for its relevance in ML applications. Furthermore we consider the application of KL to probability measures of different dimensionality, differently from previous work (Kekic et al., 2023; Dyer et al., 2024). Additionally, Zennaro et al. (2023) use a regularized Jensen-Shannon divergence, Felekis et al. (2024) an $\omega$-informed cost of transport with entropic and do-intervention regularization terms; Massida et al. (2024) perform OLS estimation between the low- and high-level SCMs data. None of the works above is similar to the Riemannian optimization problems, viz. Prob.s 2 and 3, posed in our linear CA application.
>
> _Additional Ref.s To Those in The Paper_
>
> [1] Black, F., & Scholes, M. (1973). The pricing of options and corporate liabilities. Journal of Political Economy, 81(3), 637-654.
>
> [2] Fama, E. F. (1965). The behavior of stock-market prices. The Journal of Business, 38(1), 34-105.
>
> __Weaknesses__
>
> See “Claims” above.
>
> __Comments or Suggestions__
>
> Def. 4.1 and Prob. 1 are general and suitable for any CAs. SEP requires the CA having a right-inverse (cf. lines 175-176 1st col), this is the condition to be enforced during the optimization process.

---

### Decision · Program_Chairs · 2025-05-01

**Decision:**

Accept (poster)

**Comment:**

The paper develops the semantic embedding principle to learn causal abstractions without interventional data. The semantic embedding principle is formalized through category theory, and formulate a tractable learning problem for the linear causal abstraction based on the Steifel manifold. Experiments are conducted on synthetic data and fMRI. The paper develops heavy machinery, which was checked by reviewers (see Reviewer p62B). The main concern was the role of the partial prior knowledge which was clarified in the rebuttal.